# 🦜PaPaGei: Open Foundation Models for Optical Physiological Signals

**Arvind Pillai**[2]***Dimitris Spathis**[1,3]**, Fahim Kawsar**[1,4]**, Mohammad Malekzadeh**[1]
[1]Nokia Bell Labs, Cambridge, UK, [2]Dartmouth College, NH, USA,
[3]University of Cambridge, UK, [4]University of Glasgow, Scotland, UK

## Abstract

Photoplethysmography (PPG) is the leading non-invasive technique for monitoring biosignals and cardiovascular health, with widespread adoption in both clinical settings and consumer wearable devices. While machine learning models trained on PPG signals have shown promise, they tend to be task-specific and struggle with generalization. Current research is limited by the use of single-device datasets, insufficient exploration of out-of-domain generalization, and a lack of publicly available models, which hampers reproducibility. To address these limitations, we present PaPaGei, the first open foundation model for PPG signals. The model is pre-trained on over 57,000 hours of data, comprising 20 million unlabeled PPG segments from publicly available datasets. We introduce a novel representation learning approach that leverages domain knowledge of PPG signal morphology across individuals, enabling the capture of richer representations compared to traditional contrastive learning methods. We evaluate PaPaGei against state-of-the-art time-series foundation models and self-supervised learning benchmarks across 20 tasks from 10 diverse datasets, spanning cardiovascular health, sleep disorders, pregnancy monitoring, and wellbeing assessment. Our model demonstrates superior performance, improving classification and regression metrics by 6.3% and 2.9% respectively in at least 14 tasks. Notably, PaPaGei achieves these results while being more data- and parameter-efficient, outperforming models that are 70× larger. Beyond accuracy, we examine model robustness across different skin tones, establishing a benchmark for bias evaluation in future models. PaPaGei can serve as both a feature extractor and an encoder for multimodal models, opening up new opportunities for multimodal health monitoring[1].

## 1 Introduction

Photoplethysmography (PPG), a non-invasive optical sensing technique, is widely used to monitor cardiovascular health and physiological signals in both clinical and consumer health applications (Charlton et al., 2023). From hospital pulse oximeters to smartwatches, PPG enables continuous health monitoring in various settings, bridging acute medical care and long-term health management. PPG signals help in tracking a diverse range of health indicators, including cardiovascular health, blood pressure, mood, and sleep disorders (Sadad et al., 2022; Ave et al., 2015; Reiss et al., 2019; Liang et al., 2018a; Haddad et al., 2021; Schrumpf et al., 2021). Despite its widespread adoption, PPG poses substantial challenges for machine learning applications. A primary obstacle is the high cost of data annotation, which requires specialized domain expertise. This challenge is particularly pronounced in consumer health applications, where varying sensing conditions and diverse user populations create additional complexity. PPG signals are susceptible to noise and motion artifacts (Afandizadeh Zargari et al., 2023), as well as inherent variability due to factors like skin tone and body composition (Bent et al., 2020). These complicate the development of generalizable ML models for PPG. Consequently, existing PPG datasets are often small, task-specific, and limited in their generalizability, posing a major obstacle to the development of robust and widely applicable models that could fully leverage the potential of PPG technology.

---

*Work has been done during the author's internship at Nokia Bell Labs.
[1]Models, data, and code are available at: github.com/nokia-bell-labs/papagei-foundation-model

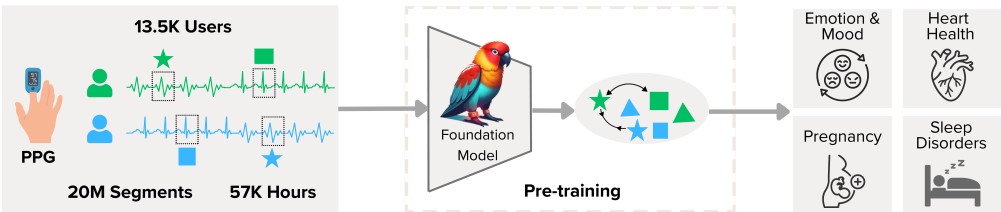

Figure 1: PАPАGEI Overview. We curate public datasets of diverse PPG signals, and train a foundation model leveraging a novel morphology-aware contrastive learning approach. To evaluate its effectiveness, we apply the embeddings generated by PАPАGEI to 20 tasks from 10 different datasets.

The PPG domain, unlike language or vision domains, lacks general-purpose foundation models (FMs), with most current works focused on single-dataset task-specific models. Although PPG can detect vital signs like heart rate variability and blood oxygen saturation, the absence of generalizable pre-trained models limits progress (Abbaspourazad et al., 2023). Despite the ongoing challenges of acquiring large-scale, high-quality data, recent expansions in diverse PPG datasets have created new opportunities (Johnson et al., 2016; Zhang et al., 2018; Lee et al., 2022). To address these challenges, we introduce **PАPАGEI**, a set of robust, pre-trained models capable of serving as a backbone for various PPG-related tasks, capturing rich PPG representations through large-scale pre-training.

The key **contributions** of PАPАGEI are:

**(1) Large-scale pre-training for PPG signals**: To our knowledge, PАPАGEI is the first open foundation model pre-trained on PPG signals, using 57,000 hours of data from 20 million signals sourced entirely from public datasets. This establishes a new benchmark for large-scale model development in wearable and clinical health monitoring.

**(2) PPG-aware self-supervised learning (SSL) framework**: We introduce a novel SSL framework with a unique PPG signal morphology augmentation module. Our approach optimizes agreement between PPG signals with similar blood volume changes while pointing the model to pay attention to the changes around the systolic peak and dicrotic notch (key PPG markers).

**(3) Comprehensive evaluation across diverse out-of-domain health tasks**: We evaluate PАPАGEI across 20 tasks, including cardiovascular health, sleep disorders, pregnancy monitoring, and overall well-being. Our results show that the model embeddings contain rich and predictive information applicable to various health conditions, outperforming existing benchmarks.

**(4) Extensive robustness studies**: We conduct ablation studies to assess the impact of key components, including signal morphology augmentation, comparisons with established contrastive learning approaches, model size, data efficiency, and the effect of skin tone.

## 2 RELATED WORK

Self-supervised learning has become a prominent paradigm for learning general representations from unlabeled datasets, with applications in physiological signal analysis including health, fitness, and brain signals (Tonekaboni et al., 2021; Zhang et al., 2022; Chen et al., 2021; Yèche et al., 2021; Spathis et al., 2021; Cheng et al., 2020; Kiyasseh et al., 2021; Sarkar & Etemad, 2020). Despite its popularity, there are no widely used models for PPG signals pre-trained through SSL. Recently, Abbaspourazad et al. (2023) demonstrated that embeddings derived from PPG signals can predict over 45 diverse downstream health-related tasks using proprietary Apple Watch data. Their approach uses an SSL framework based on patient-level positive pair contrastive learning. Similarly, (Yun et al., 2024) showed that embedding PPG signals can improve genetic discovery and risk prediction outcomes using the UK Biobank dataset. Other works (Weng et al., 2024; Ding et al., 2024; Zhou et al., 2024) explored PPG embeddings for various applications. However, these studies often used proprietary datasets, did not explore out-of-domain generalization, or did not release their models, highlighting the need for openly available, pre-trained PPG FMs (Table 17). For example, in contrast to (Abbaspourazad et al., 2023), our work exclusively uses public datasets for large-scale PPG training and introduces a novel SSL framework to incorporate PPG morphology. While Abbaspourazad et al. (2023) evaluate a single proprietary dataset, we validate on 10 diverse downstream datasets, showcasing greater generalizability and robustness across varied real-world scenarios.

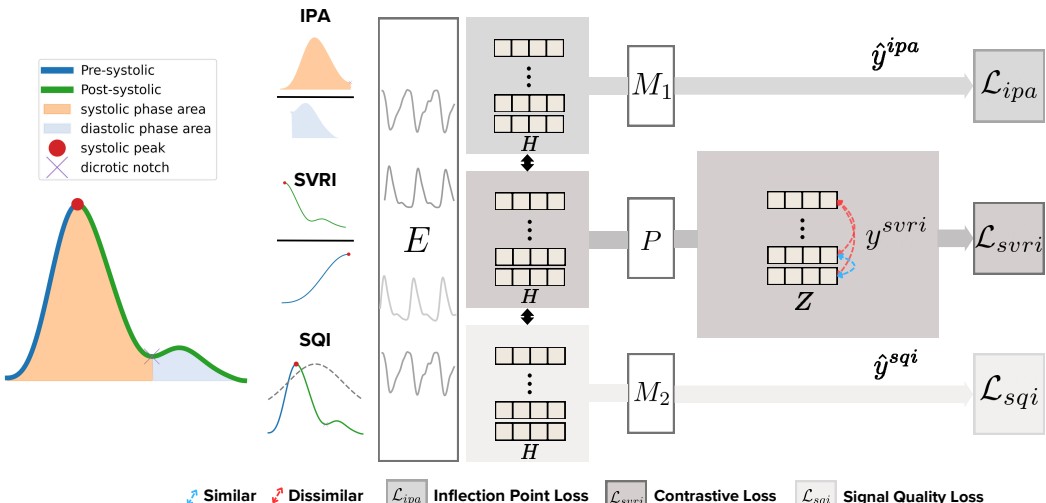

Figure 2: Overview of PAPAGEI-S. The process begins by computing three morphology metrics (IPA, SVRI, and SQI) for each PPG segment. The raw PPG signals are then processed through an encoder ($E$) to generate embeddings ($H$). These same embeddings feed into three specialized heads: a projection head ($P$) that contrasts PPG signals based on sVRI values, and two mixture-of-expert heads ($M_1$ and $M_2$) that refine the embeddings by predicting IPA and SQI values.

Generic time-series FMs, like Chronos (Ansari et al., 2024) and Moment (Goswami et al., 2024), lack physiological data representation. There is growing interest in modality-specific FMs tailored to physiological signals (Song et al., 2024; Lai et al., 2023) and human activity (Yuan et al., 2024a). Knowledge transfer from time-series FMs might benefit PPG tasks, but their performance is limited compared to PPG-specific FMs. Adapting other domain-specific models, like ECG (McKeen et al., 2024; Song et al., 2024) or EEG (Yuan et al., 2024b), is challenging due to distinct signal characteristics. We specifically design FMs for PPG signals, contributing to the growing movement toward foundation models tailored to individual modalities. See Appendix §H for an extended discussion.

## 3 METHODS

Given a dataset $\mathcal{D} = \{\mathbf{p}^1, \mathbf{p}^2, \cdots, \mathbf{p}^S\}$ representing diverse PPG signals from $S$ participants, a PPG signal $\mathbf{p}^s \in \mathbb{R}^n$ is defined as a time-series that captures variations in light intensity caused by arterial blood flow. To model granular changes in PPG signal of each subject $s$, we segment $\mathbf{p}^s$ without overlap to obtain $X^s = \{\mathbf{x}_1^s, \mathbf{x}_2^s, \cdots \mathbf{x}_N^s\}$. Here, the number of segments $N$ depends on the sampling frequency ($f$) and the desired length of time window. To train our foundation models, PAPAGEI-P employs a *patient contrastive* SSL approach that maximizes agreement between signals from the same subject. Importantly, we propose PAPAGEI-S, a *morphology-aware* self-supervised approach that maximizes agreement between PPG segments with similar morphology.

### 3.1 PARTICIPANT-AWARE OBJECTIVE: PAPAGEI-P

In PAPAGEI-P, we train an SSL model to maximize agreement between the embeddings of PPG signals from the same subject. While previous studies have demonstrated the effectiveness of this strategy for physiological signals (Kiyasseh et al., 2021; Abbaspourazad et al., 2023), our work represents the first attempt to train and evaluate a foundation model using publicly available PPG datasets.

**Training.** We define a *positive pair* as any two distinct segments of PPG signals from the same subject, denoted as $\{(\mathbf{x}_i^s, \mathbf{x}_j^s) | i \neq j\}$. Next, we apply a series of time-series augmentations such as random cropping, adding Gaussian noise, time flipping, negation, and magnitude scaling (Tang et al., 2020), each applied with a predefined probability during training. Each augmentation includes hyper-parameters that control the intensity of the data transformation. During training, the augmented version of a randomly sampled positive pair $(\mathbf{x}_i^s, \mathbf{x}_j^s)$ is passed through the encoder $E$, and subsequently projection $P$, to obtain an *embeddings* pair denoted by $(\mathbf{z}_i^s, \mathbf{z}_j^s)$. Given a batch of embeddings from $N$ positive pairs of the form $(\mathbf{z}_i, \mathbf{z}_j)$, the model optimizes the normalized temperature-scaled cross entropy (NT-Xent) loss (Sohn, 2016; Oord et al., 2018; Chen et al., 2020)

given by: $\mathcal{L}_p = \frac{1}{2}(\ell_p(i,j) + \ell_p(j,i))$, where $\ell_p(i,j) = -\frac{1}{N}\sum_{u=1}^{N}\log\frac{\exp(sim(\mathbf{z}_i^u,\mathbf{z}_j^u)/\tau)}{\sum_{v=1}^{2N}\mathbb{1}[v\neq u]\exp(sim(\mathbf{z}_i^u,\mathbf{z}_j^v)/\tau)}$ and $sim(\cdot,\cdot)$ is the cosine similarity. In contrast, vanilla SimCLR (Chen et al., 2020) would use positive pairs as augmented versions of randomly sampled PPG segments.

## 3.2 MORPHOLOGY-AWARE OBJECTIVE: PAPAGEI-S

In PAPAGEI-S, we leverage the PPG signal morphology to train a SSL model that maximizes agreement between similar physiological features of PPG signals across participants.

**PPG Morphology.** Total peripheral resistance (TPR)—the force exerted by the body's blood vessels on circulating blood—varies under certain medical conditions, such as hypertension and diabetes (Trammel & Sapra, 2020). Variations in TPR are reflected in PPG signals, presenting as distinct regions within the waveform. To capture these variations, we introduce a morphology augmentation module before training, which computes three key PPG metrics (Figure 2, left): (1) **stress-induced Vascular Response Index (sVRI)** (Lyu et al., 2015; Zhang et al., 2019): the ratio of mean PPG signal between post- to pre-systolic phases, (2) **Inflection Point Area ratio (IPA)** (Wang et al., 2009): the ratio of systolic to diastolic areas defined by the dicrotic notch, and (3) **Signal Quality Index (SQI)**: skewness of the signal as an indicator of quality (Elgendi, 2016). Prior studies have shown that incorporating the PPG signal quality during training yields positive results (Ding et al., 2024). We selected these metrics for their complementary nature: sVRI captures variations in amplitude, while IPA measures signal width. To address scenarios where computing IPA is challenging because of noisy signals or different morphology, we incorporate SQI. In particular, we empirically find that SQI is significantly larger ($p < 0.05$) in signals with a dicrotic notch (Appendix §D.5).

$$sVRI(\mathbf{x}) = \frac{sys\sum_{i=sys}^{n}x_i}{(n-sys)\sum_{i=1}^{sys}x_i}, \quad IPA(\mathbf{x}) = \frac{\int_0^{\hat{n}}\mathbf{x}\,dn}{\int_{\hat{n}}^{n}\mathbf{x}\,dn}, \text{ and } \quad SQI(\mathbf{x}) = \frac{1}{W}\sum_w\frac{m_3}{m_2^{3/2}}, \quad (1)$$

where $\mathbf{x} \in \mathbb{R}^N$ is the PPG segment, $sys$ is the systolic peak, $n$ is the length of time series, and $\hat{n}$ is the dicrotic notch. For $SQI$, we divide $\mathbf{x}$ into 5 second windows ($w$; total windows $W$) and compute the skewness $m_i = \frac{1}{5\times f}\sum_{j=1}^{5\times f}(x[j] - \mu_x[j])^i$, which gives the best signal quality discrimination.

**Training.** Before training, the morphology augmentation module takes an augmented input, by applying Gaussian noise and cropping to time series $\mathbf{x}$, and outputs $y = \{y^{svri}, y^{ipa}, y^{sqi}\} \in \mathbb{R}^3$ (Figure 2 middle). Next, we discretize $y^{svri}$ into a predefined set of $b = 8$ bins to denote positive pairs, where $y^{svri} \in \{1,\ldots,b\}$. We define positive pairs based on the sVRI labels as $\{(\mathbf{x}_i, \mathbf{x}_j)|y_i^{svri} = y_j^{svri}, i \neq j\}$. Note that positive pairs are not defined based on participants.

$$\ell_s(i,j) = -\log\frac{\exp(sim(\mathbf{z}_i,\mathbf{z}_j)/\tau)}{\sum_{k=1}^{2N}\mathbb{1}[k\neq i]\exp(sim(\mathbf{z}_i,\mathbf{z}_k)/\tau)} \quad (2)$$

$$\mathcal{L}_{svri} = \frac{1}{2N}\sum_{k=1}^{N}[\ell(2k-1,2k) + \ell(2k,2k-1)] \quad (3)$$

$$\mathcal{L}_{ipa} = \frac{1}{N}\sum_{i=1}^{N}\left|y_i^{ipa} - \hat{y}_i^{ipa}\right| \quad \mathcal{L}_{sqi} = \frac{1}{N}\sum_{i=1}^{N}\left|y_i^{sqi} - \hat{y}_i^{sqi}\right| \quad (4)$$

$$\mathcal{L}_s = \alpha\mathcal{L}_{svri} + (1-\alpha)(\mathcal{L}_{ipa} + \mathcal{L}_{sqi}), \text{ where } \alpha \in [0,1] \quad (5)$$

Given a batch of $N$ PPG signals and their morphology, we optimize three heads using the encoder ($E$) embeddings $H = \{\mathbf{h}_1, \mathbf{h}_2, \cdots, \mathbf{h}_N\}$. First, we extract the embeddings $Z = \{\mathbf{z}_1, \mathbf{z}_2, \cdots, \mathbf{z}_N\}$ from the projection ($P$), and compute the contrastive loss for sVRI (equation 3). Next, we use the embeddings $H$ to predict the IPA ($\hat{\mathbf{y}}^{ipa} \in \mathbb{R}^N$) and SQI ($\hat{\mathbf{y}}^{sqi} \in \mathbb{R}^N$) using the mixture of expert (MoE) heads $M_1$ and $M_2$. Each MoE head is composed of three fully connected neural networks (FCNNs), with the head's output calculated as a weighted sum of the FCNNs, using softmax to determine the weights. These heads are optimized using the mean absolute error (equation 4). The morphology indices encapsulate various PPG characteristics. Our rationale for utilizing MoE is that each expert can specialize in learning distinct properties that contribute to the overall index. Finally, the overall PAPAGEI-S training objective is given in equation 5.

## 4 EXPERIMENTS

### 4.1 PRE-TRAINING

**Datasets.** We pre-train PAPAGEI on three datasets: (1) VitalDB (Lee et al., 2022), which includes PPG signals collected during surgery from the patient's finger ($f$=500Hz), (2) the MIMIC-III waveform database matched subset (Johnson et al., 2016), where finger-tip PPG data is collected from an ICU monitor ($f$= 125Hz), and (3) the Multi-Ethnic Study of Atherosclerosis (MESA) sleep substudy (Zhang et al., 2018; Chen et al., 2015), which provides PPG data obtained through finger-tip polysomnography ($f$= 256Hz). In total, we have 13.5K participants with 20M segments (Table 1).

**Pre-processing.** To curate single-channel PPG signals across all datasets, we perform the following steps: (1) Apply a 4th-order Chebyshev bandpass filter with low and high pass cut-offs set at 0.5Hz and 12Hz, respectively (Lapitan et al., 2024; Liang et al., 2018c); (2) Segment the signal into 10-second windows ((Orphanidou, 2018; Koteska et al., 2022) use 10s windows whereas larger studies use 30s (Ding et al., 2024) and 60s Abbaspourazad et al. (2023)); (3) Detect flatline segments and remove any seg-

Table 1: PAPAGEI's pre-training datasets.

| Dataset | #Participants | #Segments | Hours |
|---|---|---|---|
| VitalDB | 5,866 | 6,248,100 | 17,355 |
| MIMIC-III | 5,596 | 7,196,401 | 19,990 |
| MESA | 2,055 | 7,306,705 | 20,296 |
| Total | 13,517 | 20,751,206 | 57,641 |

ment where more than 25% of the data is flat (BioBSS Documentation, 2023); (4) Normalize the segments using Z-score (Temko, 2017; Zhou et al., 2017); and (5) Resample the segments to 125Hz (the lowest sampling rate of our pre-training datasets, MIMIC-III).

**Implementation.** We adopt a ResNet-style CNN encoder, following (Ding et al., 2024). Abbaspourazad et al. (2023) also utilize an EfficientNet-style CNN. Our model has 18 convolutional blocks, starting with a filter size of 32, which doubles every 4 blocks. The projection layer is a single FC layer, generating a 512-d embedding. In the PAPAGEI-S variant, the expert block ($M_1$ & $M_2$) uses three parallel FCNNs, each with two FC layers, resulting in a 128-d embedding. For augmentations, PAPAGEI-P uses cropping (0.50), negation (0.20), flipping (0.20), and scaling (0.40). PA-PAGEI-S uses cropping (0.25) and Gaussian noise (0.25). PA-PAGEI-S avoids augmentations that alter PPG's morphology.

Table 2: PAPAGEI's evaluation datasets. Gray lines are unseen during training (out-of-domain). For those used for pre-training, we keep a held-out test-sets and use labels. Task Types are: B=binary, R=regression, M-#classes= muticlass classification.

| #ID | Dataset | Task (Task Type) | #Subj.(#Samp.) |
|---|---|---|---|
| T1 | VitalDB (Lee et al., 2022) | ICU admission (B) | 5866 |
| T2 | | Operation Type (M-9) | 5866 |
| T3 | MIMIC-III (Moody et al., 2020) | Mortality (B) | 5596 |
| T4 | MESA (Zhang et al., 2018) | Smoker (B) | 2055 |
| T5 | | AHI > 3% Oxygen Desat. (R) | 2055 |
| T6 | | AHI > 4% Oxygen Desat. (R) | 2055 |
| T7 | nuMom2B (Facco et al., 2015) | Pregnancy stage (B) | 3163 (5337) |
| T8 | | Gestation Age (R) | 3163 (5337) |
| T9 | VV (Skin Tone) (Toye, 2023) | Systolic BP* (R) | 231 |
| T10 | | Diastolic BP* (R) | 231 |
| T11 | PPG-BP (Liang et al., 2018a) | Systolic BP (R) | 219 |
| T12 | | Diastolic BP (R) | 219 |
| T13 | | Average Heart Rate (R) | 219 |
| T14 | | Hypertension (B) | 219 |
| T15 | SDB (Garde et al., 2014) | Sleep Disordered Breathing (B) | 146 |
| T16 | ECSMP (Gao et al., 2021) | Mood Disturbance (B) | 89 |
| T17 | WESAD (Schmidt et al., 2018) | Valence (B) | 15 (4497) |
| T18 | | Arousal (B) | 15 (4497) |
| T19 | PPG-DaLiA (Reiss et al., 2019) | Heart Rate (R) | 15 (64697) |
| T20 | | Activity (M-9) | 15 (64697) |

We set $\alpha = 0.6$ and train on eight V100 GPUs for 15,000 steps (lr= $10^{-4}$), with PAPAGEI-P and PAPAGEI-S having 5M and 5.7M parameters, respectively, while previous works use model sizes of 3.3M (Abbaspourazad et al., 2023) (we study scaling in Section 5.2).

### 4.2 DOWNSTREAM TASKS

To evaluate the effectiveness of PAPAGEI, we benchmark it against a diverse set of datasets, tasks, and baselines, chosen for their large size and clinical relevance (where applicable)[2]. A description of the tasks with their corresponding #ID is provided in Table 2, with further details in Appendix §B. As a motivation, identifying patient risk factors is crucial for hospitals to allocate resources effectively. To address this, we evaluate several indicators, including ICU admission (T1), type of operation (T2), mortality (T3), and smoking status (T4). For sleep apnea diagnosis, the American

---

[2]https://peterhcharlton.github.io/post/ppg_datasets/

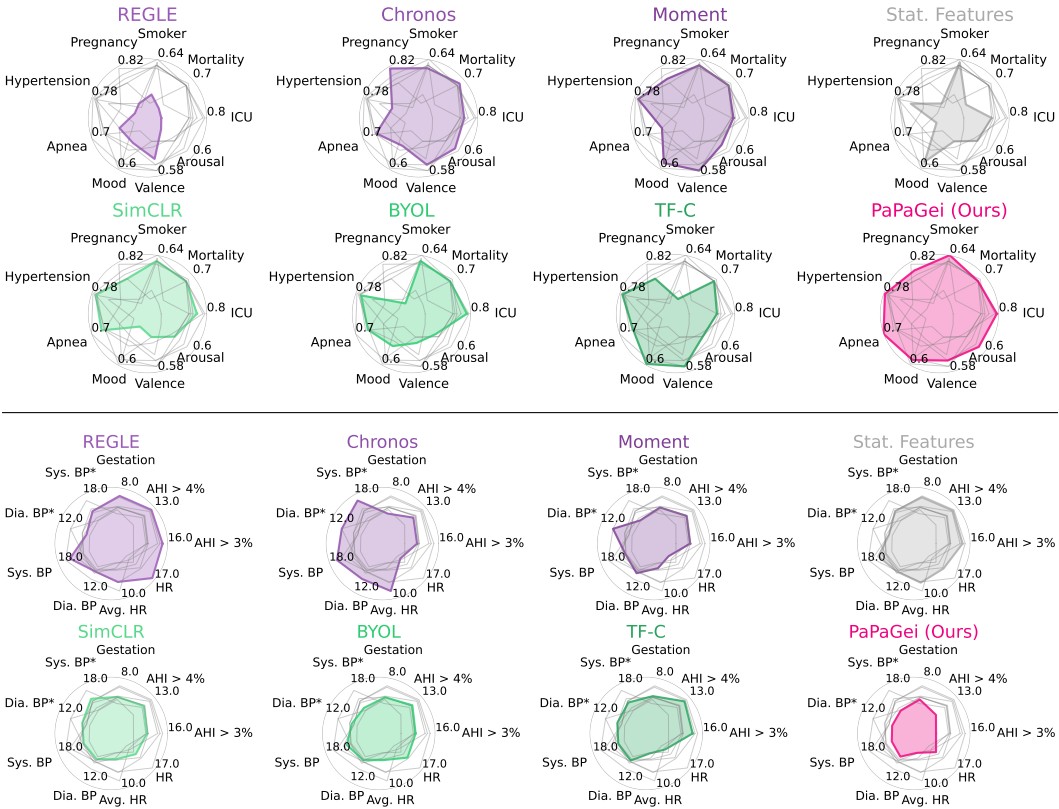

Figure 3: Radar charts of downstream tasks. (Top) *Classification* performance in **AUROC (larger area is better)**. (Bottom) *Regression* performance in **MAE (smaller area is better)**. Pre-trained models in purple: REGLE, Chronos, & Moment. Statistical feature baseline in gray. SSL methods in green: SimCLR, BYOL, & TF-C. PAPAGEI (ours), in pink. Details are in Tables 3 & 4.

Academy of Sleep Medicine recommends using the Apnea/Hypopnea Index (AHI) with at least 3% or 4% oxygen desaturation as a key metric (Ruehland et al., 2009). Thus, we predict AHI at 3% and 4% desaturation thresholds (T5 & T6) and classify sleep-disordered breathing (T15). For pregnancy outcomes, changes in gestational age and pregnancy stage are linked to risks like hypertensive disorders and small-for-gestational-age delivery (Bouariu et al., 2022; Wu et al., 2020; Crump et al., 2023), enabling us to classify pregnancy stage (T7) and predict gestational age (T8). In cardiovascular health, we estimate systolic (T9 & T11) and diastolic (T10 & T12) blood pressure (BP) using two datasets. While PPG-BP (T11 & T12) provides high-frequency, short PPG signals, the VV dataset helps explore skin tone's influence on BP estimation. We also assess hypertension classification (T14), average seated heart rate (T13), and continuous heart rate during activities (T19), along with activity classification (T20). In the emotion domain, we classify PPG signals into mood disturbance levels (T16), valence (T17), and arousal (T18).

## 4.3 BASELINES

We benchmark PAPAGEI's performance against competitive baselines. As open-source foundation models designed for physiological signals, PAPAGEI is compared to recent time-series FMs: **Chronos** (Ansari et al., 2024) and **MOMENT** (Goswami et al., 2024). To evaluate the merits of our SSL framework, we also compare PAPAGEI with common SSL methods (trained from scratch) such as **SimCLR** (Chen et al., 2020), **BYOL** (Grill et al., 2020), and **TF-C** (Zhang et al., 2022). In addition, to assess model generalizability on PPG signals, we compare against **REGLE**, a model pre-trained on UK Biobank's PPG signals (Yun et al., 2024). As a simple baseline, we employ a random forest trained on **statistical features** extracted from the PPG signal, including mean, median, maximum, minimum, and the 25th, 50th, and 75th percentiles ("Stat. Features"). This task-specific approach serves as a benchmark for comparison with more advanced techniques.

Table 3: **Downstream comparison against pre-trained models.** Feature extraction parameters are indicated next to each name. 95% CIs are reported in square brackets and the best value is **bolded**.

| Classification - AUROC (↑) | REGLE (0.07M) (Yun et al., 2024) | Chronos (200M) (Ansari et al., 2024) | Moment (385M) (Goswami et al., 2024) | PAPAGEI-P (5M) | PAPAGEI-S (5M) |
|---|---|---|---|---|---|
| ICU Admission | 0.57 [0.52-0.62] | 0.73 [0.68-0.80] | 0.72 [0.70-0.80] | 0.73 [0.67-0.78] | **0.79** [0.75-0.82] |
| Mortality | 0.55 [0.52-0.59] | **0.68** [0.65-0.71] | 0.67 [0.63-0.71] | 0.67 [0.63-0.71] | 0.67 [0.63-0.70] |
| Smoker | 0.54 [0.47-0.59] | 0.62 [0.57-0.67] | 0.62 [0.56-0.67] | **0.64** [0.58-0.69] | 0.61 [0.56-0.66] |
| Pregnancy stage | 0.64 [0.57-0.63] | **0.81** [0.79-0.82] | 0.76 [0.74-0.78] | 0.74 [0.72-0.76] | 0.78 [0.75-0.80] |
| Hypertension | 0.47 [0.34-0.58] | 0.57 [0.43-0.71] | 0.75 [0.64-0.85] | 0.74 [0.55-0.90] | **0.77** [0.68-0.87] |
| Sleep Disordered Breathing | 0.45 [0.30-0.61] | 0.58 [0.35-0.82] | 0.45 [0.23-0.66] | 0.54 [0.23-0.66] | **0.70** [0.57-0.84] |
| Mood Disturbance | 0.41 [0.16-0.66] | 0.43 [0.21-0.68] | 0.55 [0.33-0.78] | 0.53 [0.27-0.78] | **0.56** [0.33-0.77] |
| Valence | 0.55 [0.52-0.57] | 0.56 [0.53-0.59] | **0.57** [0.54-0.59] | 0.53 [0.51-0.56] | 0.56 [0.54-0.59] |
| Arousal | 0.51 [0.52-0.58] | 0.57 [0.54-0.60] | 0.56 [0.53-0.58] | **0.58** [0.55-0.61] | 0.55 [0.52-0.57] |
| Average | 0.52 ± 0.06 | 0.62 ± 0.10 | 0.63 ± 0.09 | 0.63 ± 0.08 | **0.67 ± 0.09** |
| **Regression - MAE (↓)** | | | | | |
| Apnea/Hypopnea Index > 3% | 15.54 [14.20-16.69] | 14.06 [13.05-15.16] | 14.23 [13.04-15.42] | 13.85 [12.43-15.49] | **12.97** [11.87-14.05] |
| Apnea/Hypopnea Index > 4% | 12.64 [11.47-13.78] | 11.57 [10.51-12.72] | 11.80 [10.79-12.93] | 11.24 [9.71-12.87] | **10.56** [9.59-11.62] |
| Gestation Age | 7.28 [7.16-7.39] | **5.69** [5.54-5.85] | 6.24 [6.10-6.37] | 6.40 [6.21-6.59] | 6.05 [5.91-6.17] |
| Systolic BP (VV) | 15.88 [13.67-18.36] | 17.24 [14.57-20.13] | 14.71 [12.38-17.29] | 19.11 [16.26-22.23] | **14.65** [12.50-16.78] |
| Diastolic BP (VV) | 8.65 [7.16-10.27] | 10.53 [8.91-12.19] | 10.53 [8.91-12.19] | 10.87 [9.10-12.98] | **8.29** [6.61-10.22] |
| Systolic BP (PPG-BP) | 16.32 [13.87-19.13] | 16.91 [13.31-19.34] | 14.50 [11.98-17.31] | **13.60** [10.65-16.51] | 14.39 [12.53-16.45] |
| Diastolic BP (PPG-BP) | 9.30 [7.94-10.87] | 10.26 [8.13-12.57] | 9.53 [8.28-10.96] | 8.88 [7.33-10.76] | **8.71** [7.18-10.01] |
| Average HR | 6.88 [5.81-8.12] | 8.51 [7.05-10.07] | 4.41 [3.48-5.48] | **3.47** [2.74-4.32] | 4.00 [3.34-4.67] |
| HR | 16.35 [16.20-16.50] | 9.65 [9.50-9.79] | **8.82** [8.68-8.96] | 10.92 [10.80-11.04] | 11.53 [11.40-11.66] |
| Average MAE (sMAPE) | 12.09 ± 3.83 (15.23%) | 11.60 ± 3.60 (14.20%) | 10.43 ± 3.46 (13.82%) | 10.92 ± 4.25 (14.09%) | **10.12 ± 3.47** (13.34%) |

## 4.4 LINEAR EVALUATION

We initially split the in-domain and out-of-domain datasets into training, validation, and test sets using 80/10/10 and 60/20/20 ratios at the participant-level, respectively. Hyperparameter optimization is performed on the training set using nested cross-validation, thus the validation and test sets are merged for evaluation. Models are evaluated by extracting feature representations from resampled data (125Hz) and applying linear probing for each task. For binary classification, we use logistic regression, measuring performance with AUROC. Regression tasks are evaluated using ridge regression and mean absolute error (MAE), with aggregated results reported via symmetric mean absolute percentage error (sMAPE). Multi-class classification tasks are trained using a random forest model, with accuracy as the evaluation metric. To ensure robustness, we compute 95% confidence intervals through bootstrapping (500 sampling runs with replacement). Additional details in Appendix §A.

## 5 RESULTS

### 5.1 OVERALL PERFORMANCE

In general, from Figure 3, we observe that PAPAGEI is more accurate across many tasks indicated by the larger AUROC area and smaller MAE area. Table 3 presents a more detailed comparison between PAPAGEI and other pre-trained models.

For classification tasks, PaPaGei-S achieves the highest average AUROC of 0.67, outperforming other models across several tasks, particularly in ICU Admission (0.79), Hypertension (0.77), and Sleep Disordered Breathing (0.70). In regression tasks, PaPaGei-S again demonstrates strong performance, achieving the lowest average MAE (10.12), particularly in tasks related to Apnea/Hypopnea Index and BP measurements. REGLE, a small model trained on a large PPG dataset, generally underperforms compared to other models, suggesting its compact size may limit learning complex patterns.

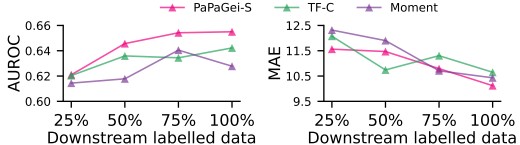

Figure 4: Downstream data-efficiency analysis. Results are averaged over all binary classification (left) and regression tasks (right). PAPAGEI-S performs better with increased label availability.

Chronos obtains good performance in predicting mortality, pregnancy stage, and smoking, likely due to their slower rate of change and reduced reliance on granular PPG-specific features. General-purpose models suffice for these high-level outcomes. However, tasks requiring finer PPG-specific granularity, such as heart rate prediction, blood pressure estimation, or sleep apnea, benefit from PAPAGEI's specialized feature extraction. Notably, PAPAGEI-S consistently outperforms PAPAGEI-P, highlighting the advantages of signal morphology objectives in enhancing predictive accuracy.

Table 4: **Downstream comparison against baseline and SSL methods.** Feature extraction parameters are indicated next to each name. 95% CIs are reported in square brackets and the best value is **bolded**. Implementation details are in Appendix §A.

| | Stat. Features | SimCLR (5M) (Chen et al., 2020) | BYOL (5M) (Grill et al., 2020) | TF-C (10M) (Zhang et al., 2022) | PAPAGEI-P (5M) | PAPAGEI-S (5M) |
|---|---|---|---|---|---|---|
| **Classification** - AUROC (↑) | | | | | | |
| ICU Admission | 0.71 [0.65-0.78] | 0.75 [0.72-0.79] | 0.78 [0.73-0.81] | 0.71 [0.67-0.75] | 0.73 [0.67-0.78] | **0.79** [0.75-0.82] |
| Mortality | 0.57 [0.54-0.61] | 0.67 [0.63-0.70] | 0.67 [0.64-0.71] | 0.67 [0.63-0.70] | 0.67 [0.63-0.71] | 0.67 [0.63-0.70] |
| Smoker | 0.63 [0.58-0.67] | 0.62 [0.57-0.68] | 0.62 [0.57-0.68] | 0.61 [0.56-0.67] | **0.64** [0.58-0.69] | 0.61 [0.56-0.66] |
| Pregnancy stage | 0.64 [0.62-0.67] | 0.74 [0.72-0.75] | 0.62 [0.57-0.68] | 0.74 [0.72-0.76] | 0.74 [0.72-0.76] | **0.78** [0.75-0.80] |
| Hypertension | 0.66 [0.47-0.83] | 0.75 [0.64-0.86] | 0.74 [0.64-0.84] | 0.76 [0.63-0.86] | 0.74 [0.55-0.90] | **0.77** [0.68-0.87] |
| SDB | 0.32 [0.14-0.55] | 0.61 [0.46-0.76] | 0.59 [0.42-0.74] | 0.58 [0.44-0.73] | 0.54 [0.23-0.66] | **0.70** [0.57-0.84] |
| Mood Disturbance | 0.54 [0.31-0.77] | 0.32 [0.12-0.55] | 0.46 [0.21-0.71] | **0.59** [0.33-0.84] | 0.53 [0.27-0.78] | 0.56 [0.33-0.77] |
| Valence | 0.52 [0.49-0.55] | 0.52 [0.49-0.55] | 0.53 [0.50-0.56] | **0.57** [0.54-0.59] | 0.53 [0.51-0.56] | 0.56 [0.54-0.59] |
| Arousal | 0.55 [0.53-0.58] | 0.55 [0.52-0.58] | 0.54 [0.30-0.78] | 0.55 [0.52-0.58] | **0.58** [0.55-0.61] | 0.55 [0.52-0.57] |
| Average | 0.57 ± 0.11 | 0.61 ± 0.13 | 0.62 ± 0.10 | 0.64 ± 0.07 | 0.63 ± 0.08 | **0.67 ± 0.09** |
| **Regression** - MAE (↓) | | | | | | |
| Apnea/Hypopnea Index > 3% | 15.31 [13.63-17.14] | 14.17 [13.04-15.38] | 14.26 [13.10-15.57] | 15.10 [13.84-16.40] | 13.85 [12.43-15.49] | **12.97** [11.87-14.05] |
| Apnea/Hypopnea Index > 4% | 12.52 [10.92-14.14] | 11.76 [10.65-12.89] | 11.88 [10.71-13.05] | 12.41 [11.33-13.49] | 11.24 [9.71-12.87] | **10.56** [9.59-11.62] |
| Gestation Age | 7.15 [6.99-7.34] | 6.28 [6.21-6.49] | 6.24 [6.09-6.38] | 6.35 [6.21-6.49] | 6.40 [6.21-6.59] | **6.05** [5.91-6.17] |
| Systolic BP (VV) | 15.76 [13.67-18.36] | 16.18 [13.73-18.85] | 15.01 [12.32-17.80] | 15.70 [13.23-18.13] | 19.11 [16.26-22.23] | **14.65** [12.50-16.78] |
| Diastolic BP (VV) | 9.75 [7.16-11.27] | 9.15 [7.65-10.65] | 8.91 [7.48-10.43] | 9.15 [7.65-10.65] | 10.87 [9.10-12.98] | **8.29** [6.61-10.22] |
| Systolic BP (PPG-BP) | 15.50 [11.68-20.25] | 14.38 [11.80-16.88] | 14.99 [13.03-17.38] | 14.45 [12.20-17.00] | **13.60** [10.65-16.51] | 14.39 [12.53-16.45] |
| Diastolic BP (PPG-BP) | 9.35 [7.44-11.66] | 9.01 [7.90-10.60] | 9.16 [8.00-10.50] | 9.20 [7.90-10.60] | 8.88 [7.33-10.76] | **8.71** [7.18-10.01] |
| Average HR | 7.01 [5.48-8.89] | 4.65 [3.99-5.39] | 4.78 [3.88-5.93] | 3.58 [2.90-4.21] | **3.47** [2.74-4.32] | 4.00 [3.34-4.67] |
| HR | 13.07 [12.90-13.23] | 11.59 [11.46-11.72] | 12.80 [12.66-12.94] | **9.99** [9.86-10.12] | 10.92 [10.80-11.04] | 11.53 [11.40-11.66] |
| Average MAE (sMAPE) | 11.60 ± 3.41 (15.12%) | 10.79 ± 3.63 (13.91%) | 10.89 ± 3.58 (14.05%) | 10.65 ± 3.88 (14.07%) | 10.92 ± 4.25 (14.09%) | **10.12 ± 3.47** (13.34%) |

Table 4 presents a comparison against three SSL methods and a baseline model trained on statistical features. In classification tasks, PaPaGei-S again shows the highest average AUROC, outperforming all others. SimCLR, BYOL, and TF-C generally outperform the statistical feature baseline but fall short of PaPaGei-S's performance. TF-C shows competitive results in some tasks, achieving the highest AUROC for Mood Disturbance and Valence. For regression tasks, PaPaGei-S again achieves the lowest average MAE. SimCLR, BYOL, and TF-C show mixed results, as each excels in different tasks. SimCLR comes second in estimating Avg HR, while BYOL does so in Systolic BP (VV). The statistical feature baseline generally underperforms compared to the advanced methods across most tasks. PaPaGei-P, while not consistently outperforming PaPaGei-S, shows strong results that are often competitive with or better than other contrastive learning methods. Overall, both PaPaGei variants offer robust performance across a wide range of tasks.

## 5.2 ABLATION STUDIES

**Pre-training data ablation.** We evaluate PAPAGEI-S using different pre-training data combinations. As shown in Figure 5, performance on downstream tasks improves with more upstream data, with the best results achieved when using all three datasets. Notably, MESA outperforms the others despite having the fewest participants but the highest number of segments. This supports findings from language models (Dubey et al., 2024) and wearable sensing research (Narayanswamy et al., 2024), indicating that the volume of segments or hours contributes more to performance than the number of users.

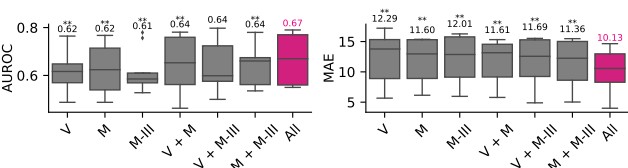

Figure 5: Ablation on pre-training data. Average performance across tasks for models trained on: V (VitalDB), M (MESA), and M-III (MIMIC-III). The mean value is displayed above the plots. The Wilcoxon signed rank test is applied to evaluate significance between the All dataset and the rest (∗∗ : $p < 0.05$ and ∗ : $0.05 \leq p < 0.10$).

**PAPAGEI-S component ablation.** We assess the impact of PAPAGEI-S components. Figure 6 shows that the full model (0.67, 10.12) consistently outperforms individual components in both mean and median metrics. On average, sVRI (0.64, 10.35) outperforms the combinations of sVRI + SQI (0.62, 10.80) and sVRI + IPA (0.64, 10.73). Our results indicate that combining SQI and IPA yields greater benefits compared to their individual contributions.

**Downstream data-efficiency analysis.** For limited-data scenarios, we assess the performance of downstream linear probing across varying levels of labeled data availability. We compare to the second best-performing baselines from Tables 3 & 4, namely TF-C and Moment. As shown in Figure 4, the classification performance of PAPAGEI-S steadily improves as more labeled data becomes

available. While TF-C and Moment also show performance gains between 25% and 100% labeled data, their improvements are less consistent and smaller than PAPAGEI-S. In regression tasks, PA-PAGEI-S achieves the lowest MAE at both 25% and 100% data availability, consistently reducing errors. At the middle breakpoints, the results are mixed with TF-C and Moment being competitive.

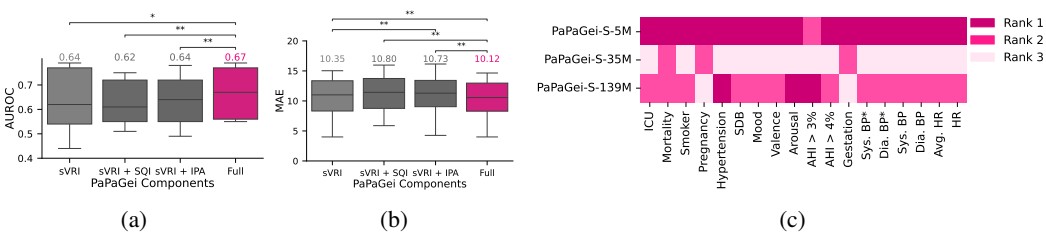

Figure 6: PAPAGEI-S component ablation study (a, b) and scaling analysis (c). (Left) The boxplot shows the performance of PAPAGEI-S components across all tasks. The Wilcoxon signed rank test is applied to evaluate pair-wise significance ($** : p < 0.05$ and $* : 0.05 \leq p < 0.10$). (Right) Heatmap ranks of PAPAGEI-S models with 5M, 35M, and 139M parameters (rank 1 denotes the best performance). Detailed results in Table 13.

**Model size and scaling analysis.** We investigated the impact of model size on performance by training PAPAGEI-S-35M and PAPAGEI-S-139M with 35M and 139M parameters, respectively.

Both models share the same number of layers, but the 35M model uses a 32-filter size while the 139M model uses 64. As shown in 6c, the smallest model (5M parameters) consistently outperformed larger models on all but one task. This suggests the 5M model is better suited for our pre-training datasets, aligning with prior findings on the proportionality between data and model size (Narayanswamy et al., 2024). While the 139M model surpassed the 35M, it still lagged behind the 5M, indicating that wider models may improve performance in classification tasks,

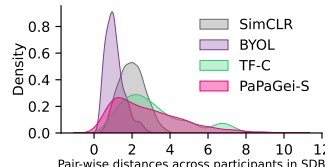

Figure 7: Pair-wise inter-participant embedding distances for SDB.

likely due to the contrastive learning objective. Nevertheless, our scaling analysis shows a non-monotonic trend, indicating other factors strongly influence performance.

**Effect of Demographics.** We evaluate the effect of demographics (age, sex) and PPG-specific features (sVRI, IPA, SQI) in Appendix §E. In demographics prediction (Table 16), PAPAGEI-S achieves 7.78 MAE in age regression, 0.85 accuracy in age classification, and 0.79 accuracy in sex classification. While our results trail larger closed studies (Abbaspourazad et al., 2023) by 2.18, 0.05, and 0.13 for segment-level SSL, and by 5.59, 0.12, and 0.25 for patient-level SSL, they mark an advancement in open-source efforts. These findings indicate that patient-level positive pair selection in SSL better captures demographic-related features for downstream prediction. Moreover, the reduced performance of PAPAGEI-S can be attributed to evaluations conducted on diverse device setups, as opposed to a single device configuration. Our ablation study (Table 15) shows that PAPAGEI-S outperforms the demo + PPG in 14 out of 18 tasks, particularly in tasks with real-time dependence such as heart rate estimation. Importantly, including demographics in addition to PAPAGEI-S creates a stronger model. These findings emphasize that **demographic features complement rather than compete with PAPAGEI-S**, showcasing the potential of integrating PAPAGEI's advanced feature extraction capabilities with demographic context to improve task outcomes.

## 5.3 CASE STUDIES

**Inter-participant embeddings.** Figure 7 shows the distribution of pair-wise embedding distances across participants in the SDB dataset (Kiyasseh et al., 2021). SimCLR and BYOL exhibit sharper peaks at lower distances, indicating that participants are more closely clustered within the embedding space. This could be interpreted as a mild form of mode collapse, where the model does not fully capture the individual differences between participants. TF-C demonstrates a more balanced distribution, with both large and small peaks, suggesting it captures both similarities and some variation between participants. In contrast, PAPAGEI-S provides the widest dispersion of embeddings,

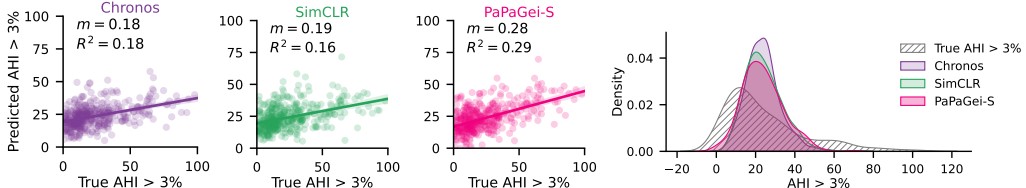

Figure 8: Regression plots and prediction distribution of different models compared to ground truth for AHI $> 3\%$. $R^2$ is the coefficient of determination and $m$ is the correlation slope.

highlighting its ability to capture a broader range of features that may be valuable for distinguishing between participants' medical conditions.

**Regression predictions.** From Figure 8, compared to the pre-trained and SSL baseline, PAPAGEI-S demonstrates steeper slopes ($m$) and higher $R^2$ values, reflecting a stronger alignment between predictions and true values. Additionally, the prediction distribution for AHI indicates that SimCLR and Chronos tend to regress more toward the mean, while PAPAGEI-S achieves a wider distribution base, highlighting its capacity to capture left tail better. Additional plots are shown in Appendix §F.

**Skin tone analysis.** We examine BP estimation performance across skin tones because it is crucial for practical use (Bent et al., 2020). As shown in Figure 9 (More details in Figure 26), PAPAGEI-S achieves the best BP estimation across light tones. Across dark tones, we notice that BYOL and REGLE obtain the lowest MAE for Systolic BP and Diastolic BP. However, identifying a single model that performs best across all skin tones remains challenging. While PAPAGEI-S obtains the best overall performance, additional work is necessary to improve robustness on darker skin tones.

## 6 DISCUSSION & CONCLUSION

Our results show that PAPAGEI outperforms baselines in at least 14 tasks, with classification and regression improvements of 4.7%-6.3% and 2.9%-4.9%, respectively. PAPAGEI-S excelled in cardiovascular tasks like BP, Hypertension, and HR, which can be attributed to the sVRI and IPA objectives, and PAPAGEI-P outperformed baselines like Moment, excelling in tasks such as Smoking and Arousal. Ablation studies confirmed that the model with all three SSL objectives performs best, with sVRI highlighted as a key component and IPA and SQI providing positive knowledge transfer in multi-task setups. To assess performance under class imbalance, we examined the F1-score. PaPaGei achieves the highest F1 in 6 out of 9 classification tasks, demonstrating its effectiveness in handling data imbalance. For regression, PaPaGei-S achieves the highest $R^2$ in 7 tasks (Appendix §D), reflecting better alignment with the true distribution. These results highlight the robustness and versatility of PaPaGei-S across classification and regression tasks. PAPAGEI is both data- and size-efficient (5M),

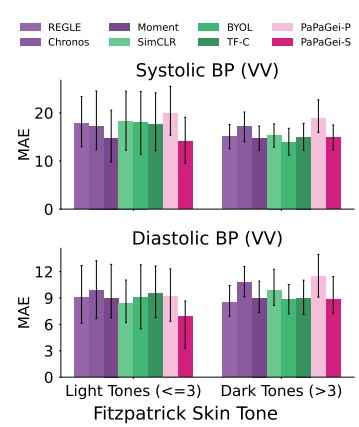

Figure 9: Skin tone analysis for Blood Pressure estimation (VV dataset)

making it ideal for medical applications where large models (200M+) are impractical due to on-device limitations or data privacy concerns with cloud model inference. While combining PAPAGEI-P and PAPAGEI-S objectives into one model might seem intuitive, it is impractical because it would constrain positive pairs on both sVRI and the number of participants, resulting in too many unique labels with limited samples per label. Our case studies also showed that PAPAGEI-S captured personal medical information due to well-dispersed embeddings, compared to baselines. Future work should focus on diversifying training data, investigating sampling rate effects, and exploring multimodal approaches or alternative architectures. Additionally, as extracting PPG features for different morphologies is non-trivial, future work benefit from systematic evaluation of PPG features and modeling. In conclusion, PAPAGEI represents a significant advancement in foundation models for analyzing PPG signals in resource-constrained medical environments, with its open-source nature encouraging further research and development in healthcare applications.

REPRODUCIBILITY STATEMENT

Models, data, and code are publicly available for reproducibility and future research. We exclusively utilize publicly accessible datasets, which can be requested or downloaded from the respective study group websites, allowing others to easily obtain the data for their own analyses. In §4 and Appendix §B, we provide comprehensive descriptions of the datasets, ground-truth annotations, and data pre-processing methods used in our experiments, ensuring transparency in our data handling procedures. The code to run our model is published with user-friendly examples. We have provided a detailed overview of the model architecture and its hyperparameters in §3, §4.1, and Appendix §A. Thus, our work is designed to be reproducible, enabling future research to build upon our findings.

ETHICS STATEMENT

Our research on PAPAGEI, utilizing publicly available PPG datasets, adheres to data privacy regulations and promotes transparency through open-source releases. We acknowledge potential biases in the training data and have evaluated performance across diverse datasets, particularly regarding skin tone variations. While PaPaGei offers significant potential for improving non-invasive health monitoring, we recognize the need to address potential misuse (Perez-Pozuelo et al., 2021). Examples of misuse could include unauthorized health monitoring, discriminatory practices in insurance or employment, unfair credit scoring, or exploiting personal health data for targeted marketing. We strongly advocate responsible use solely for beneficial healthcare applications. Our study followed established research ethics guidelines, and we declare no conflicts of interest. We encourage ongoing interdisciplinary dialogue to address potential risks and ensure responsible development and deployment of such technologies, recognizing the broader societal impacts of AI in healthcare. We remain committed to ethical AI advancement and welcome further discussion on the critical issues, including the development of governance frameworks to prevent misuse and protect data privacy.

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

## APPENDIX

## A    TRAINING AND INFERENCE DETAILS

**Architecture & Pre-training.** The architecture of our ResNet 18-block encoder is described in Tables 5, 6, and 7. Each 1D convolution layer is configured with a kernel size of 3 and a stride of 2, while the max-pooling layer utilizes a kernel size of 3 with a stride of 1. We start with a filter size of 32, which doubles every 4 blocks to capture progressively more complex features. Dropout is applied with a probability of 0.5 to prevent overfitting. This backbone architecture is used across different methods in our experiments to ensure consistency and make for a fair comparison during evaluation. Additionally, our SSL baselines use the same batch size, learning rate, input sampling frequency, and training steps as PAPAGEI. We use the same augmentation types and intensity for BYOL (Grill et al., 2020), SimCLR Chen et al. (2020), and PAPAGEI-P. Furthermore, we investigated 0.07 and 0.5 temperatures as MoCo (He et al., 2020) and SimCLR (Chen et al., 2020), respectively. In contrast to a smaller embedding size of 256 adopted by (Abbaspourazad et al., 2023), we project the learned representations to a 512-dimensional embedding after the convolutional block (we investigated larger embedding sizes of 768 and 1024 and found no significant performance changes). This embedding is then passed through two Mixture of Experts (MoE) blocks, each containing three experts. Each expert block consists of two sequential linear layers, with sizes 256 and 1, which are used for IPA and SQI prediction tasks. It is noteworthy that both BYOL and TF-C require multiple encoders and different projection heads, resulting in variations in model sizes. For these methods, we use existing implementations available online[3][4], but apply our encoder as the backbone to ensure consistency. Our models are pre-trained for 15,000 steps using the Adam optimizer, with a learning rate of $10^{-4}$. We use a batch size of 128 for training since after various trials we did not observe significant differences in performance with batch sizes of 64 and 256. We performed five iterations of pre-training and selected the best-performing model for each downstream task. For SimCLR and PAPAGEI-P, a single model consistently achieves the best performance across all tasks. For BYOL, we select two models that perform best across all tasks. Similarly, for TF-C and PAPAGEI-S, we choose three models with the highest performance. We use this approach as some models excel in certain task groups while others perform better in the rest. Note that a more robust approach would involve broader hyperparameter tuning with k-fold validation to obtain the optimal model. However, this requires substantial computational resources for pre-training. Additionally, we did not perform an exhaustive evaluation of different augmentation settings but instead used transformations and values based on prior research (Abbaspourazad et al., 2023; Tang et al., 2020). For model training, we primarily used PyTorch (Paszke et al., 2019). The NTXentLoss implementation was sourced from the PyTorch Metric Learning package[5].

Table 5: ResNet-style CNN encoder architecture used in PAPAGEI.

| Layer | Output Shape |
|---|---|
| Conv1 | [32, 32, 1250] |
| Batch Norm | [32, 32, 1250] |
| ReLU | [32, 32, 1250] |
| Basic Block Type 1 | [32, 32, 1250] |
| (Basic Block Type 2) × 3 | [32, 32, 313] |
| (Basic Block Type 2) × 4 | [32, 64, 79] |
| (Basic Block Type 2) × 4 | [32, 128, 20] |
| (Basic Block Type 2) × 4 | [32, 256, 5] |
| (Basic Block Type 2) × 2 | [32, 512, 3] |
| BatchNorm | [32, 512, 3] |
| ReLU | [32, 512, 3] |
| Linear | [32, 512] |

Table 6: Basic Block Type 1

| Layer |
|---|
| Conv1D |
| BatchNorm |
| ReLU |
| Dropout |
| Conv1D |

Table 7: Basic Block Type 2

| Layer |
|---|
| BatchNorm |
| ReLU |
| Dropout |
| Conv1D |
| BatchNorm |
| ReLU |
| Dropout |
| Conv1D |
| Maxpool |

---

[3] https://github.com/chengding0713/SiamQuality
[4] https://github.com/mims-harvard/TFC-pretraining
[5] https://github.com/KevinMusgrave/pytorch-metric-learning

**Parameters: Training and Inference**. This section outlines the training and inference parameters used in our methods. Inference parameters are those utilized for feature extraction.

- PAPAGEI-P (5M) and SimCLR (5M): Both training and inference involve 5M parameters. For SimCLR, it is worth noting that we use the projection features during inference instead of using the encoder only.

- BYOL (5M): During training, the online and target encoders each have 5M parameters, and the projector is 800K. At inference, only the online encoder is used for feature extraction, totaling 5M parameters.

- TF-C (10M): The time and frequency encoders each have 5M parameters, followed by a smaller projector ($< 100K$). Since both encoders and projectors are required for inference, the total parameter count is 10M.

- PAPAGEI-S (5M): The encoder consists of 5M parameters, while the expert heads contribute approximately 400K each. As the expert heads are not used for feature extraction, the inference parameter total remains 5M.

**Feature Extraction & Linear Evaluation** We extracted the projected embedding for linear evaluation. For Moment and Chronos, we extract the default embedding size, which is 1024 and 768, respectively. We use cross-validated grid search to identify the best parameters for our linear probes. The hyperparameters chosen for each model are as follows: (1) Logistic Regression: {'penalty': ['l1', 'l2'], 'C': [0.01, 0.1, 1, 10, 100], 'solver': ['lbfgs'], 'max_iter': [100, 200]}. (2) Linear Regression: {'alpha': [0.1, 1.0, 10.0, 100.0], 'solver': ['auto', 'cholesky', 'sparse_cg']}. (3) Random Forest: {'n_estimators': [100, 200], 'max_features': ['sqrt', 'log2'], 'max_depth': [10, 20, 30], 'min_samples_split': [2, 5], 'min_samples_leaf': [1, 2]}

## B  DATASETS AND TASKS

Table 8: The task evaluation benchmark of PAPAGEI. Datasets highlighted in gray are unseen during training, thus, the corresponding tasks are out-of-domain. The rest were used for pre-training but their test sets and labels are held out. For task type, B/M/R refer to Binary classification, Multi-class classification (#classes), and Regression, respectively.

| #ID | Dataset | SR (Hz) | Collected by | Task | Task Type | #Participants (#Samples) |
|---|---|---|---|---|---|---|
| T1 | VitalDB (Lee et al., 2022) | 500 | ICU monitor | ICU admission (Yes/No) | B | 5866 |
| T2 | | | | Operation Type | M (11) | 5866 |
| T3 | MIMIC-III (Moody et al., 2020) | 125 | ICU Monitor | Mortality | B | 5596 |
| T4 | MESA (Zhang et al., 2018) | 256 | Polysomnography finger | Smoker | B | 2055 |
| T5 | | | | AHI > 3% Oxygen Desat. | R | 2055 |
| T6 | | | | AHI > 4% Oxygen Desat. | R | 2055 |
| T7 | nuMom2B (Facco et al., 2015) | 75 | Polysomnography finger | Pregnancy stage (early/late) | B | 3163 (5337) |
| T8 | | | | Gestation Age | R | 3163 (5337) |
| T9 | VV (Skin Tone) (Toye, 2023) | 60 | Finger | Systolic BP | R | 231 |
| T10 | | | | Diastolic BP | R | 231 |
| T11 | PPG-BP (Liang et al., 2018a) | 1000 | Finger Pulse Ox | Systolic BP | R | 219 |
| T12 | | | | Diastolic BP | R | 219 |
| T13 | | | | Average Heart Rate | R | 219 |
| T14 | | | | Hypertension | B | 219 |
| T15 | SDB (Garde et al., 2014) | 62.5 | Finger Pulse Ox | Sleep Disordered Breathing | B | 146 |
| T16 | ECSMP (Gao et al., 2021) | 64 | Wrist | Mood Disturbance | B | 89 |
| T17 | WESAD (Schmidt et al., 2018) | 64 | Wrist | Valence | B | 15 (4497) |
| T18 | | | | Arousal | B | 15 (4497) |
| T19 | PPG-DaLiA (Reiss et al., 2019) | 64 | Wrist | Heart Rate | R | 15 (64697) |
| T20 | | | | Activity | M (9) | 15 (64697) |

**VitalDB.** The VitalDB dataset provides comprehensive monitoring of vital signs and physiological parameters from 6,388 surgical cases. This high-resolution dataset includes a wide range of intraoperative monitoring variables such as heart rate, blood pressure, oxygen saturation, and other critical physiological signals, collected at frequent intervals throughout surgery. The surgical operation belongs to one of the eleven categories: colorectal, biliary/pancreas, stomach, major resection, minor resection, breast, transplantation, thyroid, hepatic, vascular, and others. After the data cleaning process, we narrowed the dataset down to 5,866 participants with complete and usable information. As depicted in Figure 10, we observe that the gender distribution is relatively balanced, with nearly equal representation of male and female patients. Additionally, the majority of the participants fall

within the age range of 50 to 70, with a significant proportion being around 60 years old. The ICU label corresponds to whether the person was admitted to the ICU or not.

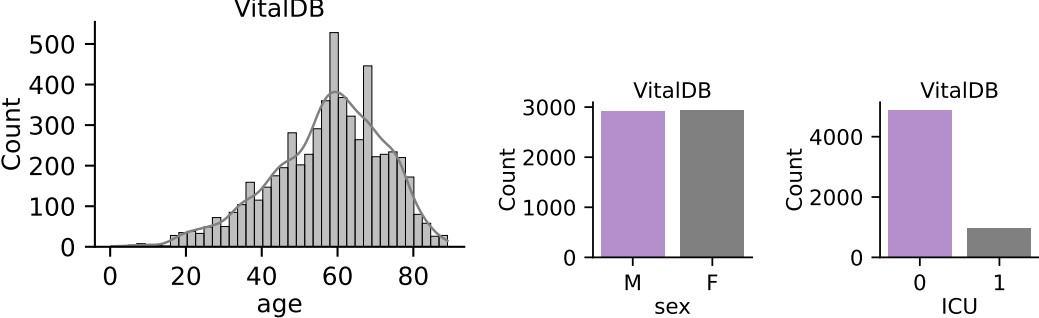

Figure 10: VitalDB dataset descriptive statistics.

**MIMIC-III.** In our analysis, we utilize the MIMIC-III waveform database matched subset, which comprises data from 10,282 ICU patients. From this dataset, we focus specifically on extracting photoplethysmogram (PPG) data, provided it is available for each patient. To ensure the quality of the data, we set a criterion of at least 1 minute of usable PPG signal that must be present. After performing a thorough data cleaning process, we end up with a cohort of 5,596 participants with reliable PPG data. As illustrated in Figure 11, the dataset shows a gender imbalance, with a higher proportion of male patients compared to female patients. Additionally, the majority of participants are aged 60 years or older, reflecting a typical ICU population that often includes elderly patients with critical health conditions.

**MESA.** The Multi-Ethnic Study of Atherosclerosis (MESA) sleep sub-study gathered data from 2,237 participants through overnight, unattended polysomnography to assess various sleep parameters. After the data cleaning process, we retained 2,055 participants for analysis. As shown in Figure 12, the dataset shows a slightly larger proportion of female participants. The age distribution reveals that most participants are between 60 and 80 years old, reflecting an older adult population, which is commonly studied concerning sleep disorders and cardiovascular risks.

In this study, we use the Apnea-Hypopnea Index (AHI) with at least 3% and 4% oxygen desaturation as the primary measure for diagnosing sleep apnea, as recommended by the American Academy of Sleep Medicine (Ruehland et al., 2009). These thresholds indicate the severity of sleep apnea, with oxygen desaturation during apneas/hypopneas being a critical factor. We predict these AHI values directly in our regression models. Additionally, we classify participants with any history of smoking as smokers. This approach allows us to account for both current and former smokers, capturing a broader range of smoking-related health risks within our analysis.

**NuMoM2B.** Changes in gestational age and pregnancy stage are risk factors associated with adverse pregnancy outcomes such as hypertensive disorders and small-for-gestational-age delivery (Bouariu et al., 2022; Parikh et al., 2021; Wu et al., 2020; Crump et al., 2023). These diseases affect heart

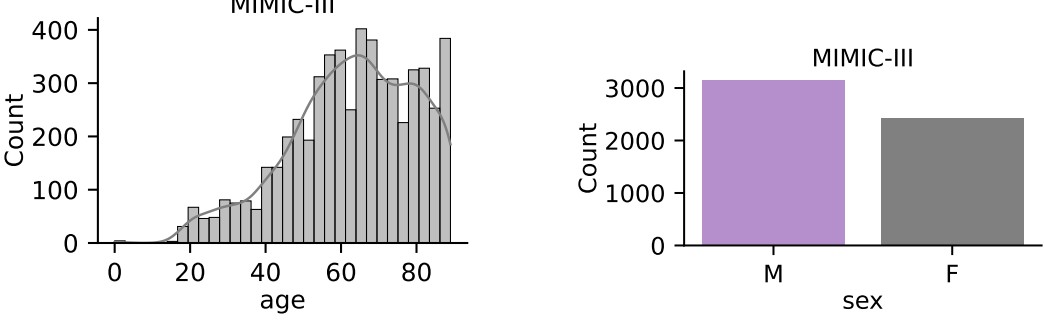

Figure 11: MIMIC-III dataset descriptive statistics.

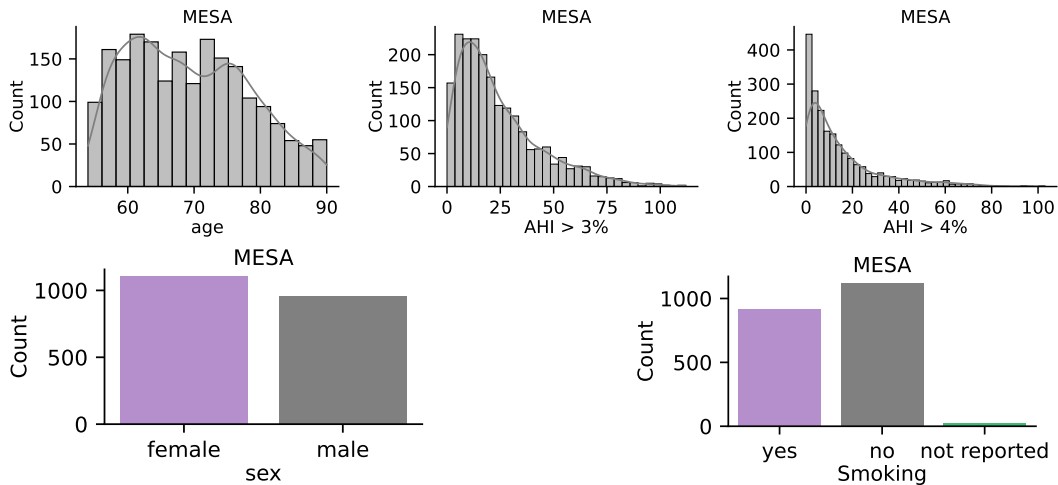

Figure 12: MESA dataset descriptive statistics.

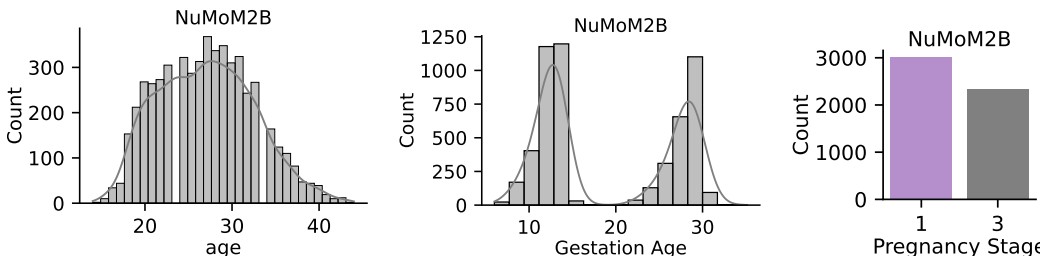

Figure 13: NuMoM2B dataset descriptive statistics.

function that can be measured using the PPG sensor (Feli et al., 2024). The Nulliparous Pregnancy Outcomes Study: monitoring mothers-to-be (nuMoM2B) sub-study examines the relationship between adverse pregnancy outcomes and sleep disorders. In particular, an overnight polysomnograph that collects PPG data is administered to the women at their homes during 6-15 weeks (early) and 22-31 weeks (late) of pregnancy. Therefore, our tasks are to classify between early and late-stage pregnancy as well as predict the gestation age of the fetus. In Figure 13, we observe that maternal age peaks around 28 years. The gestational age distribution is bimodal, which we use as a predictor in our regression task. For pregnancy stage, we classify visit 1 as early and visit 3 as late.

**VitalVideos (VV) (Skin Tone).** The Vital Videos study is an ongoing project that collects data on vital signs, videos, and blood pressure across a variety of conditions, including variations in lighting, background, and skin tone. For our analysis, we used data from two groups, totalling 231 participants, from Europe and Sub-Saharan Africa. As shown in Figure 14, most participants have a Fitzpatrick skin tone of 5 or 6, indicating darker skin. The dataset is primarily composed of female participants, with an age range between 40 and 60 years. Additionally, the majority of participants had a systolic blood pressure of around 125 and a diastolic pressure of around 80, suggesting that most individuals in the study were relatively healthy.

**PPG-BP.** The PPG-BP consists of short PPG recordings from 219 participants collected at 1000Hz. For each subject, there are three 2.1s recordings. For our analysis, we zero pad them to 10s. In Figure 15, the age distribution shows that most participants are between 40 and 80 years old, with fewer participants under 40. Furthermore, the majority of individuals have hypertension. In terms of gender, the dataset has slightly more females than males. The distribution of systolic blood pressure is centered around 120-140, indicating a population with normal to moderately elevated blood pressure, while diastolic blood pressure predominantly falls between 70 and 90. Lastly, the average heart rate for most participants ranges between 70 and 90 beats per minute.

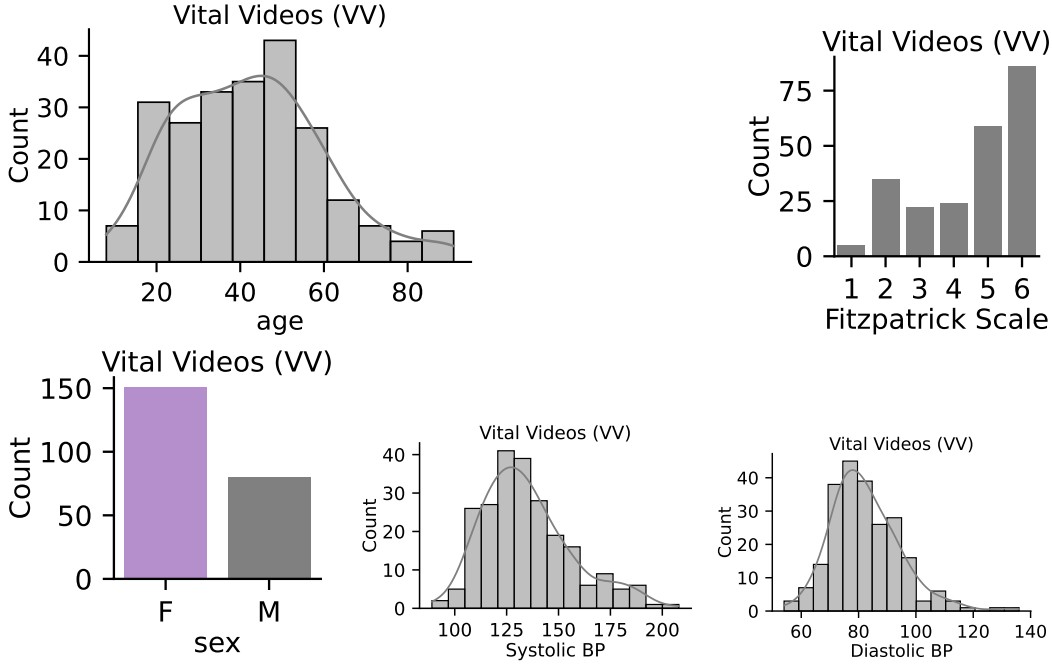

Figure 14: Vital Videos dataset descriptive statistics.

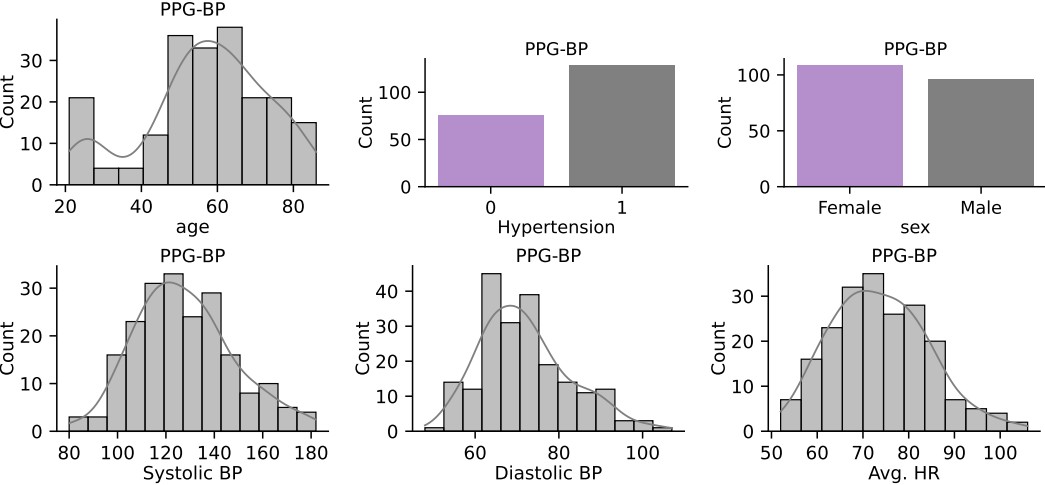

Figure 15: PPG-BP dataset descriptive statistics.

**SDB.** The sleep-disordered breathing dataset includes data from 146 children, collected through polysomnography with finger recordings lasting over three hours. Ground truth labels are provided as Apnea-Hypopnea Index (AHI) values, categorized into four levels: 0 (normal), 1 (mild, AHI between 5 and 15), 2 (moderate, AHI between 15 and 30), and 3 (severe, AHI over 30) as shown in Figure 16. For our classification task, we group AHI 0 as indicating no sleep breathing disorder, while AHI levels 1 through 3 are classified as the presence of a sleep breathing disorder.

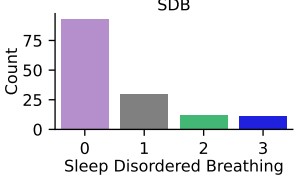

Figure 16: SDB dataset descriptive statistics.

**ECSMP.** The ECSMP dataset was gathered to study the relationship between emotion, cognition, and sleep in 89 participants. As shown in Figure 17, the majority of the participants are young adult females, with an average age of around 25 years. Mood disturbances were measured using the Profile of Mood States (POMS) scale, which captures various aspects of emotional states. To classify participants into high versus low mood disturbance categories, we binarized the Total Mood Disturbance (TMD) values by using the median as the cutoff point.

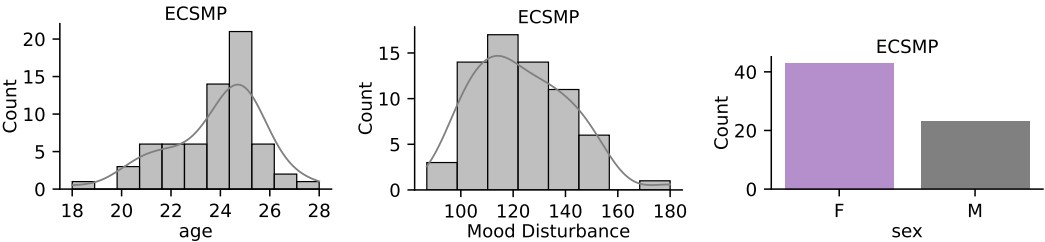

Figure 17: ECSMP dataset descriptive statistics.

**WESAD.** The wearable stress and affect detection dataset is a multi-modal dataset collected from 15 participants using various sensor modalities. In this study, participants were exposed to videos designed to elicit different affective states, such as amusement, meditation, stress, and baseline conditions. Following each session, participants completed the Self-Assessment Manikins (SAM) questionnaire (Bradley & Lang, 1994), which provided the ground-truth values for valence and arousal. In our analysis, we binarized these values by categorizing valence and arousal as low (1) when less than 5 and high (0) otherwise. Then, we perform regression at the segment level. As shown in Figure 18, arousal levels are generally low, while valence tends to be high in most cases.

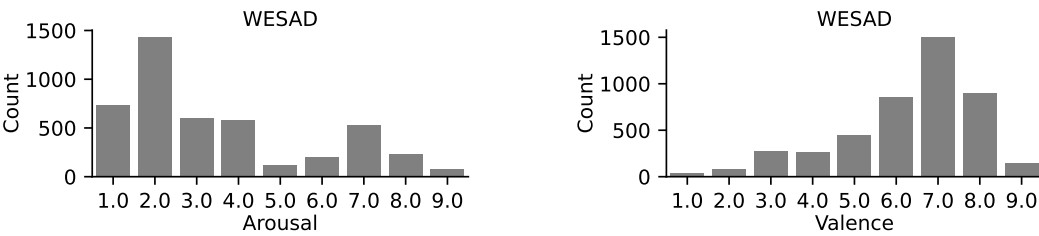

Figure 18: WESAD dataset descriptive statistics.

**PPG-DaLiA.** This dataset collects PPG signals from 15 participants for heart rate estimation while performing various daily activities. These activities include sitting, ascending/descending stairs, table soccer, cycling, driving, lunch break, walking, and working. As a result, the dataset captures a wide range of heart rates, varying from 60 to 150 beats per minute, depending on the specific activity being performed. To align the PPG signal with the activity labels, we use a 8s window with 6s and 2s overlap and shift, respectively. After this, we resample and pad the signal to facilitate modeling.

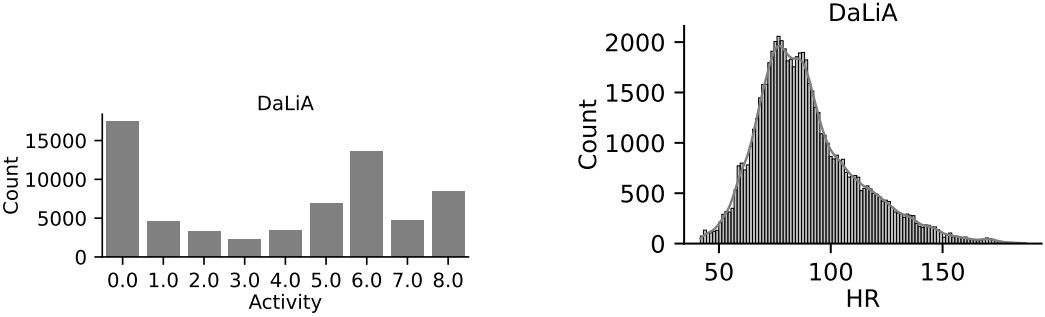

Figure 19: PPG-DaLiA dataset descriptive statistics.

## C    REPRESENTATIVE SIGNALS FROM PRE-TRAINING DATASETS

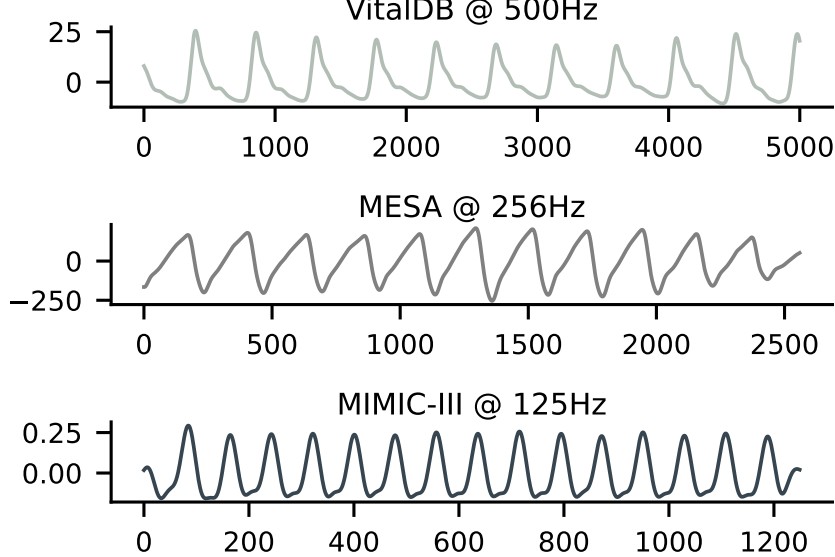

Figure 20: Representative 10-second *raw* PPG segments from VitalDB, MESA, and MIMIC-III. We observe that each signal's amplitude (y-axis) and sampling rate differ.

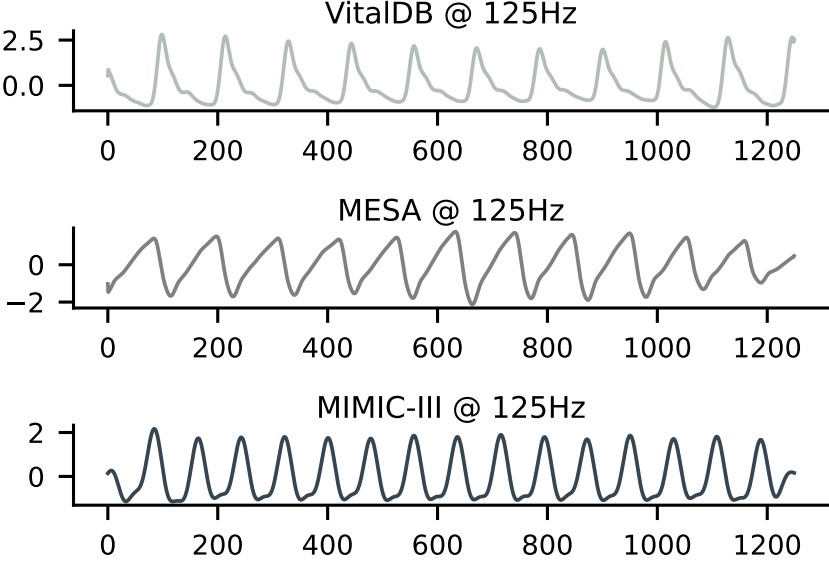

Figure 21: Normalized and resampled 10-second *pre-processed* PPG segments from VitalDB, MESA, and MIMIC-III. These signals represent the final form before being fed to our models. We observe that the signal characteristics across datasets are more consistent.

# D ADDITIONAL RESULTS

## D.1 MULTI-CLASS CLASSIFICATION

Table 9: **Multi-class classification comparison against pre-trained models.** Feature extraction parameters are indicated next to each name.. 95% CIs are reported in square brackets and the best value is **bolded**.

| Classification - ACC (↑) | REGLE (0.07M) (Yun et al., 2024) | Chronos (200M) (Ansari et al., 2024) | Moment (385M) (Goswami et al., 2024) | PAPAGEI-P (5M) | PAPAGEI-S (5M) |
|---|---|---|---|---|---|
| Operation Type | 0.21 [0.18-0.23] | 0.25 [0.22-0.29] | 0.27 [0.23-0.31] | 0.30 [0.26-0.33] | **0.30** [0.27-0.32] |
| Activity | 0.29 [0.28-0.29] | **0.41** [0.40-0.42] | **0.41** [0.40-0.42] | 0.38 [0.37-0.39] | 0.37 [0.36-0.37] |
| Average | 0.25 ± 0.04 | 0.33 ± 0.08 | **0.34 ± 0.07** | **0.34 ± 0.04** | 0.33 ± 0.03 |

Table 10: **Multi-class classification comparison against CL methods**. Feature extraction parameters are indicated next to each name.. 95% CIs are reported in square brackets and the best value is **bolded**.

| Classification - ACC (↑) | Stat. Features | SimCLR (5M) (Chen et al., 2020) | BYOL (5M) (Grill et al., 2020) | TF-C (10M) (Zhang et al., 2022) | PAPAGEI-P (5M) | PAPAGEI-S (5M) |
|---|---|---|---|---|---|---|
| Operation Type | 0.27 [0.24-0.32] | 0.27 [0.24-0.29] | **0.31** 0.27-0.34 | 0.27 [0.27-0.29] | 0.30 [0.26-0.33] | 0.30 [0.27-0.32] |
| Activity | 0.37 [0.36-0.38] | 0.36 [0.35-0.37] | 0.34 [0.33-0.35] | 0.37 [0.36-0.38] | **0.38** [0.37-0.39] | 0.37 [0.36-0.37] |
| Average | 0.32 ± 0.05 | 0.31 ± 0.04 | 0.32 ± 0.01 | 0.32 ± 0.05 | **0.34 ± 0.04** | 0.33 ± 0.03 |

## D.2 F1-SCORE AND $R^2$ EVALUATION METRICS.

Table 11: **Downstream comparison against pre-trained models (additional metrics: F1-score and $R^2$).** Feature extraction parameters are indicated next to each name. 95% CIs are reported in square brackets and the best value is **bolded**.

| Classification - F1-Score (↑) | REGLE (0.07M) (Yun et al., 2024) | Chronos (200M) (Ansari et al., 2024) | Moment (385M) (Goswami et al., 2024) | PAPAGEI-P (5M) | PAPAGEI-S (5M) |
|---|---|---|---|---|---|
| ICU Admission | 0.00 [0.00-0.00] | 0.20 [0.11-0.30] | 0.12 [0.04-0.20] | 0.12 [0.04-0.20] | **0.26** [0.18-0.33] |
| Mortality | 0.00 [0.00-0.00] | 0.14 [0.09-0.19] | 0.16 [0.11-0.21] | **0.22** [0.16-0.27] | 0.17 [0.13-0.22] |
| Smoker | 0.16 [0.10-0.23] | **0.51** [0.44-0.58] | 0.40 [0.33-0.47] | 0.45 [0.38-0.51] | 0.45 [0.37-0.50] |
| Pregnancy stage | 0.49 [0.47-0.52] | **0.69** [0.67-0.71] | 0.63 [0.60-0.65] | 0.62 [0.59-0.64] | 0.65 [0.62-0.67] |
| Hypertension | 0.77 [0.70-0.84] | 0.68 [0.58-0.77] | 0.75 [0.66-0.84] | **0.84** [0.72-0.92] | 0.78 [0.70-0.86] |
| Sleep Disordered Breathing | 0.00 [0.00-0.00] | 0.33 [0.00-0.60] | 0.22 [0.00-0.47] | 0.32 [0.00-0.60] | **0.47** [0.23-0.67] |
| Mood Disturbance | 0.00 [0.00-0.00] | 0.36 [0.10-0.59] | 0.23 [0.00-0.47] | **0.37** [0.00-0.66] | 0.32 [0.00-0.58] |
| Valence | 0.00 [0.00-0.00] | 0.10 [0.07-0.14] | 0.12 [0.09-0.16] | **0.17** [0.13-0.21] | 0.03 [0.01-0.04] |
| Arousal | 0.83 [0.81-0.84] | 0.82 [0.80-0.83] | 0.81 [0.79-0.82] | 0.81 [0.79-0.82] | **0.83** [0.81-0.84] |
| Average | 0.25 ± 0.33 | 0.42 ± 0.24 | 0.38 ± 0.26 | 0.43 ± 0.25 | **0.44 ± 0.25** |
| **Regression** - $R^2$ (↑) | | | | | |
| Apnea/Hypopnea Index > 3% | 0.02 [0.00-0.03] | 0.18 [0.08-0.26] | 0.14 [0.06-0.22] | 0.15 [0.05-0.24] | **0.29** [0.22-0.36] |
| Apnea/Hypopnea Index > 4% | 0.01 [0.00-0.03] | 0.16 [0.08-0.22] | 0.13 [0.05-0.20] | 0.12 [0.03-0.22] | **0.28** [0.20-0.34] |
| Gestation Age | 0.04 [0.02-0.06] | **0.28** [0.24-0.31] | 0.20 [0.17-0.23] | 0.18 [0.14-0.22] | 0.22 [0.19-0.25] |
| Systolic BP (VV) | -0.03 [-0.18-0.01] | -0.24 [-0.72-0.03] | 0.06 [-0.25-0.28] | -0.41 [-0.77-(-0.15)] | **0.15** [-0.09-0.30] |
| Diastolic BP (VV) | 0.01 [-0.09-0.06] | -0.29 [-0.87-(-0.01)] | 0.01 [-0.25-0.14] | -0.48 [-1.02-(-0.20)] | **0.10** [-0.11-0.23] |
| Systolic BP (PPG-BP) | -0.07 [-0.21-0.04] | -0.13 [-0.36-0.06] | 0.07 [-0.31-0.31] | **0.36** [0.16-0.49] | 0.20 [0.02-0.31] |
| Diastolic BP (PPG-BP) | 0.01 [-0.05-0.02] | -0.10 [-0.45-0.07] | -0.03 [-0.31-0.13] | **0.22** [-0.13-0.40] | 0.08 [-0.07-0.17] |
| Average HR | 0.37 [0.17-0.51] | 0.02 [-0.16-0.17] | 0.68 [0.45-0.80] | **0.79** [0.57-0.90] | 0.78 [0.69-0.83] |
| HR | 0.00 [0.00-0.01] | 0.57 [0.56-0.59] | **0.63** [0.61-0.64] | 0.52 [0.51-0.53] | 0.48 [0.42-0.46] |
| Average | 0.04 ± 0.12 | 0.05 ± 0.25 | 0.21 ± 0.24 | 0.16 ± 0.38 | **0.28 ± 0.20** |

## D.3 ABLATION RESULTS

In this section, we provide the numeric results for the scaling analysis (Table 13) and PAPAGEI-S component analysis (Table 14).

Table 12: **Downstream comparison against CL models (additional metrics: F1-score and $R^2$).** Feature extraction parameters are indicated next to each name. 95% CIs are reported in square brackets and the best value is **bolded**.

| **Classification** - F1-Score ($\uparrow$) | **Stat. Features** | **SimCLR** (5M) | **BYOL** (5M) | **TF-C** (10M) | **PAPAGEI-P** (5M) | **PAPAGEI-S** (5M) |
|---|---|---|---|---|---|---|
| ICU Admission | **0.30** [0.18-0.40] | 0.19 [0.12-0.26] | 0.17 [0.11-0.22] | 0.10 [0.05-0.16] | 0.12 [0.04-0.20] | 0.26 [0.18-0.33] |
| Mortality | 0.03 [0.01-0.06] | 0.15 [0.10-0.20] | 0.13 [0.08-0.17] | 0.15 [0.10-0.20] | **0.22** [0.16-0.27] | 0.17 [0.13-0.22] |
| Smoker | 0.47 [0.40-0.53] | 0.43 [0.35-0.49] | **0.49** [0.42-0.56] | 0.37 [0.30-0.44] | 0.45 [0.38-0.51] | 0.45 [0.37-0.50] |
| Pregnancy stage | 0.43 [0.41-0.47] | 0.60 [0.57-0.63] | 0.60 [0.57-0.63] | 0.59 [0.56-0.62] | 0.62 [0.59-0.64] | **0.65** [0.62-0.67] |
| Hypertension | 0.73 [0.58-0.85] | 0.82 [0.73-0.88] | 0.81 [0.73-0.88] | 0.81 [0.72-0.89] | **0.84** [0.72-0.92] | 0.78 [0.70-0.86] |
| Sleep Disordered Breathing | 0.00 [0.00-0.00] | 0.46 [0.21-0.64] | 0.45 [0.23-0.62] | 0.19 [0.00-0.36] | 0.32 [0.00-0.60] | **0.47** [0.23-0.67] |
| Mood Disturbance | 0.21 [0.00-0.47] | 0.21 [0.00-0.44] | 0.37 [0.00-0.67] | **0.56** [0.27-0.80] | 0.37 [0.00-0.66] | 0.32 [0.00-0.58] |
| Valence | 0.04 [0.02-0.07] | 0.09 [0.06-0.12] | 0.01 [0.00-0.03] | 0.07 [0.04-0.09] | **0.17** [0.13-0.21] | 0.03 [0.01-0.04] |
| Arousal | 0.82 [0.81-0.83] | 0.81 [0.79-0.82] | **0.83** [0.81-0.84] | 0.81 [0.80-0.83] | 0.81 [0.79-0.82] | **0.83** [0.81-0.84] |
| Average | 0.33 $\pm$ 0.28 | 0.42 $\pm$ 0.26 | 0.43 $\pm$ 0.27 | 0.40 $\pm$ 0.27 | 0.43 $\pm$ 0.25 | **0.44 $\pm$ 0.25** |
| **Regression** - $R^2$ ($\uparrow$) | | | | | | |
| Apnea/Hypopnea Index > 3% | -0.00 [-0.06-0.03] | 0.16 [0.07-0.23] | 0.16 [0.08-0.22] | 0.06 [-0.00-0.13] | 0.15 [0.05-0.24] | **0.29** [0.22-0.36] |
| Apnea/Hypopnea Index > 4% | -0.01 [-0.07-0.03] | 0.13 [0.06-0.21] | 0.13 [0.05-0.19] | 0.13 [-0.06-0.26] | 0.12 [0.03-0.22] | **0.28** [0.20-0.34] |
| Gestation Age | 0.07 [0.04-0.10] | 0.19 [0.15-0.21] | 0.19 [0.15-0.22] | 0.18 [0.15-0.21] | 0.18 [0.14-0.22] | **0.22** [0.19-0.25] |
| Systolic BP (VV) | -0.10 [-0.51-0.10] | -0.05 [-0.44-0.21] | -0.03 [-0.37-0.18] | -0.05 [-0.36-0.12] | -0.41 [-0.77-(-0.15)] | **0.15** [-0.09-0.30] |
| Diastolic BP (VV) | -0.15 [-0.31-0.11] | -0.14 [-0.29-0.08] | -0.01 [-0.40-0.20] | -0.09 [-0.45-0.16] | -0.48 [-1.02-(-0.20)] | **0.10** [-0.11-0.23] |
| Systolic BP (PPG-BP) | 0.12 [-0.04-0.21] | 0.09 [-0.20-0.31] | 0.10 [-0.16-0.30] | 0.13 [-0.06-0.26] | **0.36** [0.16-0.49] | 0.20 [0.02-0.31] |
| Diastolic BP (PPG-BP) | 0.01 [-0.18-0.14] | 0.00 [-0.20-0.18] | 0.05 [-0.11-0.17] | 0.02 [-0.15-0.12] | **0.22** [-0.13-0.40] | 0.08 [-0.07-0.17] |
| Average HR | 0.15 [-0.10-0.33] | 0.74 [0.64-0.80] | 0.65 [0.50-0.77] | **0.82** [0.73-0.88] | 0.79 [0.57-0.90] | 0.78 [0.69-0.83] |
| HR | 0.34 [0.32-0.36] | 0.45 [0.44-0.47] | 0.36 [0.35-0.37] | **0.54** [0.53-0.55] | 0.52 [0.51-0.53] | 0.48 [0.42-0.46] |
| Average | 0.05 $\pm$ 0.14 | 0.17 $\pm$ 0.25 | 0.18 $\pm$ 0.20 | 0.19 $\pm$ 0.28 | 0.16 $\pm$ 0.38 | **0.28 $\pm$ 0.20** |

Table 13: **Scaling: Downstream comparison for different PAPAGEI-S models.** 95% CIs are reported in square brackets and the best value is **bolded**.

| **Classification** - AUROC ($\uparrow$) | **PAPAGEI-S-5M** | **PAPAGEI-S-35M** | **PAPAGEI-S-139M** |
|---|---|---|---|
| ICU Admission | **0.79** [0.75-0.82] | 0.72 [0.68-0.75] | 0.77 [0.73-0.80] |
| Mortality | **0.67** [0.63-0.70] | 0.66 [0.63-0.70] | 0.66 [0.63-0.69] |
| Smoker | **0.61** [0.56-0.66] | 0.58 [0.52-0.64] | 0.59 [0.54-0.65] |
| Pregnancy stage | **0.78** [0.75-0.80] | 0.77 [0.75-0.79] | 0.76 [0.74-0.78] |
| Hypertension | **0.77** [0.68-0.87] | 0.75 [0.64-0.85] | **0.77** [0.65-0.87] |
| Sleep Disordered Breathing | **0.70** [0.57-0.84] | 0.59 [0.44-0.74] | 0.62 [0.46-0.78] |
| Mood Disturbance | **0.56** [0.33-0.77] | 0.53 [0.30-0.73] | 0.54 [0.29-0.78] |
| Valence | **0.56** [0.54-0.59] | 0.53 [0.50-0.56] | 0.54 [0.51-0.56] |
| Arousal | **0.55** [0.52-0.57] | 0.52 [0.49-0.55] | **0.55** [0.52-0.58] |
| Average | **0.67 $\pm$ 0.09** | 0.63 $\pm$ 0.09 | 0.63 $\pm$ 0.10 |
| **Regression** - MAE ($\downarrow$) | | | |
| Apnea/Hypopnea Index > 3% | 12.97 [11.87-14.05] | 13.07 [11.92-14.25] | **12.86** [11.79-13.94] |
| Apnea/Hypopnea Index > 4% | **10.56** [9.59-11.62] | 10.79 [9.85-11.83] | 10.65 [9.62-11.68] |
| Gestation Age | **6.05** [5.91-6.17] | 6.10 [5.94-6.24] | 6.17 [6.02-6.30] |
| Systolic BP (VV) | **14.65** [12.50-16.78] | 15.10 [13.10-17.21] | 14.95 [12.87-17.01] |
| Diastolic BP (VV) | **8.29** [6.61-10.22] | 9.20 [6.93-11.12] | 8.95 [6.72-10.95] |
| Systolic BP (PPG-BP) | **14.39** [12.53-16.45] | 16.70 [14.25-19.38] | 16.20 [13.73-18.85] |
| Diastolic BP (PPG-BP) | **8.71** [7.18-10.01] | 9.48 [8.24-10.90] | 9.32 [7.90-10.69] |
| Average HR | **4.00** [3.34-4.67] | 4.76 [3.94-5.86] | 4.71 [3.86-5.60] |
| HR | **11.53** [11.40-11.66] | 12.86 [12.73-12.99] | 12.20 [12.07-12.34] |
| Average | **10.12 $\pm$ 3.47** | 10.89 $\pm$ 3.73 | 10.76 $\pm$ 3.57 |

## D.4 STATISTICAL SIGNIFICANCE OF MODEL COMPARISON

In addition to confidence intervals, we perform the following steps to evaluate the significance across models on a per task basis (Tables 3 & 4). First, we ran the Friedmann Chi Square test, and identified statistically significant differences across PAPAGEI and the baseline models at $p < 0.05$. Next, we created critical difference (CD) diagrams to rank the best performing models, as suggested by the literature to compare models over multiple datasets[6] (Demšar, 2006). The CDs indicate that PAPAGEI performs the best across both classification and regression tasks. Furthermore, it has a statistically significant average rank as indicated by the lack of horizontal line.

---

[6] https://scikit-posthocs.readthedocs.io/en/latest/tutorial.html#critical-difference-diagrams

Table 14: **PAPAGEI component ablation study results.**

| | sVRI | sVRI + SQI | sVRI + IPA | Full |
|---|---|---|---|---|
| **Classification** - AUROC (↑) | | | | |
| ICU Admission | **0.79** | 0.75 | 0.78 | **0.79** |
| Mortality | **0.67** | 0.65 | **0.67** | **0.67** |
| Smoker | 0.59 | **0.61** | 0.60 | **0.61** |
| Pregnancy stage | **0.78** | 0.73 | 0.72 | **0.78** |
| Hypertension | **0.77** | 0.72 | 0.75 | **0.77** |
| Sleep Disordered Breathing | 0.62 | 0.53 | 0.64 | **0.70** |
| Mood Disturbance | 0.53 | **0.56** | 0.55 | **0.56** |
| Valence | 0.54 | 0.55 | 0.53 | **0.56** |
| Arousal | 0.44 | 0.51 | 0.49 | **0.55** |
| **Regression** - MAE (↓) | | | | |
| Apnea/Hypopnea Index > 3% | 13.36 | 13.74 | 13.42 | **12.97** |
| Apnea/Hypopnea Index > 4% | 11.01 | 11.43 | 11.29 | **10.56** |
| Gestation Age | 6.18 | 6.32 | 6.15 | **6.05** |
| Systolic BP (VV) | **14.62** | 15.97 | 15.33 | 14.65 |
| Diastolic BP (VV) | 8.32 | 8.76 | 9.04 | **8.29** |
| Systolic BP (PPG-BP) | 15.03 | **14.39** | 16.15 | **14.39** |
| Diastolic BP (PPG-BP) | 9.12 | 8.76 | 9.06 | **8.71** |
| Average HR | **4.00** | 5.88 | 4.26 | **4.00** |
| HR | **11.51** | 11.97 | 11.88 | 11.53 |

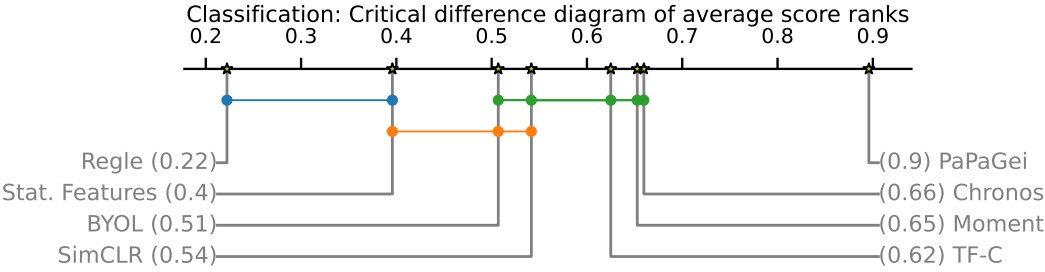

Figure 22: **Critical Difference Diagram for Classification Tasks**. The axis represents the average rank of the model. The horizontal connector lines indicate no significant differences between the models.

From the critical difference diagrams we observe that PAPAGEI is significantly better across classification (Figure 22) and regression (Figure 23) tasks. This arises because PAPAGEI is the highest ranking model across most tasks. Furthermore, we observe Moment is a strong model across both classification and regression tasks. Whereas Chronos and TF-C perform well for classification tasks only.

We conduct additional statistical significance comparisons using a structured approach. First, we randomly sample a score from within the confidence intervals for each task across all models. Next,

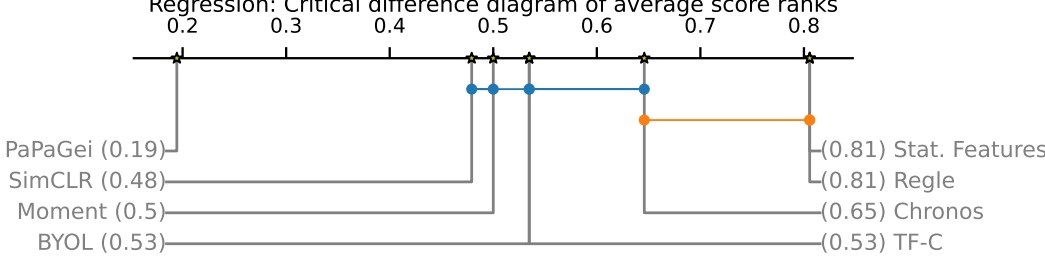

Figure 23: **Critical Difference Diagram for Regression Tasks**. The axis represents the average rank of the model. The horizontal connector lines indicate no significant differences between the models.

we apply the CD ranking procedure to the sampled scores. This process is repeated 1,000 times, and the ranks are averaged. The entire experiment is conducted five times for Tables 3 and 4, with the results presented in Figure 24. The colored cells indicate that PAPAGEI is statistically significant compared to the respective model at $p < 0.05$. Our findings show that PaPaGei consistently achieves the best average rank, ranging between 0.82-0.90 for AUROC and 0.19-0.25 for MAE. Across 35 comparisons (PaPaGei vs. the other models, repeated five times), PaPaGei demonstrates significant improvements in 30 out of 35 AUROC comparisons and 32 out of 35 MAE comparisons. Among the baseline models, we acknowledge that Chronos and TF-C are strong competitors capable of performing comparably to PAPAGEI.

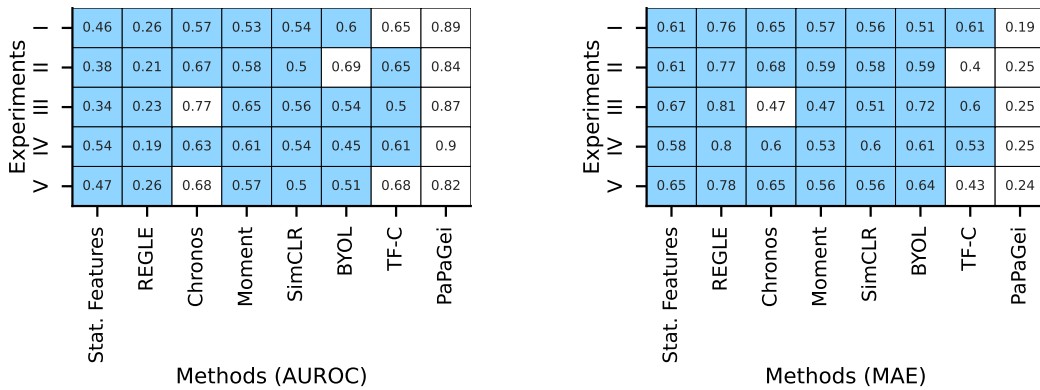

Figure 24: Bootstrap ranking repeated for five experiments: AUROC (left) and MAE (right). Colored cells indicate that PAPAGEI is significant to the baseline at $p < 0.05$.

### D.5 STATISTICAL ASSESSMENT BETWEEN IPA AND SQI

Recall that we incorporate SQI to handle situations where the dicrotic notch cannot be computed due to poor-signal quality or different morphologies. We performed a permutation test (Rice, 2008) to statistically evaluate PPG segments where IPA is unavailable. By splitting SQI values into no IPA and IPA groups and testing significance over 1000 permutations, we observed statistically significant differences ($p < 0.05$) in both mean (+0.18) and median (+0.32) SQI values, with the IPA group having larger SQI. These findings empirically motivate SQI's ability to handle limited PPG morphology.

## E ADDITIONAL BASELINES: DEMOGRAPHICS & PPG MORPHOLOGY FEATURES

In this section, we evaluate the effectiveness of demographics (Demo: age, sex) and PPG morphology (sVRI, IPA, SQI) to predict both regression (ridge) and classification (logistic regression) tasks: **(1) Ablation Study**: We compared PaPaGei with three baselines—demographics alone, PPG features alone, and demographics + PPG features. Our results show that while demographics is a stronger baseline than statistical features, PaPaGei outperforms the demographics + PPG baseline in 14 out of 18 tasks. **(2) Effect of Demographics**: We trained a model combining PaPaGei-S with demographics. The results indicate that incorporating demographic features with PaPaGei-S creates a stronger model than using PaPaGei-S alone.

From Table 15, we observe the following classification performance (Positive is better): ICU (+0.13), Mortality (+0.01), Smoker (-0.02), Pregnancy Stage (+0.22), Hypertension (0.00), SDB (no demographics), Mood Disturbance (-0.08), Valence (-0.01), Arousal (+0.04). Regression Tasks (Negative is better): AHI > 3% (-1.43), AHI > 4% (-1.61), gestation age (-1.54), SBP-VV (-0.31), DBP-VV (-0.46), SBP (-0.11) , DBP (-0.65), Avg. HR (-4.07), HR (-2.93). PaPaGei-S performs better for real-time sleep and cardiovascular outcomes such as sleep apnea, heart rate and blood pressure, respectively. In particular, we notice that outcomes such as heart rate benefit substantially from PPG rather than demographics. Demographics are useful in tasks without real-time dependence such as smoking, which is established to be associated with age and sex (Chung et al., 2020).

Table 15: **Demographics & PPG Morphology Baseline Results.**

| | Stat. Features | Demo | PPG | Demo + PPG | PAPAGEI-S or -P | PAPAGEI-S + Demo |
|---|---|---|---|---|---|---|
| **Classification** - AUROC (↑) | | | | | | |
| ICU Admission | 0.71 [0.65-0.78] | 0.64 [0.60-0.68] | 0.59 [0.54-0.64] | 0.66 [0.61-0.70] | **0.79 [0.75-0.82]** | 0.77 [0.74-0.81] |
| Mortality | 0.57 [0.54-0.61] | 0.66 [0.61-0.69] | 0.57 [0.53-0.61] | 0.66 [0.61-0.69] | 0.67 [0.63-0.70] | **0.70 [0.67-0.74]** |
| Smoker | 0.63 [0.58-0.67] | 0.64 [0.59-0.70] | 0.56 [0.52-0.62] | **0.66 [0.61-0.71]** | 0.64 [0.58-0.69] | 0.62 [0.56-0.68] |
| Pregnancy stage | 0.64 [0.62-0.67] | 0.52 [0.50-0.55] | 0.55 [0.53-0.57] | 0.56 [0.53-0.58] | 0.78 [0.75-0.80] | **0.78 [0.76-0.80]** |
| Hypertension | 0.66 [0.47-0.83] | 0.77 [0.65-0.88] | 0.53 [0.40-0.68] | 0.77 [0.65-0.88] | 0.77 [0.68-0.87] | **0.80 [0.70-0.89]** |
| SDB | 0.32 [0.14-0.55] | – | 0.46 [0.31-0.62] | – | 0.70 [0.57-0.84] | – |
| Mood Disturbance | 0.54 [0.31-0.77] | 0.54 [0.30-0.80] | **0.64 [0.42-0.85]** | 0.63 [0.36-0.87] | 0.56 [0.33-0.77] | 0.52 [0.25-0.78] |
| Valence | 0.52 [0.49-0.55] | 0.57 [0.54-0.60] | 0.44 [0.41-0.47] | **0.57 [0.54-0.60]** | 0.56 [0.54-0.59] | 0.55 [0.53-0.58] |
| Arousal | 0.55 [0.53-0.58] | 0.54 [0.52-0.58] | 0.51 [0.48-0.54] | 0.54 [0.52-0.58] | 0.58 [0.55-0.61] | **0.58 [0.54-0.59]** |
| **Regression** - MAE (↓) | | | | | | |
| Apnea/Hypopnea Index > 3% | 15.31 [13.63-17.14] | 14.53 [13.29-15.84] | 15.09 [14.01-16.54] | 14.40 [13.12-15.61] | 12.97 [11.87-14.05] | **12.35 [11.27-13.46]** |
| Apnea/Hypopnea Index > 4% | 12.52 [10.92-14.14] | 12.28 [11.19-13.39] | 12.65 [11.57-13.83] | 12.17 [11.10-13.39] | 10.56 [9.59-11.62] | **10.47 [9.53-11.50]** |
| Gestation Age | 7.15 [6.99-7.34] | 7.69 [7.61-7.77] | 7.61 [7.51-7.70] | 7.59 [7.51-7.68] | 6.05 [5.91-6.17] | **6.02 [5.88-6.17]** |
| Systolic BP (VV) | 15.76 [13.67-18.36] | 14.96 [13.21-17.35] | 15.82 [13.48-18.31] | 15.01 [13.30-17.86] | 14.65 [12.50-16.78] | **14.27 [11.92-16.44]** |
| Diastolic BP (VV) | 9.75 [7.16-11.27] | 8.75 [6.48-9.77] | 9.20 [7.21-10.71] | 8.78 [7.10-10.25] | 8.29 [6.61-10.22] | **8.26 [6.64-10.16]** |
| Systolic BP (PPG-BP) | 15.50 [11.68-20.25] | 13.71 [11.33-15.95] | 15.76 [13.36-18.30] | 13.74 [11.37-16.09] | 13.60 [10.65-16.51] | **13.20 [11.47-15.66]** |
| Diastolic BP (PPG-BP) | 9.35 [7.44-11.66] | 9.26 [7.89-10.68] | 9.36 [7.95-10.92] | 9.28 [8.00-10.56] | 8.71 [7.18-10.01] | **8.61 [7.34-9.88]** |
| Average HR | 7.01 [5.48-8.89] | 9.12 [7.86-10.61] | 8.07 [6.60-9.71] | 8.23 [6.82-9.78] | **3.47 [2.74-4.32]** | 4.00 [3.35-4.73] |
| HR | 13.07 [12.90-13.23] | 15.18 [15.03-15.33] | 16.75 [16.60-16.90] | 14.46 [14.32-14.62] | **10.92 [10.80-11.04]** | 12.38 [11.90-12.96] |

Importantly, demographics do not add much to already homogeneous populations. For example, consider the NuMoM2B dataset which has women within a specific age range. Here, we observe that PaPaGei obtains much higher AUROC and MAE than the supervised baselines.

Furthermore, We observe that adding demographics to PaPaGei-S embeddings improves over Pa-PaGei in the following tasks: Mortality (+0.03), Hypertension (+0.03), AHI > 3% (-0.62), AHI > 4% (-0.09), gestation age (-0.03), SBP VV (-0.38), DBP VV (-0.03), SBP (0.40), DBP (0.10). Based on these results, PaPaGei-S + Demo is a stronger model in many cases. Importantly, these results indicate that PaPaGei-S embeddings learn features that are complementary to demographics are not simply proxies for age or sex. However, it is important to note that while demographic features can be valuable for personalization, they may not always be readily available, and in reality, we cannot use them in isolation to predict real-time outcomes such as blood pressure or heart rate. Therefore, our PaPaGei models are designed to function effectively with real-time sensor data alone, ensuring their applicability in situations where complete demographic information is not accessible.

These findings underscore an important point: **demographic features are not competing with Pa-PaGei but rather complement it**, as previously established in studies including demographics with sensor data (Spathis et al., 2022). This highlights the synergistic potential of combining PaPaGei's advanced feature extraction with demographic context for improved task performance.

**Predicting Demographics Targets.** Using the PAPAGEI features, we predict downstream demographics such as age and sex (Table 16). PAPAGEI-S achieves 7.78 MAE in age regression, 0.85 accuracy in age classification, and 0.79 accuracy in sex classification. Although our results trail larger closed studies (Abbaspourazad et al., 2023) by 2.18, 0.05, and 0.13 for segment-level SSL, and by 5.59, 0.12, and 0.25 for patient-level SSL, they mark an advancement in open-source efforts. The superior performance of Abbaspourazad et al. (2023) can be attributed to two primary factors. First, the patient-level positive pair strategy achieves the best performance across all tasks. This approach encourages the model to form distinct clusters for each patient, effectively capturing demographic factors such as age and sex. In contrast, a segment-level approach pushes the model to cluster similar segments across individuals with varying demographics, potentially mixing demographic-specific information. Second, the single-device setup with a larger dataset is useful for effective model training (Table 17). Conversely, our evaluation, which spans three devices and utilizes smaller datasets, must handle greater data heterogeneity, thus making it more challenging.

## F    ADDITIONAL PREDICTION PLOTS

The regression plots to evaluate the agreement between true and predicted values in shown in Figure 25. From the Figure, we observe that PAPAGEI's predictions are more aligned to the true values for AHI > 3% ($R^2 = 0.28$), Avg. HR ($R^2 = 0.79$), gestation age ($R^2 = 0.28$), SBP ($R^2 = 0.36$), and DBP ($R^2 = 0.22$). Moreover, from the distribution plots in Figure 25, we notice that PAPAGEI has stronger overlap for AHI > 4%, Avg. HR and DBP, indicating its ability to capture the tails for

Table 16: **Predicting personal characteristics with embeddings**. Downstream prediction on age regression, age classification, and sex classification in our pre-training datasets (VitalDB, MESA, MIMIC-III). The regression and classification tasks are reported using MAE and AUROC, respectively. Note: training and testing are conducted with completely different cohorts in the two studies, hence comparisons are difficult.

| Study | Age Regression ($\downarrow$) | Age Classification ($\uparrow$) | Sex Classification ($\uparrow$) |
|---|---|---|---|
| Abbaspourazad et al. (2023)(Patient) | 3.19 | 0.97 | 0.99 |
| Abbaspourazad et al. (2023)(Segment) | 6.60 | 0.90 | 0.87 |
| PAPAGEI-S (Ours) | 8.78 [8.47-8.09] | 0.85 [0.83-0.87] | 0.74 [0.72-0.76] |

Table 17: **Comparison of large-scale PPG studies.** * indicates partial availability. The participants and hours indicate pre-training data.

| Study | #Participants (#Hours) | #Devices (Types) | Open Data | Open Weights | Open Code | #Tasks (#Datasets) |
|---|---|---|---|---|---|---|
| Abbaspourazad et al. (2023) | 141,207 (333K) | 1 (Smartwatch) | ✗* | ✗ | ✗ | >46 (1) |
| Ding et al. (2024) | 28,539 (300K) | 5-6 (ICU, Smartwatch) | ✗* | ✗ | ✓ | 4 (7) |
| Yun et al. (2024) | 170,714 (Varying)[7] | 1 (Finger) | ✗ | ✓ | ✓* | 2 (4) |
| PAPAGEI (Ours) | 13,517 (57K) | 7 (ICU, Smartwatch, Finger, "Phone Ox.") | ✓ | ✓ | ✓ | 20 (10) |

these tasks. Interestingly, we notice that all models are unable to capture the bi-modal nature of the gestation age measurements. Here, Chronos performs better than other methods to capture readings from the first visit.

## G    EFFECT OF SKIN TONE

We present the skin tone analysis in a more granular way in Figure 26. Here, PAPAGEI-S clearly performs better than PAPAGEI-P in most cases. Overall, we notice that PAPAGEI-S is good for lighter skin tones in the 1-2 range for SBP and 2-3 range for DBP. While PAPAGEI-S does not perform the best for darker skin tones, it's performance is comparable to other models for skin tone ratings of 4 and 5. Overall, these results indicate that PAPAGEI-S is relatively robust to skin tone variations, and that additional future work is needed to make it better darker skin tones.

## H    EXTENDED RELATED WORK

Self-supervised learning (SSL) is the most prominent paradigm for learning general representations from large unlabeled datasets, including methods like SimCLR (Chen et al., 2020), BYOL (Grill et al., 2020), and masked autoencoders (MAE) (He et al., 2022). Timeseries-specific objectives like TNC and TF-C have also shown promise (Tonekaboni et al., 2021; Zhang et al., 2022). SSL has gained traction in the domain of physiological signal analysis, with applications to health records (Chen et al., 2021; Yèche et al., 2021), fitness and personalization (Spathis et al., 2021), as well as brain (Cheng et al., 2020) and heart signals (Kiyasseh et al., 2021; Sarkar & Etemad, 2020).

However, despite the popularity of SSL, there are no widely used FMs for PPG data. While (Abbaspourazad et al., 2023) showcased the potential of foundation models for physiological signals, it was based on a single proprietary dataset and device (Apple Watch) while the models were not released, limiting its practical use in the research community. Similarly, REGLE's work (Yun et al., 2024) on the UK Biobank dataset showed that embedding PPG signals can improve genetic discovery and risk prediction outcomes. Although parts of that model and pipeline are public, the dataset is not openly accessible, and the primary goal was not to create a foundation model for PPG but rather to focus on genetics. Another work on the same data showed that PPG embeddings are promising for cardiovascular risk prediction (Weng et al., 2024). SiamQuality (Ding et al., 2024) also trained an unreleased model on 36 million PPG signals using proprietary data. Importantly, most of these works pre-trained on a single-device dataset and did not explore out-of-domain datasets or conduct transfer learning experiments, which are crucial for assessing the true generalizability of foundation

---

[7]https://biobank.ndph.ox.ac.uk/crystal/field.cgi?id=4205

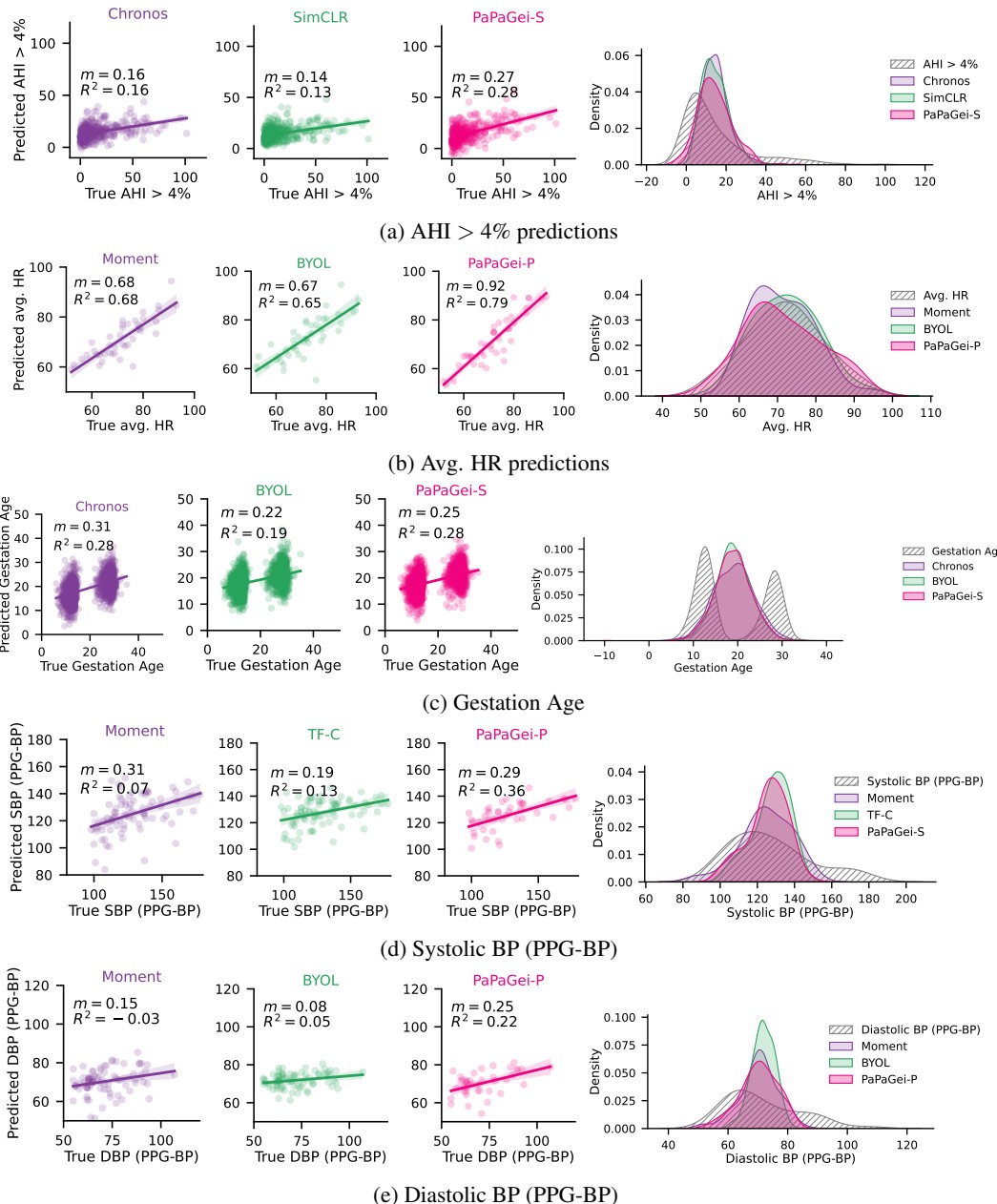

Figure 25: Regression plots and prediction distribution of different models compared to ground truth for (a) Apnea/Hypopnea Index $> 4\%$, (b) Average Heart Rate, (c) Gestation Age, (d) Systolic BP (PPG-BP), and (e) Diastolic BP (PPG-BP). $R^2$ is the coefficient of determination and $m$ is the correlation slope.

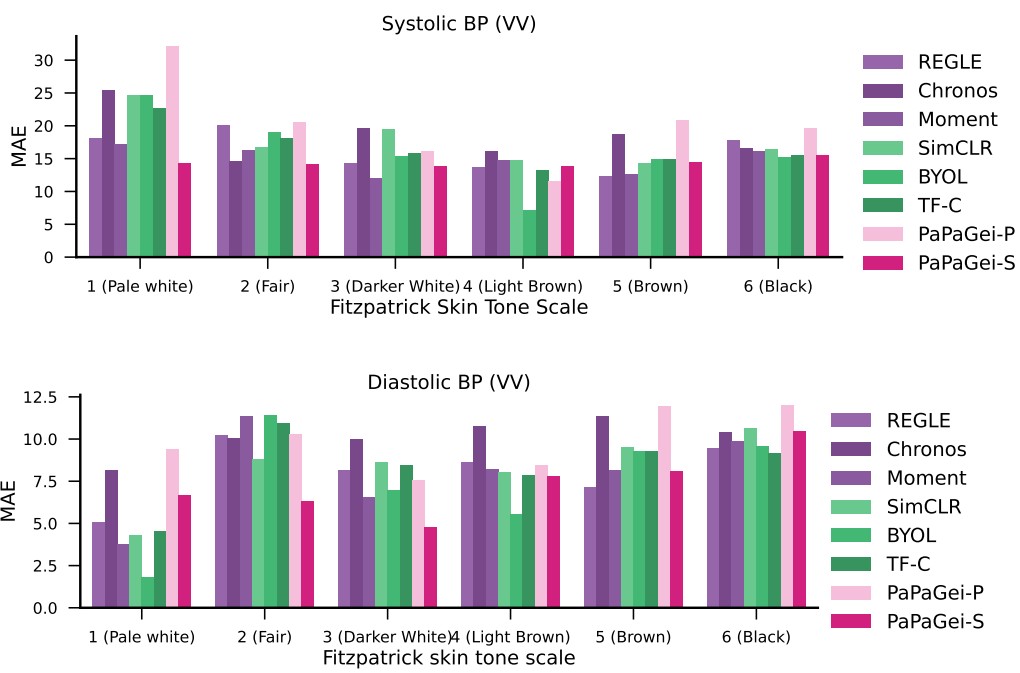

Figure 26: Detailed skin tone analysis for Blood Pressure estimation (VV dataset).

models. These studies highlight the potential of PPG-based foundation models but also underscore the need for openly available, pre-trained models that can be widely used and adapted by the research community.

On the other hand, generic time series foundation models have begun to gain popularity, mirroring the trend seen in Large Language Models (LLMs). These models are pre-trained on massive corpora of diverse time series data, aiming to learn universal representations that can be applied across various domains. For instance, Chronos (Ansari et al., 2024) was trained on an impressive 84 billion observations (analogous to tokens in NLP) from 28 distinct datasets. However, it's notable that this diverse collection does not include physiological data. Similarly, Moment (Goswami et al., 2024) was trained on billions of observations from a wide-ranging dataset that includes weather, traffic, energy, and other domains. While Moment does incorporate a small amount of ECG data, it comprises only a tiny percentage of the overall data pool.

In contrast to these generic approaches, our work takes a domain-specific focus. We curate a large pre-training and evaluation benchmark dedicated exclusively to PPG data. While knowledge gained from generic time series foundation models may transfer to domain-specific tasks like PPG, we expect the performance to be limited compared to a model trained specifically on PPG data. Furthermore, foundation models for ECG (McKeen et al., 2024; Song et al., 2024) or EEG (Yuan et al., 2024b) have shown promise but transferring from one domain-specific model (e.g., ECG) to another (PPG) is likely to be even more challenging, as the underlying signal characteristics can be quite different. For instance, Lai et al. (2023) trained a large-scale 12-lead ECG model for detecting 60 diagnostic terms, while McKeen et al. (2024) developed an open-source ECG FM using 1.6 million 12-lead signals. In brain signal analysis, Yuan et al. (2024b) introduced Brant-2, an EEG and SEEG model supporting tasks like sleep staging and seizure detection. Building on this progress, we adopt a domain-specific approach focused on photoplethysmography (PPG) signals. Building on the increasing interest in modality-specific foundation models, our specialized approach allows us to capture nuances and complexities specific to PPG signals.

An increasingly popular approach involves feeding timeseries data and prompts directly to Large Language Models (LLMs) (Gruver et al., 2024). However, despite promising results, LLMs struggle with high-dimensional signals due to their text-based processing (Spathis & Kawsar, 2024). A modality-specific encoder like PAPAGEI addresses this limitation by providing representations of

raw signals (Belyaeva et al., 2023), which can be combined with text and fed into more powerful multimodal foundation models, such as AnyMAL (Moon et al., 2023). This approach offers several advantages: computational efficiency through a fixed LLM, flexibility due to the modular design of encoder, adapter, and LLM components, and interoperability with other high-performing models (e.g., a state-of-the-art IMU encoder (Yuan et al., 2024a)). Crucially, this encoder-LLM approach does not require paired data with other modalities to train a single multimodal model. However, it may introduce complexity by limiting end-to-end gradient propagation and reduce interpretability in encoder-LLM communication compared to natural language prompts. Despite these trade-offs, PAPAGEI serves dual purposes: as a generic feature extractor for various PPG signals and applications, and as a modality encoder in next-generation frontier models. This versatility positions it as a valuable tool for advancing multimodal sensory AI systems.

## I    EXTENDED DISCUSSION

In §5.1, we observed that PAPAGEI outperforms baselines in at least 14 out of 20 tasks, with average classification and regression improvements of 4.7%-6.3% and 2.9%-4.9%, respectively. PAPAGEI-S performed best for cardiovascular parameters like BP, Hypertension, and Avg. HR, which are closely linked to metrics such as sVRI and IPA (Liang et al., 2018b). Additionally, PAPAGEI-P surpassed baselines FMs like Moment and is well-suited for tasks such as Smoking and Arousal.

By ablating different components of PAPAGEI-S (Section 5.2), we found that the full model performs best, with sVRI contributing the most. Adding IPA or SQI separately did not improve performance, suggesting that (a) IPA and SQI positively transfer in a multi-task setup, and (b) our design choice to include both to compensate for situations where IPA cannot be computed is effective (Section 3.2). While combining PAPAGEI-P and PAPAGEI-S may seem intuitive, constraining positive pairs on both sVRI and participants leads to too many unique labels with limited samples. In our scalability analysis, we observed that the smallest model (5M parameters) outperformed others, aligning with other studies using CNNs with 3.3M parameters for biosignals (Abbaspourazad et al., 2023), likely due to the size of PPG datasets. Larger models like Chronos or Moment are impractical for wearables due to their size and privacy concerns with cloud-based inference for health data. Additionally, PAPAGEI-S is more data-efficient for linear probing, showing greater performance gains with increased data availability, making it a promising backbone for small studies in future research.

Our studies in Section 5.3 reveal that PAPAGEI-S embeddings are more dispersed across participants, enhancing performance, while regression predictions more accurately reflect the true distribution. We attribute this to our positive pair selection, which chooses positive pairs across individuals based on sVRI. Moreover, our skin tone analysis shows that the method performs better on lighter skin tones, likely due to the model being trained predominantly on such data. For darker skin tones, performance was similar across models for diastolic BP, with REGLE and BYOL performing best, highlighting the need for future work creating more robust models for diverse skin tones.

To provide future direction regarding the use of PAPAGEI, we provide some suggestions. For instance, let's consider the nuMoM2B dataset which consists of pregnant women. PAPAGEI-S obtains an AUROC of 0.78 in pregnancy stage classification and 6.05 is gestation age classification. Compared to the pre-training population with diverse age and gender, the nuMoM2B consists of women generally aged between 20-35. Furthermore, the gestation age readings are collected approximately around the first and third trimester. Given these factors, the target nuMoM2B dataset has many variables contributing toward distribution shift. Therefore, PaPaGei-S can be fine-tuned to address the shift in the following ways: (1) We can align the pre-trained embeddings to the nuMoM2B embedding using unsupervised or semi-supervised domain adaptation. (2) Domain Generalization is also an option during the training phase to improve generalization robustness. (3) Newer methods such as LoRA can provide another way to quickly fine-tune. (4) Importantly, given that more women are present in the first visit compared to the third visit, we can optimize different metrics to improve accuracy under the imbalance. For example, AUPRC can be optimized instead of AUROC. Fairness of classification across genders can also be considered during training. Exploring these avenues to further enhance the performance and applicability of PaPaGei is a promising direction for future studies. Moreover, future work may benefit from exploring PPG specific augmentations such as GAN-based approaches (Kiyasseh et al., 2020); and systematically evaluating different augmentations to provide insights into useful PPG augmentations.

