# OpenReview forum: "PaPaGei: Open Foundation Models for Optical Physiological Signals"
_ICLR.cc/2025/Conference — ICLR 2025 Poster_

### Official Review · Reviewer_d898 · 2024-10-24

**Soundness:** 3
**Presentation:** 4
**Contribution:** 3
**Rating:** 8
**Confidence:** 5

**Summary:**

This paper reports on development and downstream testing of a pair of new foundation models (PaPaGei-P and PaPaGei-S) for photoplethysmography (PPG) signals in comparison to several pre-trained models and alternative pre-training methods.  All pre-training and downstream evaluation was accomplished using publicly-available PPG data sets, and the authors have shared their codebase to enable end-to-end reproducibility of the work reported in the paper.   In addition to the open codebase and use of public data sets for model development (which is itself new for published PPG foundation models) the novelty of this paper includes the incorporation of PPG-specific physiologically-informed loss functions during the pre-training phase.

The authors perform evaluation of the PPG encoder generality and utility using a battery of 20 downstream tasks, some of which have been published previously (for example blood pressure estimation and hypertension classification) and others of which are new (for example gestational age estimation).   They report performance comparison for their models against a pair recently published time series foundation models (Chronos and Moment), against an existing PPG-specific pre-trained model (REGLE), against three well-established self-supervised learning methods (SimCLR, BYOL, and TF-C), and against a baseline approach consisting of simple signal statistics fed into random forest models.

In addition to the performance comparison with other models and modeling approaches, the authors also report model size analysis, model sensitivity to skin tone, and systematic ablation analysis (evaluating the benefit of each source data set and PPG-specific loss function used in pre-training).

**Strengths:**

## This paper represents strong contributions in the following areas:
### Originality:
Although the idea of foundation models is not new, there have been few published examples of foundation model development and evaluation for physiological signals such as PPG.  Here, in addition to comparing the performance of multiple pre-trained time series foundation models, as well as adapting different contrastive and momentum-based SSL methods, the authors have also incorporated some PPG-specific elements into the pre-training (loss functions for SQI, sVRI and IPA) that have not been reported elsewhere.

**Furthermore the authors should be commended for two important things which, combined, have not been published previously:**
1. They performed all training and downstream evaluation using data sets that are publicly available, enabling straightforward reproducibility and verification of their results (as well as use of their models by other researchers for future work).
2. They shared their codebase directly in the submission, which is organized and sufficiently well-commented to understand in a reasonable amount of time.

### Clarity
The paper is well written, with clear descriptions and explanations in several key areas:
1. Data set provenance (Appendix B is very helpful), and the purpose(s) for which each data set was used;
2. Data pre-processing;
3. High-level strategy and technical details of the pre-training method and loss/objective functions;
4. Downstream task evaluation and performance reporting (the subset of cases that I checked were straightforward to evaluate and understand);
5. Analysis that was performed in addition to performance reporting on downstream tasks, such as dispersion of subject embeddings and sensitivity of some tasks to skin tone.

### Significance
1. The exploration of PPG-specific dimensions of the SSL training is valuable and important, as well as the authors’ comparison of their model with off-the-shelf time series foundation models and established (but not PPG-specific) self supervision methods.  This has the potential to impact work in other areas at the intersection of machine learning and physiological sensing.
2. PPG foundation models, in particular open/reproducible foundation models, have the potential for high impact in both health research and practical applications.

### Quality
Claims and conclusions were generally well-reasoned and justified, with some key exceptions (discussed in the following section).

**Weaknesses:**

# Brief Summary #
This paper had some important weaknesses that are listed briefly below, with more detailed comments and some suggested steps to address these weaknesses following the list:
1. Absence of downstream evaluation on basic demographic targets such as age, height/weight, BMI or sex.
2. [Most concerning] Inappropriate choice of baseline model (RF using basic signal statistics) for performance comparison with their models.
3. In the Case Studies section discussing the downstream AHI regression performance, the authors do not provide good evidence indicating that the model captures the distribution tails well.



# Additional Discussion Details #

## Issue 1: Absence of downstream evaluation on basic demographic targets
PPG is known from past publications (including at least one cited by the authors) to capture information highly predictive of age, body mass index (BMI) and sex.  These demographic factors are also highly predictive of additional health targets, indicating that a data-driven model may use demographic features as prediction ‘shortcuts’, rather than utilizing features that are specific to the health target of interest.  Performance of the authors’ models (and other models) in predicting demographic targets can give an indication of how well the models capture information relevant to these tasks, as well as provide some clues as to whether the models may be using demographic-related features as ‘shortcuts’ for other downstream tasks.

### Suggestions to address Issue 1:
Several of the data sets used for downstream evaluation have sufficient demographic metadata and subject numbers to train and test downstream tasks for age classification/regression, BMI classification/regression, and sex classification (even after reserving ~20% of data for testing).  Using the appropriate data sets (for example just MIMIC-III, MESA and VitalDB) the authors should compare performance from their PPG models against the other models (Chronos, REGLE, SimCLR, BYOL, etc).   To avoid any issues associated with splitting the pre-training data into train/val/test, this analysis could also be accomplished in combination with the pre-training data ablation experiments (e.g. reporting demographics prediction results only for one of the three large data sets).


## Issue 2 [Most concerning]: Inappropriate choice of baseline model

The choice of baseline model consisting of RF with PPG signal statistics represents an inappropriately weak baseline.  In a few ways this approach is even weaker in practice than as stated in the paper (for example the signals are z-scored in pre-processing, meaning that the signal mean is guaranteed zero for all segments and therefore provides no information to the RF model).  Additionally, the section describing the additional PaPaGei-S objectives highlights the physiological relevance of the IPA, SVRI, and SQI (and the authors seem to have all the code/tools necessary to perform these calculations across a large number of PPG segments), which would provide a simple strategy to improve the statistics-driven baseline model.

Lastly, as mentioned for the previous item (Issue 1) PPG is known to capture information highly predictive of age, body mass index (BMI) and sex, with the corresponding features then used by downstream models as ‘shortcuts’ for predicting a variety of targets that have demographic dependence.  It would be highly informative to include a simple baseline model that predicts each target value/class using the demographic factors as inputs (the model could be LR, or Ridge, depending on the target task).

The reason that this incorporation of baseline models with demographic inputs (producing a ‘stronger’ baseline for comparison) is important is that it can help **reveal whether the PaPaGei models truly represent a meaningful breakthrough** or are simply equivalent to (or even worse than) much simpler established approaches.


### My Findings regarding demographic (and constant/mean) predictor performance on a subset of the public PPG datasets:

I appreciate that the authors utilized public and (in several cases) conveniently-accessible data sets for their downstream evaluation. This made it possible for me to spot check the performance of some simple baseline models on several of the tasks.  Due to time limitations I could not do this analysis for every task, but am reporting observations for tasks associated with the data sets PPG-BP (Liang, etc al. 2018), and ECSMP (Gao et al., 2021), which covers the following downstream tasks: SBP (R), DBP (R), Avg HR (R), Hypertension (C), Mood Disturbance (C).  For these tasks I was able to achieve the following performance using either a constant mean predictor (only for DBP regression task) or a demographics-based predictor incorporating Age, BMI, sex (where available):

1. **DBP** Estimation (regression): I observed **MAE = 8.71** (identical to PaPaGei-S performance, **better than PaPaGei-P performance)** using a constant mean predictor; I observed **MAE = 8.07 (better than all models)** using a demographics-only predictor (OLS fit with age, sex, BMI up to quadratic terms).
2. **SBP** Estimation (regression): I observed **MAE = 13.56 (better than evaluated models including PaPaGei)** using a demographics-only predictor (OLS fit with age, sex, BMI up to quadratic terms), and **MAE = 12.48** using demographics + heart rate (OLS fit)
3. **Avg HR** Estimation (regression): I observed **MAE =8.11 (much worse than any reported model)** with a demographics-only predictor.
4. **Hypertension** (classification):  Using the same class grouping reported by the authors, I observed **ROC AUC = 0.78 (better than all evaluated models including PaPaGei)** with a demographics-only LR predictor (age, sex, BMI, HR up to quadratic terms)
5. **Mood Disorder** (classification):  Using the same class grouping reported by the authors, I observed **ROC AUC = 63 (better than all evaluated models including PaPaGei)** with a demographics-only LR predictor (age and sex up to quadratic terms)

I did not personally check any other data sets or tasks due to time limitations.  Despite the caveat that for the results above I did not split the data into train/test sets, for 4 our of 5 cases the performance from these much simpler models (at most 15 parameters) was better than the models having millions of parameters.  Combined with the knowledge that PPG captures significant demographic information, this finding is strongly suggestive that the models may simply be using demographic-related features as ‘shortcuts’ for the target task.


### Suggestions to address Issue 2:
1. At a minimum, the RF baseline model should incorporate additional easy-to-derive signal features such as IPA, SVRI, and SQI (other standard pulse waveform metrics are also reasonable to include if time permits— see  Elgendi, M., Liang, Y., & Ward, R. (2018). Toward Generating More Diagnostic Features from Photoplethysmogram Waveforms. Diseases, 6(1).)   This would provide a more effective baseline model without adding any learned feature extraction.
2. In place of (or in addition to 1 above), it is strongly recommended that the authors also compare with a baseline model consisting of demographic factors as inputs (the model could be LR, Ridge or OLS depending on the target task).  Including heart rate as a predictor in addition to demographic variables would also be a reasonable comparison, since all models appear to be fairly good at estimating heart rate from PPG.


## Issue 3:  Poor evidence for regression models capturing the distribution tails.

I disagree with the authors that the analysis and plots in Figure 8 (particularly 8a) provide evidence that the predicted values match the distribution of the true values.  For the plots in Figure 8a the regression slope is clearly much less than 1.0, indicating that the models tend to over-predict for small True AHI data points and tend to under-predict for large True AHI values (i.e. significant model bias present)


### Suggestions to address Issue 3:

1. For the plots in Figure 8a, use the same range for the x- and y-axes.  Overlay the true and predicted AHI distributions on a common axis (can be a separate plot), which will make it more clear that the distributions are not well matched.
2. Report the regression slope in addition to the R^2 value.  The slope is likely to be much less than 1.0 for all models (although likely highest for PaPaGei), indicative of model bias.
3. As an alternative (or in addition) to the truth-vs-predicted scatterplots, also include a Bland-Altman plot. Bland-Altman is a straightforward way to illustrate whether a prediction model is biased.
4. Repeat the same Bland-Altman plots for the other regression tasks (gestational age and BP).

**Questions:**

The paper is generally very well-written (in terms of both clarity and technical content) and overall I enjoyed reading it. I have a few additional comments/questions regarding style and content:

1. Reporting the average MAE across tasks in Tables 3 and 4 (and in Figure 4) may not be the best way to summarize the regression performance, since the regression targets have several different units (e.g. mmHg, bpm, and weeks).   To provide a ‘fair’ summary across all the regression tasks with different units, it would make more sense to use a unitless metric such as mean absolute percent error or (possibly) the mean F-statistic across all tasks.
2. In Figure 5a/b and 6, it is not obvious to the reader whether these differences are statistically significant.  Were any tests for significance performed?  If so, it would be helpful to report this briefly (simply using the name of the test and stating the significance level).
3. Expanding on Figure 5a/b (component ablation) it would be informative to see a summary of the performance changes across all downstream tasks, which may provide some insight into which PPG-specific pre-training losses are influential for certain tasks and not others.  This could be an additional appendix table (it doesn’t need to fit into the main text).
4. Figure 5c shows that, on average, the middle-sized (35M) parameter model has the worst performance while the smallest (5M) parameter model has the best performance.  Were models smaller than 5M evaluated?  Including performance for models smaller than 5M (perhaps going as small as 0.07M , the size of REGLE) would give an indication of whether 5M truly represents the optimal model size.  This would also make the overall trend more clear (a non-monotonic trend for only 3 model sizes is hard to trust).
5. I like the analysis of inter-participant embeddings (section 5.3 and Figure 7) since this is agonistic to subject-level tasks.  However, looking at Figure 7 it feels “wrong” that PaPaGei-S has some non-negligible density for distance <0.  Is this an artifact of distribution smoothing?   As an alternative to the pairwise distance distribution, it could also be informative to evaluate other measures of unsupervised embedding quality such as Rankme (or similar).
6. I’m not sure if I understand some of the language used in the Skin Tone Analysis section, specifically the phrase “however, there are no significant differences across all models” in reference to Figure 9.    Does “no significant differences across all models” refer to a comparison of light skin tones vs. dark skin tones (grouping models), or does it refer to model-to-model differences for dark skin tones vs. light skin tones, or to something else?  It would help if this can be stated with more precise language.
7. For Figure 9, it would also be informative to plot the confidence intervals as whiskers on the bar plot.  This would make it more clear (visually) whether the model-to-model differences are significant.

---

> ### Author Response · Authors · 2024-11-21
> **Response to R-d898 (W1 & W2)**
>
> Thank you for the valuable suggestions to improve our baseline. Following your recommendations, we evaluated hold-out test results using demographic features (age, sex) and PPG features (sVRI, IPA, SQI), for both regression (ridge) and classification (logistic regression) tasks (**Appendix E, Table 15**; Link to table: https://imgur.com/Rz4855N):
>
> - **Ablation Study**: We compared PaPaGei with three baselines—demographics alone, PPG features alone, and demographics + PPG features. Our results show that while demographics is a stronger baseline than statistical features, PaPaGei outperforms the demographics + PPG baseline in 14 out of 18 tasks.
>
> - **Effect of Demographics**: We trained a model combining PaPaGei-S with demographics. The results indicate that incorporating demographic features with PaPaGei-S creates a stronger model than using PaPaGei-S alone.
>
> These findings underscore an important point: demographic features are not competing with PaPaGei but rather complement it, as previously established in studies including demographics with sensor data [1]. This highlights the complementary potential of combining PaPaGei's advanced feature extraction with demographic context for improved task performance.
>
> **Ablation study using Demographics, PPG features, Demographics + PPG.**
> Comparing PaPaGei performance improvements with supervised baselines for each task: Classification Tasks (Positive is better): ICU (+0.13), Mortality (+0.01), Smoker (-0.02), Pregnancy Stage (+0.22), Hypertension (0.00), SDB (no demographics), Mood Disturbance (-0.08), Valence (-0.01), Arousal (+0.04). Regression Tasks (Negative is better): AHI > 3% (-1.43), AHI > 4% (-1.61), gestation age (-1.54), SBP-VV (-0.31), DBP-VV (-0.46), SBP (-0.11), DBP (-0.65), Avg. HR (-4.07), HR (-2.93). PaPaGei-S performs better for real-time sleep and cardiovascular outcomes such as sleep apnea, heart rate and blood pressure, respectively. In particular, we notice that outcomes such as heart rate benefit substantially from PPG rather than demographics. Demographics are useful in tasks without real-time dependence such as smoking, which is established to be associated with age and sex [2]. Importantly, demographics do not add much to already homogeneous populations. For example, consider the NuMoM2B dataset which has women within a specific age range. Here, we observe that PaPaGei obtains much higher AUROC and MAE than the supervised baselines.
>
> **Demographics complement PaPaGei-S embeddings.**
> We observe that adding demographics to PaPaGei-S embeddings improves over PaPaGei in the following tasks: Mortality (+0.03), Hypertension (+0.03), AHI > 3% (-0.62), AHI > 4% (-0.09), gestation age (-0.03), SBP VV (-0.38), DBP VV (-0.03), SBP (0.40), DBP (0.10). Based on these results, PaPaGei-S + Demo is a stronger model in many cases. Importantly, these results indicate that PaPaGei-S embeddings learn features that are complementary to demographics and are not simply proxies for age or sex. However, it is important to note that while demographic features can be valuable for personalization, they may not always be readily available, and in reality, we cannot use them in isolation to predict real-time outcomes such as blood pressure or heart rate. Therefore, our PaPaGei models are designed to function effectively with real-time sensor data alone, ensuring their applicability in situations where complete demographic information is not accessible.
>
> [1] Spathis, D., Perez-Pozuelo, I., Gonzales, T. I., Wu, Y., Brage, S., Wareham, N., & Mascolo, C. (2022). Longitudinal cardio-respiratory fitness prediction through wearables in free-living environments. NPJ Digital Medicine, 5(1), 176.
>
> [2] Chung, W. S., Kung, P. T., Chang, H. Y., & Tsai, W. C. (2020). Demographics and medical disorders associated with smoking: a population-based study. BMC Public Health, 20, 1-8.

---

> ### Author Response · Authors · 2024-11-21
> **Response to R-d898 (W3)**
>
> Thank you for the suggestion to improve the regression plots. To address Issue 3, we have made the plots more informative in the following ways:
> - **Scatterplot of True vs. Predicted Values**: We now include a scatterplot of true versus predicted regression values, annotated with the slope of the regression line (m) and the R^2 value. These metrics provide a clear indication of the model's predictive performance.
> - **Kernel Density Plot**: We overlay the true distribution with the predicted model distributions using kernel density plots. This visualization demonstrates how well PaPaGei captures the true distribution.
>
> Updated figures: (1) AHI: https://imgur.com/IkR7XYd; (2) Avg. HR: https://imgur.com/ru8ZjhM
>
> These additions make it easier to assess the alignment between predictions and true values. It is important to note that steeper slopes and higher $R^2$ values indicate better predictions, as the scatterplot illustrates the alignment between true and predicted values. As shown in the updated figures (**Figure 8, Appendix F - Figure 24**), PaPaGei-S exhibits steeper slopes and higher R-squared values for both AHI and HR prediction, indicating superior performance. In the AHI plot, PaPaGei-S demonstrates a closer match to the true distribution, particularly in the peak around 22 and the spread of the base. Similarly, for average HR, PaPaGei-P aligns more closely with the left tail of the true distribution (around the 42 to 50 range).

---

> ### Author Response · Authors · 2024-11-21
> **Response to R-d898 (Questions)**
>
> **(Q1)**
>
> To summarize the regression results, we have provided the symmetric mean absolute percentage error (sMAPE) [1] in addition to the average MAE (**Tables 3 & 4**). We choose sMAPE instead of MAPE because sleep apnea has true values close to 0, resulting in extremely large MAPE values. However, sMAPE performs weighting and scales the values between 0 to 100%. Our results indicate that PaPaGei-S obtained an sMAPE of 13.34% whereas the next best-performing method is Moment with an error of 13.82%. It is important to note that sMAPE applies weighting and scales errors. When unweighted error percentages are considered, PaPaGei-S demonstrates larger performance gains over the baseline methods, highlighting its superior accuracy.
>
> [1] https://permetrics.readthedocs.io/en/latest/pages/regression/SMAPE.html
>
> **(Q2)**
>
> Updated plots: https://imgur.com/pF4Kc4v
>
> We performed the Wilcoxon Signed Rank test (non-parametric paired t-test) to evaluate statistical significance at 90% and 95% confidence. The significance values are now indicated in **Figures 6 (a)-(b), and 5**. In Figure 6, we notice that the Full model is statistically significant ($p<0.05$) compared to both sVRI + SQI and sVRI + IPA. We also notice that sVRI is the most important component with performance matching the Full model. However, we also notice that sVRI is not significant compared to sVRI + SQI, thus indicating the need for the Full model.
>
> In Figure 5, we notice statistically significant performance gains as we increase data for MAE. We observe a similar trend for AUROC except that VitalDB + MIMIC-III is not significant compared to All the datasets.
>
> **(Q3)**
>
> This is a good idea to obtain insights about the link between PPG-specific losses and downstream tasks. We have added the results to **Table 14**.
>
> **(Q4)**
>
> Unfortunately, we did not evaluate models smaller than 5M. We hypothesised that scaling model size would result in some monotonic trend. However, we obtained a non-monotonic trend with the smaller and larger models performing better than the middle. To address this concern, we have clarified our claim about 5M being the optimal model size for PPG data in general.
>
> “This suggests the 5M model is better suited for our pre-training datasets, aligning with prior findings on the proportionality between data and model size [1]” “Nevertheless, our scaling analysis shows a non-monotonic trend, indicating other factors strongly influence performance.”
>
> [1] Narayanswamy, G., Liu, X., Ayush, K., Yang, Y., Xu, X., Liao, S., ... & McDuff, D. (2024). Scaling Wearable Foundation Models. arXiv preprint arXiv:2410.13638.
>
> **(Q5)**
>
> Yes, this is a visual artefact of smoothing and the way the KDE histogram is plotted. The smallest distance is 0.13.
>
> **(Q6)**
>
> Here, we refer to model-to-model differences. We notice clearly that PaPaGei-S performs better for light tones than other models for SBP and DBP. However, for darker skin tones it is difficult to identify one model that is the best. We have rephrased the sentence to better reflect our observation.
> “PaPaGei-S achieves the best BP estimation across light tones. Across dark tones, we notice that BYOL and REGLE obtain the lowest MAE for Systolic BP and Diastolic BP. However, identifying a single model that performs best across all skin tones remains challenging. While PaPaGei-S obtains the best overall performance, additional work is necessary to improve robustness on darker skin tones.
>
> **(Q7)**
>
> Thank you for the comment, we have added whiskers to Figure 9.

---

> ### Author Response · Authors · 2024-11-25
> **Following up with R-d898**
>
> Hi R-d898,
>
> We hope this message finds you well. This is just a follow up on our response to your review. We have carefully addressed your comments and made revisions to the manuscript accordingly. We would be grateful if you could acknowledge our rebuttal and consider raising your score if you are satisfied with our revisions.
>
> Thank you for your time and consideration.

---

> ### Author Response · Authors · 2024-12-01
> **A gentle reminder**
>
> Dear Reviewer d898,
>
> Thank you for taking the time to review our work. We sincerely appreciate your valuable feedback and have revised the paper to incorporate your suggestions. Given the approaching deadline, we would be grateful if you could acknowledge our revisions.

---

> > ### Comment · Reviewer_d898 · 2024-12-02
> >
> > I thank the authors for the modifications and updates to their paper, for responding to my questions concerns (as well as the concerns/questions provided by the other reviewers), and for their thorough and comprehensive summary of the experiments, analysis and edits that were done to address each point.
> >
> > I've re-read the paper with a focus on the new material, and they have addressed the majority of my concerns. The paper is definitely stronger now, and I've increased my overall rating.

---

### Official Review · Reviewer_XN3M · 2024-10-28

**Soundness:** 3
**Presentation:** 3
**Contribution:** 2
**Rating:** 8
**Confidence:** 4

**Summary:**

This paper provides an open foundation model (open code, data, model) for a physiological signal, PPG. The authors have pre-trained a model on a combination of 3 open-source PPG dataset, and evaluated on various datasets for various downstream targets. The comparisons with some off-the-shelf time-series models, and pre-training algorithms have been presented, and several ablations studies regarding different components have been done.

**Strengths:**

* While pre-training foundation models for PPG has been done before with closed-source models/datasets some with open-source and some with closed-source code, this work presents the first open-source foundation model for PPG, trained on open datasets, with open-source code, that can foster the use of PPG for health applications and for health research.
* The paper, in my opinion, is well-written. I enjoyed reading the paper, the comparisons are well-presented and the paper is easy to follow. I would like to thank the authors for their efforts in the quality of the analyses and presentations in the figures.
* I really appreciate the level of details in the paper including the ablations studies; authors did a good job for providing good level of information and detailed ablations for a curious reader.

**Weaknesses:**

* One of my main comments is regarding the novelty of the presented work, the language in the paper with respect to the prior work, and lack of comparison with the prior work in terms of performance. It appears to me that the idea in this paper resembles significant similarity to a prior work [1] (which is cited in this work multiple times) in terms of: 1) training a PPG foundation model on a large-scale dataset, 2) methodology in terms of loss, positive pairs, random augmentations, 3) evaluation across multiple targets, and demonstrating that one PPG foundation model is strongly predictive of a wide array of downstream health conditions. The main distinction of this work is openness of dataset, code and model. I do appreciate the openness aspect, however, the language in the paper sadly does not properly reflect the relationship to the prior work. For example, the only mention is in the introduction/related work is “Recent studies have shown promise in this area, as (Abbaspourazad et al., 2023) demonstrated that embeddings from ECG and PPG signals can generalize across multiple health-related tasks using proprietary Apple Watch data.”, which does not reflect the proper relationship between the two papers and the 3 points stated above. I highly recommend authors to reflect the appropriate relationship upfront for fairness:
    * It is also surprising to me that throughout the dataset/results, there’s no comparison table to the prior work [1] in terms of performance and choices of PaPaGei. Given the direct resemblance of the 2 works, my suggestion to the authors are:
         * Putting a table/discussion in the methods/datasets for comparison of dataset used in this study with respect to the prior work for PPG foundation models [1] in terms of number of subjects, amount of data, openness vs. closedness, number (and kinds) of different devices and etc. In addition, comparison of the (only major) training choices, size of models, compression ratios (how many channels, how many seconds of input compressed to what embedding size).
         * Putting a comparison results evaluation table/discussion for major downstream targets in comparison to reported numbers in [1], at the very least for demographics comparison such as age/sex/body mass index, whose evaluations are missing in the current manuscript (they are important!), and has not been compared to the closed work [1]. This will demonstrate the effectiveness of PaPaGei training/model, and provide the performance gap (if any) with closed source models (authors can note that evaluation datasets are different).
    * Also, the language with respect to another prior work [2] can be enhanced. Pre-training PPG models where positive pairs are with respect to signal quality was done prior to this work but there was no direct mention of this in the paper [2]. Moreover, this prior work has open-source code  (closed-source dataset) which was again not mentioned properly. I suggest authors consider citing these prior work properly.
    * In addition, the idea of pre-training foundation models on other physiological signals on large datasets in the related work section is missing. Papers such as [3] for ECG and [4] for IMU (just examples) should be added.

* One of my main concerns about the reported evaluations is the significance of the reported metrics in downstream comparisons. To provide an example, in Table 3, authors provide point estimates and confidence bounds, but for a lot of the targets, the confidence bounds are very wide and overlapping across methods. For instance, for “smoker”, it is not clear to me whether PaPaGei-P is significantly better than Chronos. My suggestion to the authors are:
    * This begs the question about the effectiveness of PaPaGei, given that an off-the-shelf time-series model that was not trained on PPG (Chronos), can capture similar amount of information regarding an important vascular target (smoking), therefore providing discussion would be valuable for the readers.
    *  Providing P-values and n for significance comparisons is essential. Similar comment for other comparisons such as Table 4, Figures 4, 5 and 6.
* In terms of comparison with SimCLR and BYOL, the authors can do a better job explaining the training choices and implementation, and demonstrate fair comparisons. They say “It is noteworthy that both BYOL and TF-C require multiple encoders and different projection heads, resulting in variations in model sizes. For these methods, we use existing implementations available online, but apply our encoder as the backbone to ensure consistency.”. I appreciate that they used similar encoder for fair comparisons, but other implementation details is not clear to be identical. My suggestions/questions to be clarified in the paper are:
    * What details are the same and what details are different? (for example positive pairs, augmentations, batch size, learning rate, and any other useful details to demonstrate fair comparisons).
    * Also kinda related to this, there’s no mention of how and what temperature was selected.


[1] Abbaspourazad, S., Elachqar, O., Miller, A. C., Emrani, S., Nallasamy, U., & Shapiro, I. (2023). Large-scale training of foundation models for wearable biosignals. arXiv preprint arXiv:2312.05409.

[2] Ding, C., Guo, Z., Chen, Z., Lee, R. J., Rudin, C., & Hu, X. (2024). SiamQuality: a ConvNet-based foundation model for photoplethysmography signals. Physiological Measurement, 45(8), 085004.

[3] Lai, J., Tan, H., Wang, J., Ji, L., Guo, J., Han, B., ... & Yang, W. (2023). Practical intelligent diagnostic algorithm for wearable 12-lead ECG via self-supervised learning on large-scale dataset. Nature Communications, 14(1), 3741.

[4] Yuan, Hang, et al. "Self-supervised learning for human activity recognition using 700,000 person-days of wearable data." NPJ digital medicine 7.1 (2024): 91.

**Questions:**

* Do all datasets contain single-channel PPG? If yes, it is worth mentioning in the paper.
* It is interesting that combining three pre-training datasets, does not provide much additional information regarding downstream targets. What’s the authors’ hypothesis? It is worth adding a discussion to the paper
* I am not sure about this statement: “Interestingly, in single datasets, MESA performs the best, despite having the fewest participants but the highest number of segments, echoing similar results in the language model domain (Dubey et al., 2024).”. Do the authors have a justification for why they think this is relevant to Llama3? If yes, please add to the paper, and if not, please consider changing the language.
* What’s the authors hypothesis for PaPaGei-S being better than PaPaGei-P in general? I find it very surprising that defining the positive pairs based on quality of the segments (and not subjects) improve the performance. In fact, one could argue PaPaGei-S should have a slightly worse performance, because it does not leverage pairing good quality and bad quality segments. Did the authors investigate the performance of a few downstream targets, w.r.t the change in segment quality in PaPaGei-P vs. PaPaGei-S?

---

> ### Author Response · Authors · 2024-11-21
> **Response to R-XN3M (W1)**
>
> **(W 1.1)**
>
> Thank you for your suggestions to better represent prior work. Following your suggestions, we discuss [1] in terms of large-scale training, method, and evaluation. In particular, we discuss specifics about their contributions and also indicate how we differ from their work (**Section 2**): “[1] demonstrated that embeddings derived from PPG signals can predict over 45 diverse downstream health-related tasks using proprietary Apple Watch data. Their approach uses an SSL framework based on patient-level positive pair contrastive learning.”
> “In contrast to [1], our work exclusively uses public datasets for large-scale PPG training and extends the SSL framework to incorporate PPG morphology. While their evaluation is limited to a single proprietary dataset, we validate our approach on 10 diverse downstream datasets, showcasing greater generalizability and robustness across varied real-world scenarios.”
>
> Importantly, we have added a new **Table 17** to highlight the key similarities and differences between PPG studies with regards to: the number of users, number of devices (& type of devices), openness, and number of downstream tasks and datasets. We hope that these additions address your concerns.
>
> Moreover, we have added model design comparisons to other studies throughout the paper: ”Segment the signal into 10-second windows (whereas larger studies use 30s [2] and 60s [1])” (**Section 4.1**) “We adopt a ResNet-style CNN encoder, following [2], whereas [1] utilize an EfficientNet-style.”  “PaPaGei-P and PaPaGei-S having 5M and 5.7M parameters, respectively, other studies use model sizes of 3.3M [1]” (**Section 4.1**)
> “In contrast to a smaller embedding size of 256 adopted by [1], we project the learned representations to a 512-dimensional embedding after the convolutional block (we investigated larger embedding sizes of 768 and 1024 and found no significant performance changes).” (**Appendix A**)
>
> We have comprehensively evaluated demographic characteristics in the newly added **Appendix E**. In particular, we have evaluated PaPaGei’s ability to predict demographic targets through age regression, age classification, and sex classification. We compare these results to the [1] in **Table 16, in Appendix E**. Furthermore, we comment on this in the main text in Section 6 as follows: “PaPaGei-S achieves 7.78 MAE in age regression, 0.85 accuracy in age classification, and 0.79 accuracy in sex classification, advancing open-source efforts despite trailing larger closed studies [1] by 2.18, 0.05, and 0.13, respectively.”
>
> **(W 1.2)**
>
> We directly acknowledge [2] in the methods section: “Prior studies have shown that incorporating PPG signal quality during training yields positive results.” Furthermore, we also discuss key similarities and differences between this study and in new **Table 17**.
>
> **(W 1.3)**
>
> We have added a discussion on foundation models for physiological signals in the extended related work (**Appendix H**): “Recently, there is growing interest in modality-specific FMs tailored to physiological signals [3] and human activity [4]. For instance,[3] trained a large-scale 12-lead ECG model for detecting 60 diagnostic terms, while [6] developed an open-source ECG FM using 1.6 million 12-lead signals. In brain signal analysis, [5] introduced Brant-2, an EEG and SEEG model supporting tasks like sleep staging and seizure detection. Building on this progress, we adopt a domain-specific approach focused on photoplethysmography (PPG) signals.”
>
> We also acknowledge the link between time series methods and LLMs in Appendix I.
>
> “An increasingly popular approach involves feeding timeseries data and prompts directly to Large Language Models (LLMs). However, despite promising results, LLMs struggle with high-dimensional signals due to their text-based processing [7]. A modality-specific encoder like PaPaGei addresses this limitation by providing representations of raw signals, which can be combined with text and fed into more powerful multimodal foundation models, such as AnyMAL. This approach offers several advantages: computational efficiency through a fixed LLM, flexibility due to the modular design of encoder, adapter, and LLM components, and interoperability with other high-performing models (e.g., a state-of-the-art IMU encoder [5]. Crucially, this encoder-LLM approach does not require paired data with other modalities to train a single multimodal model. However, it may introduce complexity by limiting end-to-end gradient propagation and reduce interpretability in encoder-LLM communication compared to natural language prompts. Despite these trade-offs, PaPaGei serves dual purposes: as a generic feature extractor for various PPG signals and applications, and as a modality encoder in next-generation frontier models. This versatility positions it as a valuable tool for advancing multimodal sensory AI systems.”

---

> > ### Author Response · Authors · 2024-11-21
> >
> > [1] Abbaspourazad, S., Elachqar, O., Miller, A. C., Emrani, S., Nallasamy, U., & Shapiro, I. (2023). Large-scale training of foundation models for wearable biosignals. arXiv preprint arXiv:2312.05409.
> >
> > [2] Ding, C., Guo, Z., Chen, Z., Lee, R. J., Rudin, C., & Hu, X. (2024). SiamQuality: a ConvNet-based foundation model for photoplethysmography signals. Physiological Measurement, 45(8), 085004.
> >
> > [3] Lai, J., Tan, H., Wang, J., Ji, L., Guo, J., Han, B., ... & Yang, W. (2023). Practical intelligent diagnostic algorithm for wearable 12-lead ECG via self-supervised learning on large-scale dataset. Nature Communications, 14(1), 3741.
> >
> > [4] Yuan, Hang, et al. "Self-supervised learning for human activity recognition using 700,000 person-days of wearable data." NPJ digital medicine 7.1 (2024): 91.
> >
> > [5] Yuan, Z., Zhang, D., Chen, J., Gu, G., & Yang, Y. (2024). Brant-2: Foundation Model for Brain Signals. arXiv preprint arXiv:2402.10251.
> >
> > [6] McKeen, K., Oliva, L., Masood, S., Toma, A., Rubin, B., & Wang, B. (2024). Ecg-fm: An open electrocardiogram foundation model. arXiv preprint arXiv:2408.05178.
> >
> > [7] Spathis, D., & Kawsar, F. (2024). The first step is the hardest: Pitfalls of representing and tokenizing temporal data for large language models. Journal of the American Medical Informatics Association, 31(9), 2151-2158
> >
> > [8] Moon, S., Madotto, A., Lin, Z., Nagarajan, T., Smith, M., Jain, S., ... & Kumar, A. (2024, November). Anymal: An efficient and scalable any-modality augmented language model. In Proceedings of the 2024 Conference on Empirical Methods in Natural Language Processing: Industry Track (pp. 1314-1332).

---

> ### Author Response · Authors · 2024-11-21
> **Response to R-XN3M (W2)**
>
> Thank you for highlighting this concern. We plan to discuss it in the following ways:
> We acknowledge the capability of general-purpose time-series models to capture information in certain downstream tasks. Indeed, models like Chronos perform comparably well for targets such as mortality and smoking. We have added the following to **Section 5.1**: “Chronos obtains good performance in predicting mortality, pregnancy stage, and smoking, likely due to their slower rate of change and reduced reliance on granular PPG-specific features. General-purpose models suffice for these broader characteristics. However, tasks requiring finer PPG-specific granularity, such as heart rate prediction, blood pressure estimation, or sleep apnea, benefit from PaPaGei's specialized feature extraction.”
>
> To evaluate the significance across models on a per-task basis, we do the following. First, we performed the Friedmann Chi-Square test and identified statistically significant differences across PaPaGei and the baseline models at p < 0.05. Next, we created critical difference (CD) diagrams to rank the best-performing models, as suggested by the literature to compare models over multiple datasets [1, 2]. The CD diagrams (**Figures 22 & 23**) are available here: https://imgur.com/cPS1yBG and https://imgur.com/k4lvydU. The CDs indicate that PaPaGei performs the best across both classification and regression tasks. Furthermore, it has a statistically significant average rank as indicated by the lack of a horizontal line.
>
> We performed the Wilcoxon Signed Rank test (non-parametric paired t-test) to evaluate statistical significance at 90% and 95% confidence. The significance values are now indicated in **Figures 5 and 6** (https://imgur.com/pF4Kc4v). In Figure 6, we notice that the Full model is statistically significant ($p<0.05$) compared to both sVRI + SQI and sVRI + IPA. We also notice that sVRI is the most important component with performance matching the Full model. However, we also notice that sVRI is not significant compared to sVRI + SQI, thus indicating the need for the Full model. In Figure 5, we notice statistically significant performance gains as we increase data for MAE. We observe a similar trend for AUROC except for the case of VitalDB + MIMIC-III which is not significant compared to All the datasets.
>
> [1] Demšar, J. (2006). Statistical comparisons of classifiers over multiple data sets. The Journal of Machine learning research, 7, 1-30.
> [2] https://scikit-posthocs.readthedocs.io/en/latest/tutorial.html#critical-difference-diagrams

---

> ### Author Response · Authors · 2024-11-21
> **Response to R-XN3M (W3)**
>
> Thanks for pointing this out. We have included additional clarification in **Appendix A**:
> “Additionally, our SSL baselines use the same batch size, learning rate, input sampling frequency, and training steps as PaPaGei. We use the same augmentation types and intensity for BYOL,  SimCLR, and PaPaGei-P. Furthermore, we investigated 0.07 and 0.5 temperatures as MoCo [1] and SimCLR [2], respectively, and selected the best-performing model.”
>
> [1] He, Kaiming, et al. "Momentum contrast for unsupervised visual representation learning." Proceedings of the IEEE/CVF conference on computer vision and pattern recognition. 2020.
>
> [2] Chen, Ting, et al. "A simple framework for contrastive learning of visual representations." International conference on machine learning. PMLR, 2020.

---

> ### Author Response · Authors · 2024-11-21
> **Response to R-XN3M (Questions)**
>
> **(Q1)**
>
> This is a good point. Yes, we use single-channel PPG (we will update the paper with this detail).
>
> **(Q2)**
>
> We have improved **Figure 5** by performing the Wilcoxon Signed Rank test (non-parametric paired t-test) to evaluate statistical significance at 90% and 95% confidence. We hypothesize that adding additional data would increase the diversity of users and signals, thus improving performance consistently. Our findings in Figure 5 indicate that adding data does indeed improve performance consistently. In particular, we notice statistically significant performance gains as we increase data for MAE. We observe a similar trend for AUROC except for the VitalDB + MIMIC-III ablation. Furthermore, considering the aggregate improvements across all tasks in AUROC and MAE of 0.61 to 0.67 and 12.29 to 10.13, respectively, indicate potentially beneficial differences for the Full model.
>
> **(Q3)**
>
> You are correct to question the relevance of the Llama3 paper (Dubey et al., 2024) to our observation about the MESA dataset. We apologize for the potentially misleading comparison. We intended to simply highlight the general trend that more data often leads to better performance, which holds in various machine learning domains, including language modeling. While the Llama3 paper specifically focuses on the relationship between compute and performance, it also demonstrates that increased compute allows for the utilization of more data points (tokens), which in turn can lead to improved performance. This general principle aligns with our observation that the MESA dataset, despite having fewer participants, yielded the best performance among the single datasets, possibly due to its higher number of segments. To avoid any misinterpretation, we have revised the statement in the paper to more accurately reflect our intended meaning. We emphasize the general observation about the relationship between data size and performance without drawing a direct comparison to the specific findings of the Llama3 paper. Furthermore, recent work around scaling wearable signals hinted that increasing the signal segments (hours) results in better performance compared to increasing the number of users [1] (**Section 5.2**).
>
> [1] Narayanswamy, G., Liu, X., Ayush, K., Yang, Y., Xu, X., Liao, S., ... & McDuff, D. (2024). Scaling Wearable Foundation Models. arXiv preprint arXiv:2410.13638.
>
> **(Q4)**
>
> Thank you for your thoughtful question. We believe that **PaPaGei-S outperforms PaPaGei-P** due to the following reasons: **(1) Similarity in PPG Characteristics Across Users**: PaPaGei-S defines positive pairs based on the sVRI. Our intuition here is grounded in the observation that different users often exhibit similar PPG characteristics. By defining positive pairs this way, we encourage the model to learn inter-user similarities and form clusters of users based on shared signal properties. In contrast, PaPaGei-P encourages the model to learn a different cluster for each user. **(2) Explicit Learning of Signal Quality (SQI)**: PaPaGei-S incorporates signal quality explicitly into its training by predicting the signal quality index (SQI). This means the embeddings generated by the model inherently encode signal quality information, which is crucial for downstream tasks. In comparison, PaPaGei-P does not explicitly account for signal quality, potentially leaving the embeddings less robust to variations in signal conditions. **(3) Physiologically-Informed Multi-Objective Loss**: PaPaGei-S is trained with a multi-objective loss function that leverages domain knowledge about PPG signals. This SSL approach guides the model to learn representations that are both physiologically meaningful and robust. Contrastive learning methods that incorporate physiological insights, as highlighted in prior work ([1]), have shown significant success in improving model performance.
>
> To further substantiate our assumptions, here we describe some empirical results to describe the utility of PaPaGei-S. In Figure 7, we evaluate the inter-participant distances on the out-of-domain SDB dataset. PaPaGei-S demonstrates appropriate participant separation, which contributes to its superior performance in downstream tasks. Figure 9 highlights that PaPaGei-S is notably more robust to variations in skin tone compared to PaPaGei-P. This robustness stems from its ability to encode signal quality and inter-user similarities effectively, leading to improved performance in diverse populations. We hope this clarifies the reasoning behind the stronger performance of PaPaGei-S.
>
> [1] Gopal, B., Han, R., Raghupathi, G., Ng, A., Tison, G., & Rajpurkar, P. (2021, November). 3KG: Contrastive learning of 12-lead electrocardiograms using physiologically-inspired augmentations. In Machine Learning for Health (pp. 156-167). PMLR.

---

> ### Comment · Reviewer_XN3M · 2024-11-23
> **Response to authors**
>
> I command the authors' efforts in revising their manuscript and addressing my concerns in the short window of revision period. I have carefully read the manuscript and most of my concerns have been addressed or at least have been discussed in the paper. However, I have 3 remaining (minor) comments:
>
> 1. Upon my review, I noticed the reported numbers in Appendix Table 16 and Discussion do not match with the prior work's Table 2 top row. I suggest the authors revise these numbers to their correct values, and there seems to be a large gap and the potential  reasons need to be further discussed in the Discussion.
>
> 2. I recommend adding amount of waveform data (e.g., in units of time or any other alternative) for each row in Appendix Table 17.
>
> 3. It appears to me that the recently added analysis about comparisons with demographics (Appendix E) are not included in the main text or not even discussed. I highly recommend properly including and discussing these results in the main text to better contextualize the study results and findings.
>
> Given that these comments are straightforward to address in the manuscript, authors efforts in the presentation / implementation / rebuttal, the openness of this work and that an open weight/code PPG foundation model can foster the health research, I have raised my score contingent upon my final review of the above changes.

---

> > ### Author Response · Authors · 2024-11-25
> > **Reply to R-XN3M**
> >
> > We appreciate your recognition of our efforts and your detailed feedback. We have addressed the remaining comments as follows:
> >
> > **(C1) Comparison with prior work**: We have incorporated results from prior work regarding the patient-level positive pair strategy, now presented in **Table 16**. Additionally, we have included discussions of both strategies in a newly added subsection of the main text titled **"Effect of Demographics" (Section 5.2)**. A more detailed discussion is also provided in **Appendix E**.
> >
> > **(C2) Data in hours:** We have updated **Table 17** to include the amount of data in hours within the Subjects column. Regarding Yun et al. (2024), which uses the UK Biobank, the PPG data in that study is time-normalized, making it challenging to compute the number of hours directly.
> >
> > **(C3) Demographic results**: To elaborate on the demographic results previously discussed in Appendix E, we have introduced a new subsection in **Section 5.2** titled **Effect of Demographics**. This subsection highlights our key findings, including using demographics as a baseline, integrating demographics into PaPaGei, and predicting demographics.
> >
> > We hope these additions address your concerns. Thank you for your valuable feedback and for providing actionable suggestions to improve our work!

---

> > > ### Comment · Reviewer_XN3M · 2024-11-29
> > >
> > > I want to thank the authors for their prompt response and updates to the paper. The revised paper looks good to me!

---

### Official Review · Reviewer_MitC · 2024-11-01

**Soundness:** 3
**Presentation:** 2
**Contribution:** 1
**Rating:** 1
**Confidence:** 4

**Summary:**

This work proposes foundational models for photoplethysmography (PPG) signals. The presented model is trained more than 57,000 hours of 20 million unlabeled segments of PPG signals using publicly available datasets. Authors evaluate the model on 20 tasks of 10 diverse datasets including cardiovascular health, sleep disorders, and pregnancy monitoring.

**Strengths:**

The motivation of the works regarding the open and public foundational models with datasets is important. The paper focuses on applications of healthcare with monitoring which is relatively less studied in machine learning community. The authors perform the skin tone analysis, which can be important to evaluate fairness.

**Weaknesses:**

The technical contribution of the paper is limited. All the components, extracted features, architectures, and augmentations, of the framework are not novel. Only the integrated loss values seem novel in training but there is no clear motivational evidence for this integration. Worse, extracting those features is not trivial for noisy PPG segments, especially the sVRI and IPA as they depend on the waveform. This limits the training to clean datasets, which are collected under very limited motion, including ICUs. For example, VitalDB, one of the datasets that authors used for pre-training, is collected during surgery. This is a significant drawback in terms of the generalization of the framework.

Second, the manuscript does not explain why this framework (extracting waveform features and using them during training) performs well. For example, DaLiA is a noisy dataset where extracting those morphology features is extremely prone to error. Although DaLiA is not used for training, the manuscript does not explain why these features help extract useful representations from noisy datasets.


Third, the extracted statistical features are extremely basic (mean, median, maximum, minimum, and percentiles), but their performance is very close to the proposed model in several tasks (most confidence interval overlaps). Compared to these features with a neural network of 5M parameters, the results are not significant, which makes the usefulness of the presented work controversial.


Fourth, SimCLR and BYOL are self-supervised learning frameworks that allow someone to change the encoder architecture. In the comparison, BYOL has 12M parameters. Why not explore other networks to match the parameter count and then compare?
For example, the authors did an ablation study regarding the model size of their network and stated "As shown in 5c, the smallest model with 5M parameters performs best in all tasks except one, indicating that smaller models are better suited for PPG data." According to this statement, small models with other self-supervised baselines might perform better than the presented work. This statement by the authors raises significant questions about the fairness of the evaluation.


Fifth, some statements in the manuscript are too powerful, if they are not completely wrong. For example, the authors stated "PAPAGEI-S uses cropping (0.25) and Gaussian noise (0.25). PAPAGEI-S avoids augmentations that alter PPG’s morphology." Adding a Gaussian noise would change the morphology of the features. If the morphology refers to the waveform shape of the signal in the manuscript. All the applied transformations should have zero or linear phase response, Gaussian noise would change the phase values nonlinearly so the statement is wrong. Another statement "We propose PAPAGEI-P, a patient contrastive approach that maximizes agreement between signals from the same subject.", the subject/patient wise contrastive learning is already proposed by previous works (i.e., CLOCS,  ICML 21). Proposing a method that is already out might mislead the readers, the language of the manuscript needs a major revision.


---
----Post rebuttal----

I have decreased my rating from 3 to 1 after identifying significant technical flaws in the implementation.

**Questions:**

1) I checked the code, but I could not find the processing and evaluation of DaLiA. Could the authors specify the path to it? Thanks!

2) The authors stated that "Additionally, we did not perform an exhaustive evaluation of different augmentation settings but instead used transformations and values based on prior research  (Abbaspourazad et al., 2023; Tang et al., 2020).". The importance of the augmentations are well-known in self-supervised learning, especially depending on the application. The application of the second reference is inertial measurements instead of PPG signals. And, only the cropping and Gaussian noise are similar with the first reference. Is there a specific reason why PPG-based data augmentations are not explored?

---

> ### Author Response · Authors · 2024-11-21
> **Response to R-MitC (W1)**
>
> We appreciate the reviewer’s feedback, but we respectfully disagree with the suggestion that “All the components … are not novel”. Particularly, we believe our approach introduces an important innovation: employing morphology targets, such as sVRI and IPA, within a contrastive and multi-task learning framework. To the best of our knowledge, such inductive biases have not been previously applied to training a foundation model for biosignals. We believe this is crucial and highlights the contributions of our work (as also recognized by other reviewers). Additionally, our work contributes towards openness by conducting an extensive study on integrating datasets from multiple institutions into a unified framework, which we believe offers substantial value to the community.
>
> On PPG morphology, we clarify that our motivation is supported by the following evidence: (1) Skewness is a critical measure of signal quality (SQI), as demonstrated in [1]. As stated in Section 3.2, we use SQI to complement signals where IPA cannot be computed, addressing diverse and noisy PPG morphologies. (2) We performed a permutation test [6] to statistically evaluate PPG segments where IPA is unavailable. By splitting SQI values into no IPA and IPA groups and testing significance over 1000 permutations, we observed statistically significant differences (p < 0.05) in both mean (+0.18) and median (+0.32) SQI values, with the IPA group having larger SQI, consistent with results in [1]. These findings empirically validate SQI's ability to handle limited PPG morphology. This integration provides initial empirical insights into our framework. Additionally, IPA has demonstrated utility in estimating cardiovascular metrics like blood pressure (BP) [2], while sVRI, originally proposed for cognitive load measurement, also correlates with BP and heart rate. Our ablation study further supports this integration, as shown in Fig. 5(a) and (b), where the full model with all components outperforms individual approaches using SQI and IPA.
>
> To better contextualize the significance of our contribution, we emphasize the following:
>
> First, we fill an important gap in the research community: the absence of a **large-scale open-source study** focused on training models for PPG data. To train PaPaGei, we unify the largest openly available datasets (as highlighted in [3]): VitalDB, MIMIC-III, and MESA. In contrast, other comparable large-scale datasets, such as Apple’s AHMS and Google’s Fitbit dataset, are proprietary and not feasible for researchers to obtain. Our work addresses this critical need by offering a foundation for reproducibility in PPG analysis. We strongly believe this contribution will be useful to the research community in this domain.
>
> Second, we conduct **comprehensive generalization and out-of-domain evaluations.** Unlike most existing studies, which either do not include cross-dataset experiments or perform limited evaluations [4, 5], we evaluate our model across a diverse range of datasets. Importantly, We do not conduct a selective study, nor do we claim that our model achieves the best performance across all datasets or settings. Instead, we have expanded our evaluation to as many datasets as possible. We position our work as a significant step forward in this field, though not the final one. While we acknowledge that our model does not achieve the best performance on the DaLiA dataset, it outperforms others in 9 out of 13 out-of-domain tasks. This demonstrates the model's strong generalization ability and robustness when applied to unseen, non-ICU datasets, underscoring its practical utility across diverse real-world scenarios. Considering the popularity of the DaLiA dataset in the community, we would not want to hide this evaluation, despite the fact that it does not favor our model.
>
> [1] Elgendi, Mohamed. "Optimal signal quality index for photoplethysmogram signals." Bioengineering 3.4 (2016): 21.
>
> [2] Robb, Richard A., et al. "Proceedings of the 15th Annual International Conference of the IEEE Engineering in Medicine and Biology Society. Part 3 (of 3)." Proceedings of the 15th Annual International Conference of the IEEE Engineering in Medicine and Biology Society. Part 3 (of 3). Publ by IEEE, 1993.
>
> [3] https://peterhcharlton.github.io/post/ppg_datasets/
>
> [4] Abbaspourazad, S., Elachqar, O., Miller, A. C., Emrani, S., Nallasamy, U., & Shapiro, I. (2023). Large-scale training of foundation models for wearable biosignals. arXiv preprint arXiv:2312.05409.
>
> [5] Ding, C., Guo, Z., Chen, Z., Lee, R. J., Rudin, C., & Hu, X. (2024). SiamQuality: a ConvNet-based foundation model for photoplethysmography signals. Physiological Measurement, 45(8), 085004.
>
> [6] https://faculty.washington.edu/kenrice/sisg/SISG-08-06.pdf

---

> ### Author Response · Authors · 2024-11-21
> **Response to R-MitC (W2)**
>
> We address your concern by providing reasoning for our design choices and empirical evidence that showcases that our method generalizes across different scenarios.
> We believe that **PaPaGei-S performs well** due to the following design choices: **(1) Similarity in PPG Characteristics Across Users**: PaPaGei-S defines positive pairs based on the sVRI. Our intuition here is grounded in the observation that different users often exhibit similar PPG characteristics. By defining positive pairs this way, we encourage the model to learn inter-user similarities and form clusters of users based on shared signal properties. **(2) Explicit Prediction of Signal Quality (SQI)**: PaPaGei-S incorporates signal quality explicitly into its training by predicting the signal quality index (SQI). This means the embeddings generated by the model inherently encode signal quality information, which is crucial for downstream tasks. **(3) Physiologically-Informed Multi-Objective Loss:** PaPaGei-S is trained with a multi-objective loss function that leverages domain knowledge about PPG signals. This SSL approach guides the model to learn representations that are both physiologically meaningful and robust. Contrastive learning methods that incorporate physiological insights, as highlighted in prior work ([1]), have shown significant success in improving model performance.
>
> To further substantiate the effectiveness of PaPaGei-S, here we describe some **empirical results**. (1) In Figure 7, we evaluate the inter-participant distances on the out-of-domain SDB dataset. PaPaGei-S demonstrates **appropriate participant separation**, which contributes to its superior performance in downstream tasks. (2) In Figure 8 and other prediction plots, we notice that PaPaGei-S’s **predictions better align with the true target distributions** indicated by steeper slopes and larger R2 values. Furthermore, we notice that PaPaGei-S has **less tendency to regress to the mean** indicated by the lower peak in the Figure 8 distribution plot. This leads to **better capture of the left tail**. (3) In Figure 9, we demonstrate that PaPaGei-S is the best model for light skin tones and in the top 3 for darker skin tones. Overall, this showcases relatively better **robustness to tone variations** than other methods. (4) Finally, in comparison to other PPG studies (**Table 17**), we **evaluate 10 datasets with 7 different kinds of devices**. Our results showcase superior performance in the majority of the tasks, suggesting good robustness to noise and data shifts.
>
> [1] Gopal, B., Han, R., Raghupathi, G., Ng, A., Tison, G., & Rajpurkar, P. (2021, November). 3KG: Contrastive learning of 12-lead electrocardiograms using physiologically-inspired augmentations. In Machine Learning for Health (pp. 156-167). PMLR.

---

> ### Author Response · Authors · 2024-11-21
> **Response to R-MitC (W3)**
>
> We appreciate this feedback, and we address this concern in two ways: (1) Evaluating the significance of the statistical features and (2) Comparing against a stronger supervised baseline.
>
> **Evaluating the significance of the statistical features.**
>
> We conducted a paired t-test between the statistical features and PaPaGei-S. For both classification and regression tasks, we observed a p-value of < 0.01 with t-statistic values of 7.75 and -6.45, respectively. These results strongly demonstrate that PaPaGei-S significantly outperforms the statistical baseline. Furthermore, the t-statistic values indicate that the proposed model's performance is not close to the statistical baseline. Among the 18 tasks evaluated, only four tasks—Mood Disturbance, Arousal, Smoker, and Diastolic BP (PPG-BP)—show results that can be considered “close” to the baseline. It is important to note that cross-dataset generalization in mental health using sensor data is an inherently challenging problem. Previous studies have reported AUROC values ranging between 0.55–0.60 for various mental health tasks in similar setups [1]. Therefore, we believe it is neither surprising nor controversial that all methods, including ours, exhibit similarly low performance for Mood Disturbance and Arousal tasks under these circumstances.
>
> **Improved supervised baseline using demographics and handcrafted features.**
>
> We evaluate supervised baselines with demographics and PPG features as requested by other reviewers (https://openreview.net/forum?id=kYwTmlq6Vn&noteId=0Ol4N2R521). We conducted two key experiments for our regression (ridge) and classification (logistic regression) tasks:
>
> - **Ablation Study**: We compared PaPaGei with three baselines—demographics alone, PPG features alone, and demographics + PPG features. Our results show that while demographics is a stronger baseline than statistical features, PaPaGei outperforms the demographics + PPG baseline in 14 out of 18 tasks.
> - **Effect of Demographics**: We trained a downstream model combining PaPaGei-S embeddings with demographics. The results indicate that incorporating demographic features with PaPaGei-S creates a stronger model than using PaPaGei-S alone.
>
> These findings underscore an important point: demographic features are not competing with PaPaGei but rather complement it, as previously established in studies including demographics with sensor data [1]. This highlights the synergistic potential of combining PaPaGei's advanced feature extraction with demographic context for improved task performance. However, it is important to note that while demographic features can be valuable for personalization, they may not always be readily available, and in reality, we cannot use them in isolation to predict real-time outcomes such as blood pressure or heart rate. Therefore, our PaPaGei models are designed to function effectively with real-time sensor data alone, ensuring their applicability in situations where complete demographic information is not accessible.
>
> [1] Xu, X., Zhang, H., Sefidgar, Y., Ren, Y., Liu, X., Seo, W., ... & Dey, A. (2022). GLOBEM dataset: multi-year datasets for longitudinal human behavior modeling generalization. Advances in Neural Information Processing Systems, 35, 24655-24692.

---

> ### Author Response · Authors · 2024-11-21
> **Response to R-MitC (W4)**
>
> While we acknowledge that SSL frameworks can use different encoder architectures, we decided to match the encoder architecture and parameters, rather than the total model parameters, to maintain consistency in encoder design. For example, SimCLR and PaPaGei-P have the same parameters/architectures (5M). In contrast, BYOL and TF-C use two parallel encoders, resulting in model sizes of 12M and 9.7M. Importantly, please note that each encoder used in BYOL and TF-C  is the same as SimCLR, PaPaGei-P, and PaPaGei-S. Furthermore, the extra +0.7M in PaPaGei-S arises from the Linear layers in the multi-task heads. We believe this provides a fair basis for evaluating performance across tasks while accommodating the requirements of different SSL frameworks.

---

> ### Author Response · Authors · 2024-11-21
> **Response to R-MitC (W5)**
>
> We appreciate your thoughtful feedback. You are correct that adding Gaussian noise can introduce small deviations in the PPG waveform, which may affect its morphology. In our context, we meant to refer to IPA, SVRI, and SQI features which are more robust to Gaussian noise. Considering that our Gaussian noise is zero-mean, it introduces random variations in the signal amplitude across the waveform without systematically shifting the entire signal. For metrics such as IPA and SVRI, the addition of zero-mean Gaussian noise does not significantly affect the final computed values. This is because the noise averages out over the signal, with its mean value across different points being zero, thereby minimally impacting these aggregate measures. However, we will revise the manuscript to clarify this. Furthermore, we have acknowledged other works faithfully in the revision including CLOCS: for example, “Prior work has shown that this strategy is effective for physiological signals [CLOCS, ICML 21]” (**Section 3.1**)

---

> ### Author Response · Authors · 2024-11-21
> **Response to R-MitC (Questions)**
>
> **(Q1)**
>
> We have added the code to ‘DaLiA.ipynb’ in the repository. Please let us know if you have any questions.
>
> **(Q2)**
>
> We would like to clarify that we performed an initial investigation into different augmentations and intensities. However, we did not systematically evaluate them. [1] notes that cropping (cut out) and channel permute were the most important PPG augments. We use a strong cut-out intensity and do not use channel permute because we use single-channel PPG where we wanted to maintain temporal dependency for morphology calculation. Our initial experiments with Warping required significant time for the cubic spline relative to their performance contribution, so we did not adopt them. The augmentations that we used such as scaling, flipping, and negation are popular time series approaches. Additionally, we emphasize that the multi-task predictions of SQI and IPA in PaPaGei-S are similar to pretext tasks in SSL.
>
> To the best of our knowledge, the field of PPG-specific data augmentation remains underdeveloped, and we could not identify any comprehensive studies systematically evaluating the most effective data augmentation techniques for PPG signals. While there are examples, such as a GAN-based approach [2], we emphasize that exploring various data augmentation strategies could enhance our methodology. However, this is not the primary focus or novelty of our work. Given the limited revision time and significant computational resources required, this lies beyond our current scope and time budget. We leave this for future investigations. We added a discussion to Appendix I and discussed this direction for future research.
>
> [1] Abbaspourazad, S., Elachqar, O., Miller, A. C., Emrani, S., Nallasamy, U., & Shapiro, I. (2023). Large-scale training of foundation models for wearable biosignals. arXiv preprint arXiv:2312.05409.
>
> [2] Kiyasseh, D., Tadesse, G. A., Thwaites, L., Zhu, T., & Clifton, D. (2020). PlethAugment: GAN-based PPG augmentation for medical diagnosis in low-resource settings. IEEE journal of biomedical and health informatics, 24(11), 3226-3235.

---

> ### Comment · Reviewer_MitC · 2024-11-22
>
> I appreciate the authors' detailed response and the effort put into the revision. However, after reviewing both the responses and the updated manuscript, I find that the revision raises more concerns than it resolves.
>
> While the Friedman test and CD diagrams are reasonable tools for comparing models across tasks, the significant overlap in confidence intervals between the proposed method and the baselines reduce the robustness of the ranking differences. Rankings based solely on mean performance risk overstating the significance of small differences when confidence intervals suggest no clear separation. Even the approach that is used during revision potentially introduces bias into the evaluation. This issue is particularly problematic in this case, where the reported improvements are on the order of 0.01–0.03, accompanied by substantial overlap in confidence intervals.
>
> Also, the compared baseline models do not have any expert blocks with additional FCNs. The comparison is even unfair. How can a reader can understand if the performance improvement comes from additional FCNs or extracted features? Why not excluding the heads during the evaluation of S model such that a fair comparison can be made?
>
>
> What do the authors mean by BYOL use two parallel encoders? It uses only one encoder during inference. In the original paper [1], the authors managed to match the parameter number of models with SimCLR, also mentioned multiple times, *at the end of training, we only keep the encoder, $f_{\theta}$*. I also checked the code, I could not find BYOL implementation, authors emphasized multiple times the open-source of the work, however, most of the implementations are not given in the code.
>
> [1] Bootstrap your own latent: A new approach to self-supervised Learning. NeurIPS 2020.
>
> I do not think IPA, SVRI, and SQI features can be referred as morphology of PPG, they are the features that are affected from morphology of PPG signals. Also, how do the authors claim these features are more robust to Gaussian noise? Gaussian noise would change these features depending on the magnitude, the statement is wrong. Even, the Gaussian noise would change the location of the systolic peak so SVRI should change.
>
>
> In the revisited script Table 14, adding SQI feature to the model decreases the performance in more than half of the tasks, but the authors claim *(2) Explicit Prediction of Signal Quality (SQI): PaPaGei-S incorporates signal quality explicitly into its training by predicting the signal quality index (SQI). This means the embeddings generated by the model inherently encode signal quality information, which is crucial for downstream tasks.* The experiments and claims by the authors even contradict with each other.
>
>
> Given these contradictions with controversial/unclear baseline comparison with the marginal improvements, the discussion of novelty is unnecessary at this point for the current work. However, it is important to note that extracting (non-trivial) features (like SVRI and IPA for PPG) and forcing models to estimate them is a well-established method in the ML community [2]. More importantly, as mentioned before but could not see any answer from authors, extracting the presented features is not trivial for noisy PPG segments, especially the sVRI and IPA as they depend on the waveform. This is a significant drawback in terms of the generalization of the presented framework.
>
>
> Considering the evaluation issues (additional FCNs and unclear baseline implementations), marginal improvements, limited novelty (extracting features and training models to estimate them), and the framework’s lack of scalability (requiring clean signals for feature extraction) all require significant attention.
>
>
> While the work may hold value in more specialized contexts, its contribution in both findings and methodology is significantly limited for a general ML conference like ICLR. Therefore, I will maintain my original rating.
>
>
> [2] Masked Feature Prediction for Self-Supervised Visual Pre-Training, CVPR 2022.

---

> > ### Author Response · Authors · 2024-11-25
> > **Reply to R-MitC (1/n)**
> >
> > We appreciate the reviewer's detailed response and the effort put into the revision. We have carefully reviewed your comments and made further revisions to address them.
> >
> > **Regarding fair comparisons:**
> >
> > *“Why not excluding the heads during the evaluation of S model such that a fair comparison can be made?”*
> >
> > To ensure we are on the same page regarding the evaluation, we would like to clarify that the expert heads are **not** included in the evaluation. After the multi-task contrastive training, we evaluate the features extracted only from $P$ (the contrastive head) using a linear probe. This is analogous to BYOL where two encoders are used during training, and only one encoder ($f_{\theta}$) is used to extract features for evaluation. Therefore, we emphasize that the comparison is fair.
> > The BYOL code is available here: https://anonymous.4open.science/r/PaPaGei_ICLR_Review-6FC2/baselines/BYOL/training_byol.py
> >
> > We are happy to provide more clarification if needed. More documentation and examples will be available with the public release, as stated in our abstract: “Models, data, and code will be available upon our public release. Preliminary code for reviewing purposes is available at [LINK]”.
> >
> > **Regarding the role of SQI:**
> >
> > Our message has always been that including SQI and IPA together is the best, for example, the full model performs the best. Here, we provide excerpts from previous paper versions and rebuttal discussions.
> >
> > Response to R-MitC (W1): *“Our ablation study further supports this integration, as shown in Fig. 5(a) and (b), where the full model with all components outperforms individual approaches using SQI and IPA.”* [https://openreview.net/forum?id=kYwTmlq6Vn&noteId=WgHfyyNj5Z]
> >
> > Discussion (**Section 6**): “Ablation studies confirmed that the model with all three SSL objectives performs best, with sVRI highlighted as a key component and IPA and SQI providing positive knowledge transfer in multi-task setups.” This indicates that both IPA and SQI need to be included.
> >
> > Method (**Section 3.2**): “SQI addressing cases where IPA cannot be computed due to poor-quality signals lacking a dicrotic notch.”
> >
> > We acknowledge the need to explicitly discuss this result, and we have rephrased our language in **Section 5.2**: "Our results indicate that combining SQI and IPA yields greater benefits compared to their individual contributions."
> >
> > **Evidence and additions to acknowledge PPG feature extraction challenge:**
> >
> > We have included additional information to acknowledge challenges of PPG feature extraction in the updated paper.
> > To this end, we have added *“To address scenarios where computing IPA is challenging because of noisy signals or different morphology, we incorporate SQI. In particular, we empirically find that SQI is significantly larger ($p < 0.05$) in signals with a dicrotic notch (Appendix D.5).”*  (**Section 3.2**)
> >
> > *“Additionally, as extracting PPG features for different morphologies is non-trivial, future work will benefit from systematic evaluation of PPG features and modeling.”* (**Section 6**)
> >
> > We further acknowledge this and explain the empirical motivation behind our design choices in **Appendix D.5**

---

> > ### Author Response · Authors · 2024-11-25
> > **Reply to R-MitC (2/n)**
> >
> > **Regarding confidence intervals and ranking:**
> >
> > We acknowledge that the confidence intervals of some tasks overlap, which might introduce bias when evaluating the statistical significance of the observed differences. However, we would like to emphasize that this overlap is not consistent across all tasks. While non-overlapping CIs provide a clear indication of statistical significance, we recognize that overlapping CIs necessitate additional statistical testing. To address this, we evaluated statistical significance using critical differences across pairwise models and tasks.
> >
> > We also respectfully disagree with the notion that the ranking differences are not robust. As shown in the critical difference diagrams (**Figures 22 and 23**), PaPaGei consistently ranks as the best model across most tasks.
> >
> > We'd like to highlight that the use of critical difference diagrams for comparing multiple classifiers across multiple datasets is a well-established practice in the machine learning community. This method is specifically designed to address the issue of bias in ranking when evaluating performance across diverse tasks.
> >
> > A key advantage of critical difference diagrams is their ability to provide a statistically sound interpretation of performance differences, even when confidence intervals overlap. This is achieved by considering the average rank of each classifier across all tasks, rather than focusing solely on pairwise comparisons. This approach helps to mitigate the risk of overstating the significance of small differences in specific tasks, as it takes into account the overall performance across a broader range of scenarios.
> >
> > Moreover, the use of critical difference diagrams is supported by its strong theoretical foundation in statistical hypothesis testing. It is based on the Friedman test, a non-parametric statistical test that is widely used for comparing multiple groups on a single outcome variable. This ensures that the ranking results are not only visually informative but also statistically robust.
> >
> > In our study, the critical difference diagrams show that PaPaGei consistently outperforms the baseline models across the majority of the evaluated tasks. This provides strong evidence that our proposed method offers significant improvements in performance, even when considering the overlap in confidence intervals for specific tasks.
> > _____________
> >
> > We hope these clarifications and revisions address your concerns. We thank you for your thorough review and hope that you will consider raising your score, as we believe our work offers valuable contributions to the ICLR community.

---

> > > ### Comment · Reviewer_MitC · 2024-11-27
> > >
> > > I appreciate the authors' responses and changing the language for the role of SQI and PPG feature extraction challenge; however, some points remain unclear. I listed them below.
> > >
> > >
> > > If the authors exclude expert heads, where do the additional parameters come from? Specifically, why does the S model have 700k more parameters? Similarly, if BYOL uses only one encoder during inference, why does it have 12M parameters listed in the table? In the original BYOL paper, the authors matched the parameter count with SimCLR for a comparison. Is this also the case in this paper? If so, where is this discussed in the text? Text stated *"It is noteworthy that both BYOL and TF-C require multiple encoders and different projection heads, resulting in variations in model sizes."*. But, did the authors use this additional models during inference? If not? Why 12M parameters in BYOL?
> > >
> > >
> > > Regarding ranking and confidence intervals, I agree that ranking is a widely used and useful practice. However, I have concerns about its application in this case. For instance, when we examine Sleep Disordered Breathing, SimCLR achieves a score of 0.46 [0.21–0.64], while the proposed method scores 0.47 [0.23–0.67]. These values are extremely close, and their overlapping confidence intervals suggest no significant difference between them.
> > >
> > >
> > > My initial concern was that rankings, by design, ignore the actual magnitude of the differences between scores. As a result, close values such as 0.47 and 0.46 can lead to different rankings. When using fractional ranking methods, the method with the slightly higher score (0.47 in this case) will receive higher average ranks, regardless of the overlap in confidence intervals. Based on Appendix D.4, it seems the authors followed this ranking approach, which may explain the notable disparity in ranks (e.g., the far 0.9 rank).
> > >
> > > I suggest the authors to elaborate further on the methodology for rankings (maybe even sharing the data and code would help). Additionally, alternative ranking strategies could also be used, i.e., bootstrap ranking (resampling the scores multiple times (considering confidence intervals) and computing rankings for each sample, then averaging the results).
> > >
> > >
> > > Finally, the authors claimed, *"To the best of our knowledge, such inductive biases have not been previously applied to training a foundation model for biosignals."*. However, I referred to a paper where self-supervised learning was conducted by training a model to estimate non-trivial features, which aligns with the inductive bias described by the authors. From the authors' response, I did not see any acknowledgment or clarification regarding this point. If the contribution lies solely in the application to biosignals, I argued that this does not represent a methodological contribution to the machine learning community. Do the authors agree with this assessment? It remains unclear, as no response was provided. Moreover, since the features used in this work are already well-established in the medical domain, it appears to provide neither a novel contribution to the application nor a methodological advancement in the field of ML.
> > >
> > >
> > > I still believe that these questions and topics require attention from the authors as these are important points for readers to understand the novelty and findings of the paper.

---

> > > > ### Author Response · Authors · 2024-12-02
> > > > **Response to R-MitC (4/n)**
> > > >
> > > > **(C2) Statistical Comparisons**
> > > >
> > > > **Alternative ranking strategies:**
> > > >
> > > > We acknowledge your concern and we have conducted additional comparisons (bootstrap testing) as you recommended:
> > > > (1) We uniformly sample a score between the confidence intervals for each task across the models.
> > > > (2) We perform the critical difference ranking procedure on the sampled scores.
> > > > (3) Repeat (1) and (2) 1000 times, and average the rankings.
> > > >
> > > > We repeated the above experiment five times for **Tables 3 and 4**. The results are available here: https://imgur.com/ZnyHAQY .* indicates a significant difference between the model and PaPaGei. First, we observe that PaPaGei obtains the best average rank across all cases ranging between 0.82-0.90 and 0.19-0.25 for AUROC and MAE, respectively. Across the 35 comparisons (PaPaGei vs. rest repeated 5 times), we observe that PaPaGei obtains significant gains in 30 out of 35 AUROC comparisons and 32 out of 35 MAE comparisons. In particular, we observe that the average rank of PaPaGei ranges between 0.82-0.90 and 0.19-0.25 for AUROC and MAE, respectively. Among baselines, we notice that Chronos and TF-C are strong baselines in many tasks.
> > > >
> > > > **Clarifying potential misinterpretation of CD Diagram:**
> > > >
> > > > *“For instance, when we examine Sleep Disordered Breathing, SimCLR achieves a score of 0.46 [0.21–0.64], while the proposed method scores 0.47 [0.23–0.67].” and “As a result, close values such as 0.47 and 0.46 can lead to different rankings. When using fractional ranking methods, the method with the slightly higher score (0.47 in this case) will receive higher average ranks, regardless of the overlap in confidence intervals. Based on Appendix D.4, it seems the authors followed this ranking approach, which may explain the notable disparity in ranks (e.g., the far 0.9 rank).”*
> > > >
> > > > We would like to respectfully **correct a potential misinterpretation** in your example. The sleep-disordered breathing example you have provided is from **Table 12**, and these scores correspond to the F1-score. However, the critical difference ranking diagram (**Figures 22 & 23**) is based on **Table 3 and Table 4** (AUROC and MAE) as stated in Appendix D.4: ”In addition to confidence intervals, we perform the following steps to evaluate the significance across models on a per task basis (Tables 3 & 4)”. Therefore, this particular case **cannot explain the disparity because it does not correspond to the diagram.**
> > > >
> > > > Furthermore, it is worth noting that the average rank is computed by making pairwise comparisons across tasks. As PaPaGei outperforms many methods across tasks, it obtains a higher rank, even though it might not be the best-performing method for tasks such as Pregnancy Stage, Mood disturbance, or valence. We believe that this is a reasonable and comprehensive way to summarize model performance across datasets and tasks. We use the sample implementation as referenced in Appendix D.4 [https://scikit-posthocs.readthedocs.io/en/latest/tutorial.html#critical-difference-diagrams].

---

> ### Author Response · Authors · 2024-12-02
> **Response to R-MitC**
>
> **(C1) Fair Comparisons**
>
> We appreciate your feedback. We believe we have identified the differences between our understanding. Specifically, we report **“training parameters”** in Table 4, whereas the **BYOL paper reports “inference-time weights”**, as clarified in Section 3.1 of [1]: “When comparing to other methods, we consider the number of inference-time weights only in the final representation $f_{\theta}$.” Below, we provide a detailed explanation based on “training” and “inference” parameters.
>
> **Training Parameters:**
> - **PaPaGei-S:** Our multi-task SSL framework comprises three components. The ResNet encoder block ($E$ and $P$ in Figure 2) includes 5M parameters. And two expert blocks, $M_1$ and $M_2$, each containing fully connected layers of 350K parameters, totaling 700K parameters. Thus, PaPaGei-S has a total of 5M + 0.7M = **5.7M parameters** during training.
> - **BYOL:** The training setup employs two identical encoder blocks: one for the online encoder and another for the target encoder, each comprising 5M parameters, resulting in 5M x 2 = **10M parameters**. Additional projections for these encoders contribute ~1.7M parameters. The total training parameters for BYOL amount to approximately **12M parameters.**
> - **SimCLR:** SimCLR utilizes the same encoder block as PaPaGei-S, with a total of **5M parameters** during training.
>
> **Inference Parameters:**
> - **PaPaGei-S:** Feature extraction during inference is performed using only the encoder block, which contains **5M parameters**. Importantly, the outputs from the expert blocks ($M_1$ and $M_2$) are scalars for specific tasks ($y_{ipa}$ and $y_{sqi}$) and are not used for feature extraction.
> - **BYOL:** As noted in [1], inference employs only the online encoder ($f_{\theta}$), which has **5M parameters**, with other components discarded.
> - **SimCLR:** As SimCLR does not involve discarding components, inference also uses **5M parameters** for feature extraction.
>
> Based on these details, we emphasize that **our comparisons are fair** since the downstream evaluation of BYOL, SimCLR, and PaPaGei-S consistently uses **~5M parameters** during inference. We will provide explicit distinction between training and inference parameters with additional clarifications.
>
> [1] Grill, J. B., Strub, F., Altché, F., Tallec, C., Richemond, P., Buchatskaya, E., ... & Valko, M. (2020). Bootstrap your own latent-a new approach to self-supervised learning. Advances in neural information processing systems, 33, 21271-21284.

---

> ### Author Response · Authors · 2024-12-02
> **Response to R-MitC (5/n)**
>
> **(C3) Clarification on referred work**
>
> Our work focuses on developing an open foundation model for **optical biosignals** using a self-supervised learning (SSL) framework, incorporating PPG-specific metrics as inductive biases to predict **health outcomes**. In contrast, the aforementioned paper [1] is not directly related to our work, as it proposes an SSL framework for **videos** by predicting HOG features. We respectfully disagree with the notion that methods designed for video analysis can be directly extended to biosignal processing. Below, we outline key distinctions between these domains that highlight why such generalizations are not straightforward:
>
> - **Physiological Context:** PPG is inherently physiological, capturing vital information from an individual's body. In contrast, the paper [1] focuses on videos and is unrelated to medical tasks or biosignal processing. This fundamental difference in application domains significantly impacts the design of the model and evaluation metrics.
> - **Sampling Characteristics:** The sampling rates of PPG signals range from 64Hz to 1KHz, which is significantly more granular than the frame rates of videos (24–60 fps). This disparity necessitates the development of architectures and preprocessing techniques tailored to the high-frequency, continuous nature of biosignals.
> - **Data and Resource Constraints:** While computer vision has benefited immensely from large, web-scale standardized datasets, biosignal processing datasets are relatively minuscule and domain-specific. In our paper, we compare and contrast the available PPG datasets, highlighting the unique data challenges faced in this domain. These constraints make biosignal research significantly different.
> - **Architectural Differences:** Deep learning architectures for video and image analysis often rely on 2D or 3D convolutions, designed to capture spatial and temporal patterns. In contrast, biosignal processing typically employs 1D convolutions, which are optimized to extract features from sequential data. These architectural differences underscore the unique challenges in modeling biosignals.
>
> In summary, we want to emphasize that **developing open-source foundation models for biosignals to address real-world health problems is highly non-trivial and understudied.** This has also been acknowledged as a strength in your review: *“The paper focuses on applications of healthcare with monitoring which is relatively less studied in machine learning community.”* The challenges posed by physiological, sampling, data, and architectural differences underscore the need for domain-specific approaches. Therefore our work presents an opportunity to explore these areas, and we acknowledge that additional future work addressing these challenges is critical (as discussed in our paper).
>
> [1] Masked Feature Prediction for Self-Supervised Visual Pre-Training, CVPR 2022.

---

> ### Author Response · Authors · 2024-12-02
> **Response to R-MitC (6/n)**
>
> **(C4) Significance to the ICLR community.**
>
> As novelty is subjective, we focus on the contribution and significance of our work to the ICLR community:
> - **Practical real-world impact:** PPG is one of the most widely utilized biosignals for deriving critical medical metrics, including heart rate and blood pressure. Prior to our work, there was no foundation model available for researchers to extract embeddings or fine-tune for new tasks and scenarios, especially when working with limited data. Now, ML and healthcare researchers have a foundational tool that enables streamlined and effective exploration of PPG-based applications. We emphasize the practical implications of this as acknowledged by Reviewers DtmN (*“The framework presented can have practical implications for cardiovascular monitoring and be used for several applications”*), XN3M (*“foster the use of PPG for health applications and for health research”* & *“the openness of this work and that an open weight/code PPG foundation model can foster the health research”*), and d898 (*“PPG foundation models, in particular open/reproducible foundation models, have the potential for high impact in both health research and practical applications.”*)
> - **OOD Evaluation:** A defining characteristic of foundation models is their adaptability to a wide range of downstream tasks under diverse conditions, such as varying devices, populations, and environments. Prior work in this domain primarily focuses on in-domain evaluations using datasets collected from the same device, with limited exploration of out-of-distribution (OOD) generalization. In contrast, our work evaluates the proposed model across 20 tasks spanning 10 datasets, encompassing diverse devices, environments, and clinical populations. To the best of our knowledge, this is the most comprehensive exploration of PPG foundation models for predicting downstream tasks across datasets, addressing a critical gap in the field. This contribution is a strength of our work acknowledged by Reviewers DtmN (*“performs extensive testing on different tasks.”*).
> - **Extensive analyses:** Beyond overall performance metrics, our thorough analyses provides the community with useful findings about: (1) incorporating domain-specific features into SSL, (2) the effect of additional pre-training data, (3) downstream data-efficiency analysis, (4) the utility of demographics features, (5) impact of scaling, (6) analysis on underlying participant embeddings, (7) assessment regression models to capture distribution tails, and (8) robustness towards skin-tone. The importance of these analyses has been acknowledged by all reviewers. Reviewer MitC: *“The authors perform the skin tone analysis, which can be important to evaluate fairness.”*, Reviewer DtmN: *“It also includes ablation studies that prove the performance gains of each component.”*. Reviewer XN3M: *“I really appreciate the level of details in the paper including the ablations studies; authors did a good job for providing good level of information and detailed ablations for a curious reader.”*. Reviewer d898: *“Analysis that was performed in addition to performance reporting on downstream tasks, such as dispersion of subject embeddings and sensitivity of some tasks to skin tone.”*

---

> > ### Comment · Reviewer_MitC · 2024-12-03
> >
> > Thank you for your responses. However, they do not address my concerns and instead raise additional concerns regarding the paper's methodology, conclusions, and comparisons.
> >
> > Regarding fair comparisons, how can a reader understand the reported values are for training but not for inference?, I checked several SSL papers again, including BYOL and SimCLR, as well as time-series-specific methods like TS2Vec, and I did not observe similar ambiguities. In your script, the term "The parameter size" is referenced, but this explanation is unclear to readers if this is inference or training.
> >
> > The authors state: *"SimCLR: As SimCLR does not involve discarding components, inference also uses 5M parameters for feature extraction."* This is incorrect. SimCLR discards the non-linear projection head after training and only uses the encoder with a linear probe for inference.
> >
> > As I was curious about this statement, I checked the updated code by the authors for `training_vanilla_simclr.py`, but I noticed ResNet was initialized (Line 162) as
> > ```
> >     model = ResNet1D(in_channels=1,
> >                 base_filters=model_config['base_filters'],
> >                 kernel_size=model_config['kernel_size'],
> >                 stride=model_config['stride'],
> >                 groups=model_config['groups'],
> >                 n_block=model_config['n_block'],
> >                 n_classes=model_config['n_classes'])
> > ```
> > while the projection head for ResNet1D is defined (`models/resnet.py` Line 178) as below
> > ```
> > class ResNet1D(nn.Module):
> >     def __init__(..., use_projection=False):
> >         super(ResNet1D, self).__init__()
> >         ...
> >
> > ```
> > Despite this, I found no indication in the code where `use_projection=True` is set, either in the model initialization or elsewhere. How, then, was SimCLR trained without using projection heads? Am I overlooking a critical part of the code? I carefully reviewed it twice but could not find any relevant comments or settings for enabling `use_projection`.
> >
> > Regarding the additional bootstrap experiments, thank you for the additional experiments and clarification.
> > In my experiments (using a simulated distribution with the confidence interval and mean values), I also found that for some runs, the presented model does not consistently show significant improvements. I suggest that the authors conduct this experiment with F1 scores as well, as the results show that TF-C and BYOL perform closely to the proposed S model.
> >
> > Regarding the referred work, I appreciate the authors listing the differences; however, these appear to be application-specific and addressed with trivial solutions. The core principle—extracting features and training the model to predict them—remains the same. My reasoning for other points is outlined below.
> >
> > 1. Physiological Context:
> >    - **Claim:** The referred paper [1] deals with video analysis, whereas the authors’ work focuses on biosignals, which are inherently physiological and tied to health outcomes.
> >    - This point highlights a domain-specific application difference (biosignals vs. videos) but does not introduce a novel ML methodology. Methodologically, the principle of SSL—extracting features and forcing the model to predict them—is unchanged.
> >
> > 2. Sampling Characteristics:
> >    - **Claim:** Biosignals like PPG have higher sampling rates (64Hz to 1KHz) compared to video frame rates (24–60 fps).
> >    - The authors’ solution of resampling/downsampling is trivial and standard, offering no novel method for addressing high-frequency signals in SSL.
> >
> > 3. Data and Resource Constraints:
> >    - **Claim:** Biosignal datasets are smaller and domain-specific compared to the large, standardized datasets available in computer vision.
> >    - I disagree with this. Vision datasets often face distribution shifts, and the MIMIC itself can be larger than many datasets. Additionally, the contribution of curating three public datasets is controversial, in my opinion, in conferences like ICLR, as it is not typically viewed as a core methodological advancement.
> >
> > 4. Architectural Differences:
> >    - **Claim:** Video and image analysis often uses 2D or 3D convolutions, whereas biosignals rely on 1D convolutions optimized for sequential data.
> >    - This is a standard distinction between data modalities (spatial-temporal vs. purely temporal). Using 1D convolutions is not novel but an expected choice for sequential data like signals. There is no architectural innovations tailored to SSL for biosignals in the script. Even, one can argue that ablations regarding the architectures are poor, i.e., dilated CNNs, LSTMs, Transformers.
> >
> > Regarding the OOD, I disagree with the claim. If the same dataset is used for training the linear layer, how does this constitute an OOD evaluation?
> >
> > While the work may hold value for its specific application, the results are not significantly better than some SSL methods, and the paper offers limited methodological advancements for ML. Therefore, I believe it falls well below the ICLR threshold.

---

> ### Author Response · Authors · 2024-12-03
> **Response to R-MitC**
>
> We appreciate your response. However, we respectfully disagree with the comments that misrepresent our work.
>
> First, we are happy to address the ambiguities about “parameter size" by adding **“training parameters are reported”** in the Tables.
>
> Second, in our code, the ```use_projection``` enables us to test different projection layers. The initial projection is always present as it is necessary for training as shown in Lines 238-241 in ```resnet.py```:         ```self.dense = nn.Linear(out_channels, n_classes)```.
>
> Third, in the bootstrap ranking experiment, we observe that PaPaGei obtains significant gains in 30 out of 35 AUROC and 32 out of 35 MAE comparisons. Moreover, we have shown and acknowledged that Chronos and TF-C perform well across both classification and regression: *“Among baselines, we notice that Chronos and TF-C are strong baselines in many tasks.”*. We want to reiterate that we do not claim our model is the best across all cases. We have discussed the strengths of baselines throughout the paper. Regarding the reference to “In my experiments”, we cannot respond, as we do not have complete details or data on how the “simulated distribution” was estimated.
>
> Fourth, regarding the referred work, we respectfully clarify that **none of these are claims from our paper.** R-MitC’s previous review refers to a video analysis paper and argues that the underlying principles are similar. In our response, we outlined the key distinctions between video and biosignal domains, emphasizing why such generalizations are not straightforward. This is stated in our reply: *“Below, we outline key distinctions between these domains that highlight why such generalizations are not straightforward”* & *“Therefore our work presents an opportunity to explore these areas, and we acknowledge that additional future work addressing these challenges is critical (as discussed in our paper).”* Importantly, we clarify that these are **facts about the domain differences, not claims.** For example, our work focuses on optical biosignals from 10 publicly available PPG datasets with sampling rates ranging from 60Hz to 1KHz, as shown in Table 8. In contrast, video frame rates are generally lower, ranging from 24–60 fps, and are further downsampled during training. Additionally, computer vision (CV) benefits from extensive resources like common crawl [1], which provides large datasets, such as the 400 million image-text pairs used to train CLIP [2]. Moreover, CV has a rich variety of open datasets, and combining datasets to enhance training for foundation models is a common practice [3 (Section 2.4), 4]. For instance, [5] combines the COCO and VG datasets. Finally, 1D convolutions are typically used for time-series data such as biosignals [6].
>
> Unfortunately, the current response misunderstands these domain distinctions as claims, and the overall intent and scope of our work.
>
> Lastly, about “Regarding the OOD”, we refer to data used to train the foundation model. **Table 17** compares our work to previous large-scale PPG studies. It shows that we have compared against diverse devices, tasks, and datasets.
>
> [1] https://commoncrawl.org/
>
> [2] Radford, A., Kim, J. W., Hallacy, C., Ramesh, A., Goh, G., Agarwal, S., ... & Sutskever, I. (2021, July). Learning transferable visual models from natural language supervision. In International conference on machine learning (pp. 8748-8763). PMLR.
>
> [3] Awais, M., Naseer, M., Khan, S., Anwer, R. M., Cholakkal, H., Shah, M., ... & Khan, F. S. (2023). Foundational models defining a new era in vision: A survey and outlook. arXiv preprint arXiv:2307.13721
>
> [4] Madan, N., Møgelmose, A., Modi, R., Rawat, Y. S., & Moeslund, T. B. (2024). Foundation Models for Video Understanding: A Survey. arXiv preprint arXiv:2405.03770
>
> [5] Chen, Y. C., Li, L., Yu, L., El Kholy, A., Ahmed, F., Gan, Z., ... & Liu, J. (2020, August). Uniter: Universal image-text representation learning. In European conference on computer vision (pp. 104-120). Cham: Springer International Publishing.
>
> [6] Abbaspourazad, S., Elachqar, O., Miller, A. C., Emrani, S., Nallasamy, U., & Shapiro, I. (2023). Large-scale training of foundation models for wearable biosignals. arXiv preprint arXiv:2312.05409.

---

### Official Review · Reviewer_DtmN · 2024-11-03

**Soundness:** 4
**Presentation:** 4
**Contribution:** 3
**Rating:** 8
**Confidence:** 4

**Summary:**

This paper proposes two open foundation models for PPG signals, PAPAGEI-P (patient-aware) and PAPAGEI-S (morphology-aware). These models mainly differ in their training strategy—one uses a self-supervised approach to maximize agreement between embeddings of the same patient and the other maximizes the agreement between signal features across patients. The two models are tested on several datasets to evaluate their performance on different tasks, including binary and multi-class classification and regression, and they are compared with other general foundation models and SSL approaches. Ablation studies are carried out to understand the performance impact of the datasets used for pre-training, the impact of each model component, the effect of different levels of labeled data, and model size and scaling. Some case studies are also presented to evaluate different subject- and data-related aspects.

**Strengths:**

1. The paper claims to propose the first open foundation model for PPG signals, which tries to improve the current models' performances using publicly available data and data augmentation, while also being relatively light-weight (~5M parameters)
2. The paper is well-organized, rich in technical details regarding the data preprocessing, model architecture and training, and performs extensive testing on different tasks. It also includes ablation studies that prove the performance gains of each component.
3. The methods and results are well explained and covered, and are accompanied by relevant tables and figures to facilitate comprehension and readability
4. The framework presented can have practical implications for cardiovascular monitoring and be used for several applications

**Weaknesses:**

1. While performing slightly better in some tasks relative to other foundation and SSL models, the proposed models lack comparison with task-specific non-SSL models. Despite including statistical feature models for baseline, which can give a basic comparison, state-of-the-art task-specific models, with or without engineered features, are not included. Such comparison would provide insights into whether the SSL approach yields better representations than supervised-learning ones.
2. Task performances are evaluated using AUROC and MAE, but the manuscript doesn't seem to comment on the F1 scores and R2 in some tasks, which are often related to class imbalance and yield poor results. Addressing this topic or proposing a strategy to handle these cases would be appreciated.

**Questions:**

1. Do you have any suggestions to improve the model's performance in tasks where performance is low or when class imbalance is an issue?
2. Regarding the out-of-domain datasets, do they all use different devices from the ones in the pretraining datasets?
3. And have you looked into other biases that could be present in the datasets (sex, age, gender, weight, health conditions)?

Some other suggestions:
- Report the train and validation training loss curves for both models would help understand the convergence stability in both approaches
- Make sure to reference all the appendices in the main text

---

> ### Author Response · Authors · 2024-11-21
> **Response to R-DtmN (W1)**
>
> Thank you for highlighting this. We have added a supervised baseline experiment with demographics and hand-crafted PPG signals. (https://openreview.net/forum?id=kYwTmlq6Vn&noteId=0Ol4N2R521)
>
> Our new regression (ridge) and classification (logistic regression) tasks experiments (added in **Appendix E, Table 15**) present the following findings:
>
> - **Ablation Study**: We compared PaPaGei with three baselines—(1) demographics alone, (2) hand-crafted PPG morphology features alone, and (3) demographics + hand-crafted PPG morphology features. Our results show that while demographic features create a stronger baseline than statistical features, PaPaGei outperforms the demographics + PPG baseline in 14 out of 18 tasks.
> - **Effect of Demographics**: We trained a model combining PaPaGei-S with demographics. The results indicate that incorporating demographic features with PaPaGei-S creates a stronger model than using PaPaGei-S alone.
> These findings highlight an important point: demographic features are not competing with PaPaGei but rather complement it, as previously established in studies including demographics with sensor data [1]. This highlights the complementary potential of combining PaPaGei's extracted features with demographic context for improved task performance. However, it is important to note that while demographic features can be valuable for personalization, they may not always be readily available, and in reality, we cannot use them in isolation to predict real-time outcomes such as blood pressure or heart rate. Therefore, our PaPaGei models are designed to function effectively with real-time sensor data alone, ensuring their applicability even in situations where complete demographic information is not accessible.
>
> [1] Spathis, D., Perez-Pozuelo, I., Gonzales, T. I., Wu, Y., Brage, S., Wareham, N., & Mascolo, C. (2022). Longitudinal cardio-respiratory fitness prediction through wearables in free-living environments. NPJ Digital Medicine, 5(1), 176.

---

> ### Author Response · Authors · 2024-11-21
> **Response to R-DtmN (W2)**
>
> Thank you for your feedback. We have included the $R^2$ metric in **Figure 8**. As shown in Figure 8, PaPaGei-S demonstrates steeper slopes and higher $R^2$ values for both the AHI prediction task ($R^2$=0.29 compared to 0.18 and 0.16). Additional regression plots are available in Figure 24. These results highlight the superior performance of PaPaGei-S in regression tasks. Moreover, the higher $R^2$ values indicate better alignment between the true and predicted values, further validating the quality of the model's predictions.
>
> Furthermore, we have added the following to **Section 6**. To assess performance under class imbalance, we examined the F1-score, a robust metric in such cases. PaPaGei achieves the highest F1 in 6 out of 9 classification tasks, demonstrating its effectiveness in handling data imbalance. For regression, PaPaGei-S achieves the highest $R^2$ in 7 tasks (**Appendix D**), reflecting better alignment with the true distribution. These results highlight the robustness and versatility of PaPaGei-S across classification and regression tasks.
>
> We believe the addition of this result and discussion provides further clarity.

---

> ### Author Response · Authors · 2024-11-21
> **Response to DtmN (Questions)**
>
> **(Q1)**
>
> Thank you for your insightful suggestion. Presently, we evaluate the downstream performance using a simple logistic regression model, following the linear probing paradigm. To address low performance or class imbalance, domain generalization (DG) and domain adaptation (DA) approaches could be quite useful. While DA requires access to the downstream dataset, DG methods use losses during training to make models more generalizable. The biosignal processing community can use our models to achieve this in future work. To this end, we have added a discussion in **Appendix I** providing an example:
>
> For instance, let’s consider the nuMoM2B dataset which consists of pregnant women. PaPaGei-S obtains an AUROC of 0.78 in pregnancy stage classification and 6.05 in gestation age classification. Compared to the pre-training population with diverse ages and gender, the nuMoM2B consists of women generally aged between 20-35. Furthermore, the gestation age readings are collected approximately around the first and third trimesters. Given these factors, the target nuMoM2B dataset has many variables contributing toward the distribution shift. Therefore, PaPaGei-S can be fine-tuned to address the shift in the following ways: (1) We can align the pre-trained embeddings to the nuMoM2B embedding using unsupervised or semi-supervised domain adaptation. (2) Domain Generalization is also an option during the training phase to improve generalization robustness. (3) Newer methods such as LoRA can provide another way to quickly fine-tune. (4) Importantly, given that more women are present on the first visit compared to the third visit (Figure 13), we can optimize different metrics to improve accuracy under the imbalance. For example, AUPRC can be optimized instead of AUROC. Fairness of classification across genders can also be considered during training. Exploring these avenues to further enhance the performance and applicability of PaPaGei is a promising direction for future studies.
>
> **(Q2)**
>
> You're correct. Most of the out-of-domain (OOD) datasets utilize different devices for PPG signal collection compared to the devices used in our pre-training datasets. This difference in devices is a key factor contributing to the OOD nature of these datasets, as it introduces variations in signal characteristics, hardware calibration, and potential biases. There's one exception: the nuMoM2B dataset. This dataset, which focuses on pregnant women, uses a Nonin pulse oximeter device, similar to the one used in the MESA dataset, which was part of our pre-training data. A caveat is that MESA uses a sampling rate of 256Hz whereas nuMoM2B uses 75Hz. This overlap in devices presents an interesting opportunity to analyze how PaPaGei performs when the device type remains consistent while other factors, such as the study population and physiological conditions, vary. To highlight this aspect, we will add a note in the revised manuscript specifying the devices used in each OOD dataset and their relation to the pre-training datasets. This clarification will provide readers with a clearer understanding of the OOD evaluation and the potential influence of device variability on model performance. To further expand on devices, we have a new **Table 17** comparing different PPG studies which includes no. of devices and types.
>
> **(Q3)**
>
> We acknowledge that potential biases may exist in the datasets beyond just skin tone, arising from factors like sex/gender imbalances, age skewness towards older populations, and inclusion of participants with specific health conditions. These factors may not be representative of the general population. This was the reason we included a detailed description of each dataset and its most important features in Appendix B. To mitigate these biases, we are actively working on diversifying our training data, evaluating PaPaGei on a wider range of OOD datasets, and exploring fairness metrics during training. Additionally, we have evaluated PaPaGei’s ability to predict demographic targets through age regression, age classification, and sex classification. We compare these results to the [1] in **Table 16, in Appendix E**. Furthermore, we comment on this in the main text in Section 6 as follows: “PaPaGei-S achieves 7.78 MAE in age regression, 0.85 accuracy in age classification, and 0.79 accuracy in sex classification, advancing open-source efforts despite trailing larger closed studies [1] by 2.18, 0.05, and 0.13, respectively.” We hope this answers some of your questions.

---

> ### Author Response · Authors · 2024-11-25
> **Following up with R-DtmN**
>
> Hi R-DtmN,
>
> We hope this message finds you well. This is just a follow up on our response to your review. We have carefully addressed your comments and made revisions to the manuscript accordingly. We would be grateful if you could acknowledge our rebuttal, and we are happy to answer any further questions or concerns you might have.
>
> Thank you for your time and consideration.

---

> ### Author Response · Authors · 2024-12-01
> **A gentle reminder**
>
> Dear Reviewer DtmN,
>
> Thank you for taking the time to review our work. We sincerely appreciate your valuable feedback and have revised the paper to incorporate your suggestions. Given the approaching deadline, we would be grateful if you could acknowledge our revisions.

---

### Author Response · Authors · 2024-11-21
**Summary of Changes (1/n)**

We would like to thank all reviewers for their thoughtful feedback. We are encouraged that the reviewers agreed with the strengths of our work:

 - [R-DtmN]: “The paper is well-organized, rich in technical details regarding the data preprocessing, model architecture and training, and performs extensive testing on different tasks.”
- [R-XN3M]: “I really appreciate the level of details in the paper including the ablations studies; authors did a good job for providing good level of information and detailed ablations for a curious reader.”
- [R-d898]: “Furthermore the authors should be commended for two important things which, combined, have not been published previously: They performed all training and downstream evaluation using data sets that are publicly available, enabling straightforward reproducibility and verification of their results (as well as use of their models by other researchers for future work)” & “in addition to comparing the performance of multiple pre-trained time series foundation models, as well as adapting different contrastive and momentum-based SSL methods, the authors have also incorporated some PPG-specific elements into the pre-training (loss functions for SQI, sVRI and IPA) that have not been reported elsewhere.”

We are grateful for the positive comments.

Importantly, we thank the reviewers for their insightful suggestions. By incorporating these valuable inputs, we have made several improvements and additions to the manuscript. A **revised version** of the manuscript has been uploaded, with key text changes highlighted in **blue** and updates to figures and tables indicated by **yellow-caption highlights**. While detailed responses to each reviewer’s suggestions are directly addressed below the respective reviews, here we briefly summarize the key changes:

**(I) Additional experiments that include demographics and (non-generic) PPG-specific features for downstream classification (Appendix E, Table 15).** To address reviewers’ concerns about the supervised baseline, we ran a dedicated experiment (with a portion of each dataset as a held-out test set) on downstream targets using as inputs demographic features (age, sex) and PPG features (sVRI, IPA, SQI), for both regression (ridge) and classification (logistic regression) tasks. Findings:

- **Ablation Study**: We compared PaPaGei to three baselines—(i) demographics alone, (ii) new PPG features alone, and (iii) demographics + new PPG features. Our results show that **while demographics is a stronger baseline than statistical features, PaPaGei outperforms the demographics + PPG baseline in 14 out of 18 tasks.**
- **Effect of Demographics:** We trained a model combining PaPaGei-S with demographics. The results indicate that incorporating demographic features with PaPaGei-S creates a stronger model than using PaPaGei-S alone. These findings highlight an important point: **demographic features are not competing with PaPaGei but rather complement it. Therefore, we would like to emphasize the complementary potential of combining PaPaGei's advanced feature extraction with demographic context for improved health monitoring.**

**(II) Comparison to prior work: Language, Study characteristics, & Demographic targets.** To better contextualize our work and better represent prior work, we have made the following changes:
- We have added **Table 17** to compare different large-scale PPG studies.
- Added text in the related work section to better highlight the similarities and differences between our work compared to previous studies (**Section 2**).
- Discussed other foundation models for physiological signals (**Appendix H**).
- Added additional text to compare and contrast the methodological and design choices between PaPaGei and other works (**Section 3.1, 3.2, 4.1, Appendix A**).
- Added additional details regarding training and architectural choices to ensure fair comparisons (**Appendix A**).

**(III) Effectiveness of PaPaGei compared to baselines and performing statistical significance tests.** To address comments regarding the advantages of PaPaGei compared to baselines, we have made the following changes:
- Included discussion on tasks where general-purpose time series FM performance is comparable to PaPaGei (**Section 5.1**).
- Included new ‘critical difference’ diagrams to evaluate the statistical significance of models across all tasks based on their average performance rank. Our findings indicate that PaPaGei achieves statistically significant gains compared to other methods in most cases (**Appendix D.4 and Figures 22 and 23**).
- Added statistical significance tests for ablation studies on PaPaGei components (**Figure 6**) and Pre-training datasets (**Figure 5**).

---

> ### Author Response · Authors · 2024-11-21
> **Summary of Changes (2/n)**
>
> **(IV) Redesigned figures with additional information:** As requested, to provide additional evidence that our predictions match the true distributions, we have improved the regression prediction results to include slope ($m$) and $R^2$. Furthermore, we also overlaid the baselines and PaPaGei prediction distributions with the true distribution (**Figure 8**).
>
> **(V) References to additional metrics:** To address comments related to class imbalance and generalization, we have included a discussion in **Section 6**. Furthermore, as requested, to provide an alternative to aggregating overall regression performance across tasks, we have included the symmetric mean absolute percentage error (sMAPE) metric (**Tables 3 & 4**).
>
> **(VI) Improved discussion:** To provide users with useful insights, we have extended the discussion to include suggestions for future research such as fine-tuning models for distribution shifts and systematic evaluation of PPG-specific augmentations (**Appendix I**).
>
> **(VII) Reorganized sections:** We have reorganized the ablation study section to improve readability between text and figures (**Section 5.2**).

---

> ### Author Response · Authors · 2024-11-25
> **Summary of Changes (3/n)**
>
> **(VIII) Ablation study on demographic analysis**: We have added a new section **"Effect of Demographics"** in **Section 5.2** of the main text. This highlights our analysis and findings of the demographic analysis.
>
> **(IX) Additional statistical motivation of design choices**: To address questions regarding inclusion of model components, we provide our statistical motivation analysis comparing SQI and IPA in **Appendix D.5**

---

### Meta-Review · Area_Chair_1wwu · 2024-12-23

**Metareview:**

This paper was reviewed by four experts in the field and received 8, 1, 8, 8 as the final ratings. The reviewers acknowledged that this work presents the first open-source foundation model for PPG, trained on open datasets with open-source code, the methods and results are well explained, and are accompanied by relevant tables and figures, and that the proposed method can have practical implications for cardiovascular monitoring and be used for several applications.

The reviewers mentioned that the F1 score and $R^{2}$ should also be used as evaluation metrics, besides AUROC and MAE. In the rebuttal, the authors have analyzed the performance of PapaGei in terms of the F1 score and $R^{2}$, demonstrating its robustness and versatility for classification and regression tasks. It was also mentioned that p-values and statistical tests of significance should be provided to evaluate the proposed method against the baselines. In the rebuttal, the authors have performed the Wilcoxon Signed Rank test to evaluate the statistical significance of PaPaGei at 90% and 95% confidence levels. Other queries regarding discussion on foundation models for physiological signals, evaluation of demographic characteristics, and the reason for PaPaGei-S outperforming PaPaGei-P have all been addressed convincingly by the authors in the rebuttal.

Reviewer MitC had raised a couple of concerns about the implementation of the proposed method. This was discussed extensively with the authors during the rebuttal period and also between the reviewers and the AC during the post-rebuttal discussion period. The main concerns were the following: (i) the projection head of SimCLR is non-linear, however, that has not been used in the implementation; and (ii) after the training of SimCLR, the non-linear projection layers should be discarded from the architecture and only the encoder should be used during inference; however, these components were not discarded, as per the implementation. Reviewer XN3M also agreed that the implementation does not follow the SimCLR algorithm rigorously. However, this reviewer also mentioned that the comparison across the baselines is still fair, since the same implementation strategy was used for all the baseline methods as well. The only issue is that it may not be appropriate to refer to this as "SimCLR", since some of the algorithmic details of the vanilla SimCLR were not rigorously followed in the implementation. The AC agrees with this argument and does not consider this to be a reason to reject the paper.

Reviewers DtmN, XN3M and d898 have a strong positive opinion about the paper and its contributions. Reviewer XN3M has even commended the authors' efforts in revising their paper and addressing his concerns within a short time window. Based on the reviewers' feedback, the decision is to recommend the paper for acceptance to ICLR 2025. The authors are encouraged to carefully look into the issues mentioned by Reviewer MitC (discussed above) and revise their writing accordingly for the final version of the paper. We congratulate the authors on the acceptance of their paper!

**Additional Comments On Reviewer Discussion:**

Please see my comments above.

---

### Decision · Program_Chairs · 2025-01-22

Accept (Poster)